# URLOST: Unsupervised Representation Learning without Stationarity or Topology

**Zeyu Yun**[1], **Juexiao Zhang**[3], **Yann Lecun**[3,4], **Yubei Chen**[2]
[1] UC Berkeley, [2] UC Davis, [3] New York University, [4] FAIR at Meta

## Abstract

Unsupervised representation learning has seen tremendous progress. However, it is constrained by its reliance on domain specific stationarity and topology, a limitation not found in biological intelligence systems. For instance, unlike computer vision, human vision can process visual signals sampled from highly irregular and non-stationary sensors. We introduce a novel framework that learns from high-dimensional data without prior knowledge of stationarity and topology. Our model, abbreviated as URLOST, combines a learnable self-organizing layer, spectral clustering, and a masked autoencoder (MAE). We evaluate its effectiveness on three diverse data modalities including simulated biological vision data, neural recordings from the primary visual cortex, and gene expressions. Compared to state-of-the-art unsupervised learning methods like SimCLR and MAE, our model excels at learning meaningful representations across diverse modalities without knowing their stationarity or topology. It also outperforms other methods that are not dependent on these factors, setting a new benchmark in the field. We position this work as a step toward unsupervised learning methods capable of generalizing across diverse high-dimensional data modalities. Code is available at this repository.

## 1 Introduction

Unsupervised representation learning aims to develop models that autonomously detect patterns in data and make these patterns readily apparent through specific representations. Over the past few years, there has been tremendous progress in the unsupervised representation learning community, especially self-supervised representation learning (SSL) method. Popular methods SSL methods like contrastive learning and masked autoencoding [87; 12; 33; 90] work well on typical data modalities such as images, videos, time series, and point clouds. However, these methods make implicit assumptions about the data domain's **topology** and **stationarity**. **Topology** refers to the low-dimensional structure arisen from physical measurements, such as the pixel grids in images, the temporal structures in time series and text sequences, or the 3D structures in molecules and point clouds [7]. **Stationarity** refers to the characteristic that the statistical properties of the signal are invariant across its domain [6]. For instance, the statistics of pixels and patches in images are invariant to their spatial locations. A vase is still a vase no matter placed at the corner or the center of the image.

The success of state-of-the-art self-supervised representation learning (SSL) methods largely depends on these two crucial assumptions. For example, in computer vision, popular SSL techniques, such as Masked Autoencoders [34] and joint-embedding methods like SimCLR [12] require the construction of image patches. Both approaches typically rely on convolutional neural networks (CNNs) or Vision Transformer (ViT) backbones [22]. These backbones consist of shared-weight convolutional filters or linear layers, which inherently exploit the stationarity and regular topology present in natural images. The geometric deep learning community has made significant efforts to extend machine learning to domains beyond those with regular topology. However, graph neural network (GNN)-based methods empirically struggle to scale with self-supervised learning objectives and large datasets [69; 10; 9]. Methods that do scale well with large data still assume a minimal level of stationarity and regular topology [32].

What if we have high-dimensional signals without prior knowledge of their domain topology or stationarity? Can we still craft a high-quality representation? This is not only the situation that

biological intelligence systems have to deal with but also a practical setting for many scientific data analysis problems. Taking images as an example, computer vision system takes in signal from digital cameras. Biological visual systems, on the other hand, have to deal with signals with less domain regularity. Unlike the uniform grid in camera sensors, the cones and rods in the retina are distributed unevenly and non-uniformly. Yet, biological visual systems can establish a precise retinotopy map from the retina to neurons in visual cortex based on spontaneous, locally-propagated retinal activities and external stimuli [86; 47; 25] and leverage retinotopic input to build unsupervised representations. This motivates us to build unsupervised representations without relying on the prior stationarity of the raw signal or the topology of the input domain. The ability to build unsupervised representations without relying on topology and stationarity has huge advantages. For example, it allows humans to utilize irregular sensors for both high resolution and broad coverage, which is much more efficient than a regular camera sensor for dynamic environments. Developing such a model also allows us to create a powerful AI system that computes any high-dimensional signal.

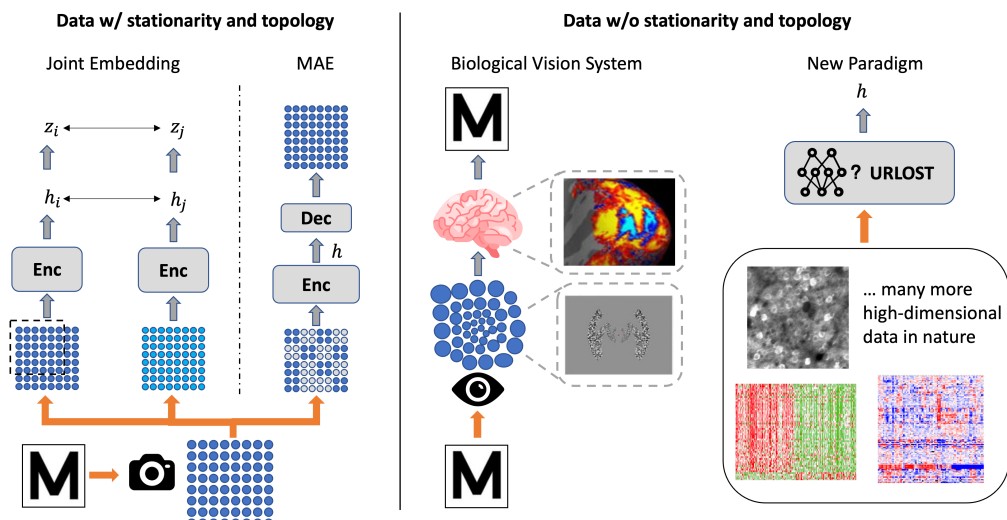

Figure 1: From left to right: the unsupervised representation learning through joint embedding and masked auto-encoding; the biological vision system that perceives via unstructured sensor and understands signal without stationarity or topology [61]; and many more such diverse high dimensional signal in natural science [57; 89] that our method supports while most existing unsupervised methods don't.

In this work, we aim to build unsupervised representations for general high-dimensional data and introduce *unsupervised representation learning without stationarity or topology* (**URLOST**). Taking images as an example again, let's assume we receive a set of images whose pixels are shuffled in the same order. In this case, the original definition of topology of images is destroyed, i.e. each pixel should no longer the neighbor of the pixels that's physically next to it. How can we build representations in an unsupervised fashion without knowledge of the shuffling order? If possible, can we use such a method to build unsupervised representations for general high-dimensional data? Inspired by Roux et al. [67], we use low-level statistics and spectral clustering to form clusters of the pixels, which recovers a coarse topology of the input domain. These clusters are analogous to image patches except that they are slightly irregularly shaped and different in size. We mask a proportion of these "patches" and utilize a Vision Transformer [22] to predict the masked "patches" based on the remaining unmasked ones. This "learning to predict masked tokens" approach is proposed in masked autoencoders (MAE) [34] and has demonstrated effectiveness on typical modalities. Firstly, we test the proposed method on the synthesized biological visual dataset, derived from the CIFAR-10 [44] using a foveated retinal sampling mechanism [14]. Then we generalize this method to two high-dimensional vector datasets: a primary visual cortex neural response decoding dataset [73] and the TCGA miRNA-based cancer classification dataset [79; 84]. Across all these benchmarks, our proposed method outperforms existing SSL methods, demonstrating its effectiveness in building unsupervised representations for signals lacking explicit stationarity or topology. Given the emergence of new modalities in deep learning from natural sciences [75; 31; 62; 46; 85; 17; 2], such as

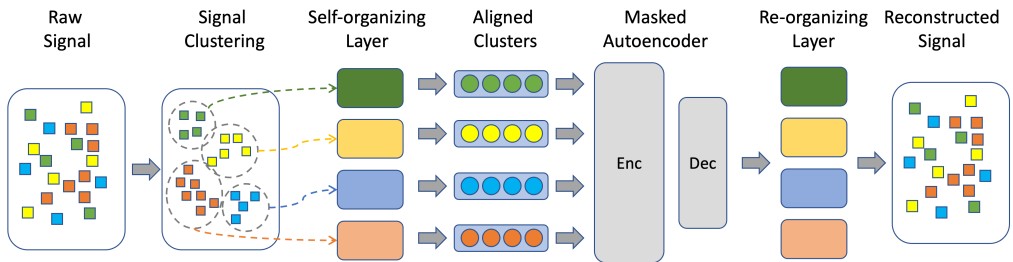

Figure 2: **The overview framework of URLOST.** The high-dimensional input signal undergoes clustering and self-organization before unsupervised learning using a masked autoencoder for signal reconstruction.

chemistry, biology, and neuroscience, our method offers a promising approach in the effort to build unsupervised representations for high-dimensional data.

## 2 METHOD

### 2.1 MOTIVATION AND OVERALL FRAMEWORK

Our objective is to build robust unsupervised representations for high-dimensional signals without prior information on explicit topology and stationarity. The learned representations are intended to enhance performance in downstream tasks such as classification. We begin by using low-level statistics and clustering to approximate the topology of the signal domain. The clusters then serve as input to a masked autoencoder. As depicted in Figure 1, the masked autoencoder randomly masks out patches in an image and trains a Transformer-based autoencoder unsupervisedly to reconstruct the original image. After unsupervised training, the latent state of the autoencoder yields high-quality representations. In our approach, the clusters derived from the raw signal are input to the masked autoencoder.

Taking images as an example data modality, clusters formed in images are bags of pixels, which differs from image patches in several key aspects. First, they are unaligned, exhibit varied sizes and shapes. Second, each pixel in a cluster is not confined to fixed 2D locations like pixels in image patches. To cope with these differences, we introduce a self-organizing layer responsible for aligning these clusters through learnable transformations. The parameters of this layer are jointly optimized with those of the masked autoencoder. Our method is termed URLOST, an acronym for **U**nsupervised **R**epresentation **L**earning with**O**ut **S**tationarity or **T**opology. Figure 2 provides an overview of the framework. URLOST consists of three core components: density adjusted spectral clustering, self-organizing layer, and masked autoencoder. The functionalities of these components are detailed in the following subsections.

### 2.2 DENSITY ADJUSTED SPECTRAL CLUSTERING

Given a dataset $S \in \mathbb{R}^{n \times m}$, where each row denote a high dimensional data point. Let $S_i \in \mathbb{R}^{n \times 1}$ be $i$th column of $S$, which represents the $i$th dimension of the signal. Since we want to process dimensions that share similar information together, we use spectral clustering to group these dimensions. Let $A_{ij} = I(S_i; S_j)$ be the mutual information between dimension $i$ and $j$. Consider each dimension as a node of a graph and $A$ is the the affinity matrix of a graph. $L = D - A$ is the graph Laplacian, where D is the diagonal matrix whose $(i,i)$-element is the sum of $A$'s $i$-th row. We can formulate the spectral embedding problem as finding $Y$ such that $\min_{YY^T=I} tr(YLY^T)$. A clustering algorithm is then applied to the embedding $Y$ as explained in appendix A.2. The size and shape of the clusters strongly affect the unsupervised learning performance. To adjust the size and shape, we apply a density adjustment matrix $P$ to adjust $L$ in the spectral embedding objective. The optimization problem becomes the following:

$$\min_{YY^T=I} tr(YP^{1/2}LP^{1/2}Y^T) \tag{1}$$

where $P = diag(p(i))$, $p(i)$ is the unnormalized density function defined on each node $i$. We set $p(i) = q(i)^\alpha n(i)^{-\beta}$, where $n(i) = \sum_{j \in \text{Top}_K(A_{ji})} A_{ji}$ and $q(i)$ is the prior density which depends

on specific dataset and is defined in the experiment section. $\alpha$, $\beta$ and $K$ are hyper-parameters. Setting $\alpha = 0$ and $K = m$ will recover the normalized graph Laplacian. In appendix A.1, we provide a detailed interpretation and motivation of density adjustment. We further verify its effectiveness with ablation study in section 4.3 and appendix A.4

### 2.3 SELF-ORGANIZING LAYER

Transforming a high-dimensional signal into a sequence of clusters using the above method is not enough because it does not capture the internal structure within individual clusters. As an intuitive example, given an image, we divide it into a set of image patches of the same size. If we apply different permutations to these image patches, their inner product will no longer reflect their similarity properly. Clusters we obtained from section 2.2 are analogous to image patches, but elements in each cluster have arbitrary ordering. Thus, the inner product between two clusters is also arbitrary due to the ordering mismatch. In Transformers, since self-attention depends on the inner products between different "clusters," we need to align these clusters in a space, where their inner products reflect their similarity. To align these clusters, we propose a *self-organizing layer* with learnable parameters. Specifically, let vector $x^{(i)}$ denote the $i$th cluster. Each cluster $x^{(i)}$ is passed through a differentiable function $g(\cdot, w^{(i)})$ with parameter $w^{(i)}$, resulting in a sequence $z_0$:

$$z_0 = [g(x^{(1)}, w^{(1)}), \cdots g(x^{(M)}, w^{(M)})] \tag{2}$$

$z_0$ is comprised of projected and aligned representations for all clusters. The weights of the proposed self-organizing layer, $\{w^{(1)}, \cdots w^{(M)}\}$, are jointly optimized with the subsequent neural network introduced in the next subsection.

### 2.4 MASKED AUTOENCODER

After the self-organizing layer, $z_0$ is passed to a Transformer-based masked autoencoder (MAE) with an unsupervised learning objective. Masked autoencoder (MAE) consists of an encoder and a decoder which both consist of stacked Transformer blocks introduced in Vaswani et al. [82]. The objective function is introduced in He et al. [34]: masking random image patches in an image and training an autoencoder to reconstruct them, as illustrated in Figure 1. In our case, randomly selected clusters in $z_0$ are masked out, and the autoencoder is trained to reconstruct these masked clusters. After training, the encoder's output is treated as the learned representation of the input signal for downstream tasks. The masked prediction loss is computed as the mean square error (MSE) between the values of the masked clusters and their corresponding predictions.

## 3 RESULT

Since the biological vision system inspires our method, we first validate its ability on a synthetic biological vision dataset created from CIFAR-10. Then we evaluate the generalizability of URLOST on two high-dimensional natural datasets collected from diverse domains. Detailed information about each dataset and the corresponding experiments is presented in the following subsections. Across all tasks, URLOST consistently outperforms other strong unsupervised representation learning methods. For all the experiments in this work, we use linear layer to parametrize the self-organizing layer, i.e. $g(x, W^{(i)}) = W^{(i)}x$. Additionally, we provide training hyperparameters and experiments for effect of hyperparameters in Appendix A.6.

### 3.1 SYNTHETIC BIOLOGICAL VISION DATASET

As discussed in the introduction, the biological visual signal serves as an ideal data modality to validate the capability of URLOST. In contrast to digital images captured by a fixed array of sensors, the biological visual signal is acquired through irregularly positioned ganglion cells, inherently lacking explicit topology and stationarity. However, it is hard to collect real-world biological vision signals with high precision and labels to evaluate our algorithm. Therefore, we employ a retinal sampling technique to modify the classic CIFAR-10 dataset and simulate imaging from the biological vision signal. The synthetic dataset is referred to as *Foveated CIFAR-10*. To make a comprehensive comparison, we also conduct experiments on the original CIFAR-10, and a *Permuted CIFAR-10* dataset obtained by randomly permuting the image.

**Permuted CIFAR-10.** To remove the grid topology inherent in digital imaging, we simply permute all the pixels within the image, which effectively discards any information related to the grid structure of the original digital image. We applied such permutation to each image in the CIFAR-10

dataset to generate the *Permuted CIFAR-10* dataset. Nevertheless, permuting pixels only removes an image's topology, leaving its stationarity intact. To obtain the synthetic biological vision that has neither topology nor stationarity, we introduce the *Foveated CIFAR-10*.

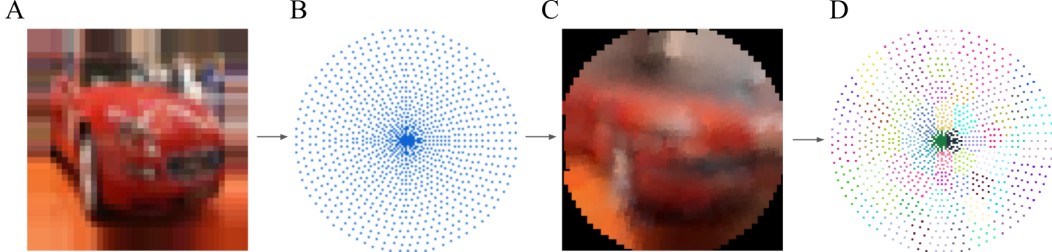

Figure 3: **Retina sampling** (A) An image in CIFAR-10 dataset. (B) Retina sampling lattice. Each blue dot represents the center of a Gaussian kernel, which mimics a retinal ganglion cell. (C) Visualization of the car image's signal sampled using the retina lattice. Each kernel's sampled RGB value is displayed at its respective lattice location for visualization purposes. (D) density-adjusted spectral clustering results are shown. Each unique color represents a cluster, with each kernel colored according to its assigned cluster.

**Foveated CIFAR-10.** Unlike cameras that use uniform photosensors with consistent sampling patterns, the human retina features ganglion cells whose receptive field sizes vary—smaller in the central fovea and larger in the periphery. This results in foveated imaging [80], enabling primates to have both high-resolution central vision and a wide overall receptive field. However, this distinctive sampling pattern causes visual signals in the retina to lack stationarity; the statistical properties differ between the center and periphery, with ganglion cell responses being highly correlated in the fovea but less correlated in the periphery. To mimic the foveated imaging with CIFAR-10, we adopt the retina sampling mechanism from Cheung et al. [14]. Like shown in Figure 3, each dot represents a retinal ganglion cell, which together form a non-stationary sampling lattice with irregular topology. Details on the implementation of foveation sampling is provided in Appendix A.3.

**Experiments.** We compare URLOST on both of the synthetic vision datasets as well as the original CIFAR-10 with popular unsupervised representation learning methods SimCLR [12] and MAE [34]. We also compared the ViT backbone used in URLOST to other backbones such as convolutional neural network (CNN). Moreover, since CNN works better with stationary signals, we further compared our methods to SimCLR with a graph neural network (GNN) backbone, which doesn't rely on the stationarity assumption of the data. For the GNN baseline, we use Vision GNN (ViG), which is state of the art graph neural network on images proposed in [32]. After the model is trained without supervision, we use linear probing to evaluate it. This is achieved by training a linear classifier on top of the pre-trained model with the given labels. The evaluations are reported in Table 1. SimCLR excels on CIFAR-10 but struggles with both synthetic datasets due to its inability to handle data without stationarity and topology. MAE gets close to SimCLR on CIFAR-10 with a $4 \times 4$ patch size. However, the patch size no longer makes sense when data has no topology. So we additionally tested MAE masking pixels instead of image patches. The performance of pixel level MAE is invariant to the removal of topology. The performance on *Permuted CIFAR-10* is the same as the performance on CIFAR-10, though poorly. But it still drops greatly to $48.5\%$ on the *Foveated CIFAR-10* when stationarity is also removed. In contrast, only URLOST is able to maintain consistently strong performances when there is no topology or stationarity, achieving $86.4\%$ on *Permuted CIFAR-10* and $85.4\%$ on *Foveated CIFAR-10* when the baselines completely fail.

## 3.2 V1 NEURAL RESPONSE TO NATURAL IMAGE STIMULUS

After testing URLOST's performance on synthetic biological vision data, we take a step further to challenge its generalizability with high-dimensional natural datasets. The first task is decoding neural response recording in the primary visual area (V1) of mice.

**V1 neural response dataset.** The dataset, published by [57], contains responses from over 10,000 V1 neurons captured via two-photon calcium imaging. These neurons responded to 2,800 unique images from ImageNet [19], with each image presented twice to assess the consistency of the neural response. In the decoding task, a prediction is considered accurate if the neural response to a given

Table 1: **Evaluation on computer vision and synthetic biological vision dataset.** ViT (Patch) stands for the Vision Transformer backbone with image patches as inputs. ViT (Pixel) means pixels are treated as input units. ViT (Clusters) means clusters are treated as inputs instead of patches. The number of clusters is set to 64 for both *Permuted CIFAR-10* and *Foveated CIFAR-10* dataset. Eval Acc stands for linear probing evaluation accuracy.

| Dataset | Method | Backbone | Eval Acc |
|---|---|---|---|
| CIFAR-10 | MAE | ViT (Patch) | 88.3 % |
| | MAE | ViT (Pixel) | 56.7 % |
| | SimCLR | ResNet-18 | **90.7 %** |
| | SimCLR | ViG | 53.8 % |
| Permuted CIFAR-10 | URLOST MAE | ViT (Cluster) | **86.4 %** |
| | MAE | ViT (Pixel) | 56.7 % |
| | SimCLR | ResNet-18 | 47.9 % |
| | SimCLR | ViG | 40.0 % |
| Foveated CIFAR-10 | URLOST MAE | ViT (Cluster) | **85.4 %** |
| | MAE | ViT (Pixel) | 48.5 % |
| | SimCLR | ResNet-18 | 38.0 % |
| | SimCLR | ViG | 42.8 % |

Table 2: **Evaluation on V1 response decoding and TCGA pan-cancer classification tasks.** "Raw" indicates preprocessed (standardized and normalized) raw signals. Best $\beta$ values are used for $\beta$-VAE. For URLOST MAE, cluster sizes are 200 (V1) and 32 (TCGA). We pick 15 seeds randomly to repeat the training and evaluation for each method. For the MAE baseline, we randomly group dimensions instead of using spectral clusterings. We report the 95% confidence interval for all methods.

| Method | V1 Response Decoding Acc | TCGA Classification Acc |
|---|---|---|
| Raw | 73.9% $\pm$ 0.00 % | 91.7 $\pm$ 0.24% |
| MAE | 70.6% $\pm$ 0.22 % | 90.6% $\pm$ 0.63% |
| $\beta$-VAE | 75.64% $\pm$ 0.11% | 94.15% $\pm$ 0.24% |
| URLOST MAE | **78.75% $\pm$ 0.18 %** | **94.90% $\pm$ 0.25 %** |

stimulus in the first presentation closely matches the response to the same stimulus in the second presentation within the representation space. This task presents greater challenges than the synthetic biological vision described in the prior section. For one, the data comes from real-world neural recordings rather than a curated dataset like CIFAR-10. For another, the geometric structure of the V1 area is substantially more intricate than that of the retina. To date, no precise mathematical model of the V1 neural response has been well established. The inherent topology and stationarity of the data still remain difficult to grasp [56; 55]. Nevertheless, evidence of retinotopy [25; 26] and findings from prior research [54; 13; 74] suggest that the neuron population code in V1 are tiling a low dimensional manifold. This insight led us to treat the population neuron response as high-dimensional data and explore whether URLOST can effectively learn its representation.

**Experiments.** Following the approach in Pachitariu et al. [57] we apply standardization and normalization to the neural firing rate. The processed signals are high-dimensional vectors, and they can be directly used for the decoding task, which serves as the "raw" signal baseline in Table 2. For representation learning methods, URLOST is evaluated along with MAE and $\beta$-VAE [35]. For MAE baseline, we obtain patches by randomly selecting different dimensions from the signal. Note that the baseline methods need to handle high-dimensional vector data without stationarity or topology. Since SimCLR and other constrastive learning model leverage these two properties to make positive pair, they are no longer applicable. We use $\beta$-VAE as a baseline instead. We first train the neural network with an unsupervised learning task, then use the latent state of the network as the representation for the neural responses in the decoding task. The results are presented in table 2. Our method surpasses the original neuron response and other methods, achieving the best performance.

| Masking Unit | Permuted Cifar10 | Foveated Cifar10 | TCGA Gene | V1 Response |
|---|---|---|---|---|
| Clusters (URLOST) | **86.4%** | **85.4%** | **94.9%** | **78.8%** |
| Random patch | 55.7% | 51.1% | 91.7% | 73.9% |
| Individual dimension | 56.7% | 48.5% | 88.3% | 64.8% |

Table 3: **Performance of different datasets using different masking units.** "Clusters (URLOST)" refers to the clusters formed using pairwise mutual information and spectral clustering. "Random patch" refers to patches formed by aggregating random dimensions. "Individual dimension" refers to using individual dimensions as masking units without any aggregation. The model is trained for the mask prediction task, and the probing accuracy is reported in the table.

## 3.3 GENE EXPRESSION DATA

In this subsection, we further evaluate URLOST on high-dimensional natural science data from a completely different domain, the gene expression data.

**Gene expression dataset.** The dataset comes from The Cancer Genome Atlas (TCGA) [79; 84], which is a project that catalogs the genetic mutations responsible for cancer using genome sequencing and bioinformatics. The project molecularly characterized over 20,000 primary cancers and matched normal samples spanning 33 cancer types. We focus on the pan-cancer classification task: diagnose and classify the type of cancer for a given patient based on his gene expression profile. The TCGA project collects the data of 11,000 patients and uses Micro-RNA (miRNA) as their gene expression profiles. Like the V1 response, no explicit topology and stationarity are known and each data point is a high-dimensional vector. Specifically, 1773 miRNA identifiers are used so that each data point is a 1773-dimensional vector. Types of cancer that each patient is diagnosed with serve as the classification labels.

**Experiments.** Similar to Section 3.2, URLOST is compared with the original signals, MAE, and $\beta$-VAE, which is the state-of-the-art unsupervised learning method on TCGA cancer classification [91; 92]. We also randomly partition the dataset do five-fold cross-validation and report the average performance in Table 2. Again, our method learns meaningful representation from the original signal. The learned representation benefited the classification task and achieved the best performance, demonstrating URLOST's ability to learn meaningful representation of data from diverse domains.

## 4 ABLATION STUDY

### 4.1 AGGREGATING SIMILAR DIMENSIONS FOR MASKING UNIT

Clustering is at the heart of the proposed method. Why should similar dimensions be aggregated to form a masking unit? Intuitively, we assume that similar dimensions are used to sample similar regions of the underlying signals. Instead of masking each dimension, making the entire cluster force the model to learn high-level structure from the signal, allowing the model to learn a rich representation. For example, for the CIFAR10 example, similar pixels tend to sample similar regions, which together form image patches. For V1 data, each neuron encodes a simple pattern like an oriented edge, but neurons together code patterns like shape and contours. We provide experiments for using different masking units, as shown in table 3, aggregating similar dimensions significantly improves the performance.

### 4.2 SELF-ORGANIZING LAYER VS SHARED PROJECTION LAYER

Conventional SSL models take a sequential input $x = [x^{(1)}, \cdots x^{(M)}]$ and embed them into latent vectors with a linear transformation:

$$z_0 = [Ex^{(1)}, \cdots Ex^{(M)}] \tag{3}$$

which is further processed by a neural network. The sequential inputs can be a list of language tokens [21; 63], pixel values [11], image patches [22], or overlapped image patches [12; 33; 90]. $E$ can be considered as a projection layer that is shared among all elements in the input sequence. The self-organizing layer $g(\cdot, w^{(i)})$ introduced in Section 2.3 can be considered as a non-shared projection layer. We conducted an ablation study comparing the two designs to demonstrate the effectiveness of the self-organizing layers both quantitatively and qualitatively. To facilitate the ablation, we further synthesized another dataset.

**Locally-permuted CIFAR-10**. To directly evaluate the performance of the non-shared projection approach, we designed an experiment involving intentionally misaligned clusters. In this experiment, we divide each image into patches and locally permute all the patches. The $i$-th image patch is denoted by $x^{(i)}$, and its permuted version, permutated by the permutation matrix $E^{(i)}$, is expressed as $E^{(i)}x^{(i)}$. We refer to this synthetic dataset as the *Locally-Permuted CIFAR-10*. Our hypothesis posits that models using shared projections, as defined in Equation 3, will struggle to adapt to random permutations, whereas self-organizing layers equipped with non-shared projections can autonomously adapt to each patch's permutation, resulting in robust performance. This hypothesis is evaluated quantitatively and through the visualization of the learned weights $w^{(i)}$.

**Permuted CIFAR-10**. Meanwhile, we also run the ablation study on the *Permuted CIFAR-10*. Unlike locally permuted CIFAR-10, a visualization check is not viable since the permutation is done globally. However, we can still quantitatively measure the performance of the task.

**Quantitative results**. Table 4 confirms our hypothesis, demonstrating a significant performance decline in models employing shared projections when exposed to permuted data. In contrast, the model with self-organizing layer maintains stable performance.

**Visual evidence**. Using linear layers to parameterize the self-organizing layers, i.e. let $g(x, W^{(i)}) = W^{(i)}x$, we expect that if the projection layer effectively aligns the input sequence, $E^{(i)T}W^{(i)}$ should exhibit visual similarities. That is, after applying the inverse permutation $E^{(i)T}$, the learned projection matrix $W^{(i)}$ at each location should appear consistent or similar. The proof of this statement is provided in Appendix A.5. The model trained on *Locally-Permuted CIFAR10* provides visual evidence supporting this claim. In Figure 4, the weights show similar patterns after reversing the permutations. These observations demonstrate that URLOST can also be used as an unsupervised learning method to recover topology and enforce stationary on the signal. Note that Figure 4 shows that the self-organizing layer learns to "undo" the permutation. Additionally, the self-organizing layer also does some extra regular transformation to each patch. It is likely that the self-organizing layer learns to encode position information as transformations of some simple group structures.

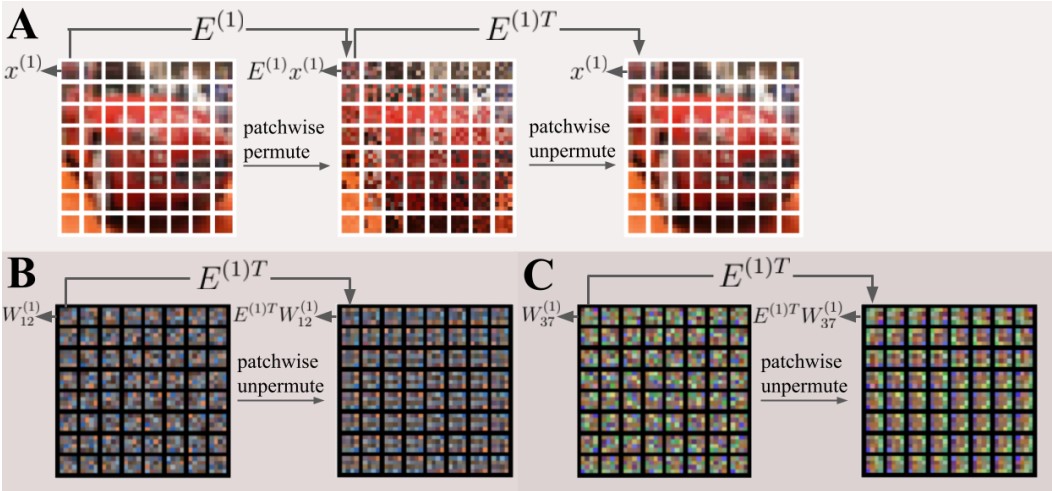

Figure 4: **Learnt weights of a self-organizing layer.** (A) Image is cropped into patches, where each patch $x^{(i)}$ first undergoes a different permutation $E^{(i)}$, then the inverse permutation $E^{(i)T}$. (B) The learned weight of the linear self-organizing layer. The 12th column of $W^{(i)}$ at all positions $i$ are reshaped into patches and visualized. When $W^{(i)}$ undergoes the inverse permutation $E^{(i)T}$, they show similar patterns. (C) Visualization of the 37th column of $W^{(i)}$. Similar to (B).

## 4.3 DENSITY ADJUSTED CLUSTERING VS UNIFORM DENSITY CLUSTERING

As explained in Section 2.2, the shape and size of each cluster depend on how the density function $p(i)$ is defined as:

$$p(i) = q(i)^{\alpha}n(i)^{-\beta} \tag{4}$$

where $n(i) = \sum_{j \in \text{Top}_k(A_{ji})} A_{ji}$, $A$ is the affinity matrix, $q(i)$ represent the eccentricity, the distance from $i$th kernel to the center of the sampling lattice. Setting $\alpha$ and $\beta$ nonzero, the density function is eccentricity-dependent. Setting both $\alpha$ and $\beta$ to zero will make $n(i)$ constant which recovers the uniform density spectral clustering. We vary the parameters $\alpha$ and $\beta$ to generate different sets of clusters for the foveated CIFAR-10 dataset and run URLOST using each of these sets of clusters. Results in Table 5 validate that the model performs better with density-adjusted clustering. The intuitive explanation is that by adjusting the values of $\alpha$ and $\beta$, we can make each cluster carry similar amounts of information (refer to Appendix A.4.). A balanced distribution of information across clusters enhances the model's ability to learn meaningful representations. Without this balance, masking a low-information cluster makes the prediction task trivial, while masking a high-information cluster will make the prediction task too difficult. In either scenario, the model's ability to learn effective representations is compromised.

Table 4: **Ablation study on self-organizing layer**. Linear probing accuracy with varying parameters, keeping others constant. For *Locally-Permutated CIFAR-10*, we use $4 \times 4$ patch size. For *Permutated CIFAR-10* and *Foveated CIFAR-10*, we set the number of clusters to 64 for the spectral clustering algorithm. We kept the hyperparameter of the backbone model the same as in table 1.

(a) Replacing the self-organizing layer with the shared projection layer entails a significant drop in performance.

(b) "SC" denotes spectral clustering with uniform density clustering and "DSC" denotes density adjusted spectral clustering. Using DSC to create clusters outperform model without DSC.

| Dataset | Projection | Eval Acc |
|---|---|---|
| Locally-Permuted CIFAR-10 | shared | 81.4 % |
| | non-shared | **87.6** % |
| Permuted CIFAR-10 | shared | 80.7 % |
| | non-shared | **86.4** % |

| Dataset | Cluster | Eval Acc |
|---|---|---|
| Foveated CIFAR-10 | SC | 82.7 % |
| | DSC | **85.4 %** |

## 5  ADDITIONAL RELATED WORKS

Several interconnected pursuits are linked to this work, and we will briefly address them here:

**Topology in biological visual signal.** 2-D topology of natural images is a strong prior that requires many bits to encode [20; 3]. Such 2-D topology is encoded in the natural image statistic [71; 37]. Optic and neural circuits in the retina result in a more irregular 2-D topology than the natural image, which can still be simulated [66; 59; 60; 58; 78; 41]. This information is further processed by the primary visual cortex. Evidence of retinotopy suggests the low-dimensional geometry of visual input from retina is encoded by the neuron in primary visual cortex [53; 27; 36; 26; 83; 61]. These study provide evidence that topology under retinal ganglion cell and V1 neurons can be recovered. The theory and computational model of how visual system code encodes such 2-D is well-studied in computational neuroscience. The self-organizing map (SOM) was proposed as the first computational model by Kohonen in 1982. The algorithm produces a low-dimensional representation of a higher-dimensional dataset while preserving the topological structure of the data [43]. SOM is also motivated to solve the "unscramble" pixels problem by "descrambling" by mapping pixels into a 2D index set by leveraging the mutual information between pixels at different locations. More detail is in Appendix A.12. [67] tackles the same problem with manifold learning.

**Evidence of self-organizing mechanism in the brain.** In neuroscience, many works use the self-organizing maps (SOM) as a computational model for V1 functional organization [23; 76; 1; 24; 52; 43]. In other words, this idea of self-organizing is a principle governing how the brain performs computations. Even though V1 functional organizations are present at birth, numerous studies indicate that the brain's self-organizing mechanisms continue after full development [30; 68; 40].

**Learning with signal on non-euclidean geometry.** In recent years, researchers from the machine learning community have made efforts to consider geometries and special structures beyond classic images, text, and feature vectors. This is the key motivation for geometric deep learning and graph neural networks (GNN). Many works generalizes common operator for processing Euclidean data like 2d convolution and attention [7; 49; 18; 28]. Due to the natural of graph neural network, they often only work on limited data regime and do not scale to large data [69; 10; 9]. However, recent development in [32] shows this direction is prominent. They successfully scales GNN to ImageNet.

We compared their proposed neural network architecture with the ViT backbone used in URLOST. Recent research also explores adapting the Transformer to domains beyond Euclidean spaces such as [16; 15; 29]. Ma et al. [45] treats an image as a set of points but relies on 2D coordinates. UR-LOST employs a single mutual information graph to define the topology of the high-dimensional signal. Gao et al. [29], on the other hand, is designed to handle graph data, where each data point corresponds to a distinct graph. It segments a graph into "subgraphs," processes them with a GNN, and then passes the output to a transformer. This approach is undeniably more flexible but requires all subgraphs to be globally aligned. Furthermore, the self-organizing layer in URLOST generalizes the "patch resizer" mechanism from FlexiViT used in Beyer et al. [4]. Finally, the lines of works follows the "perceiver" architecture is very related to URLOST [39; 38]. "Perceiver" also implicitly aggregates similar dimensions of high dimensional data and processes the similar dimension with an attention mechanism. [88] is a follow-up work that combines the Perceiver architecture and masked-prediction objective. However, the key difference is that the "perceiver" computes similarity between dimensions on a single example signal. For image data, this is essentially clustering pixels based on the pixel intensity of color in a signal image. On the other hand, URLOST computes similarity over statistics of the distribution of the signal. Additionally, "perceiver" implicitly aggregates similar dimensions, while URLOST explicitly aggregates similar dimensions and recovers the original topology of the signal.

**Self-supervised learning.** Self-supervised learning (SSL) has made substantial progress in recent years. Different SSL method is designed for each modality, for example: predicting the masked/next token in NLP[21; 63; 8], solving pre-text tasks, predicting masked patches, or building contrastive image pairs in computer vision [48; 34; 87; 12; 33; 90]. These SSL methods have demonstrated descent scalability with a vast amount of unlabeled data and have shown their power by achieving performance on par with or even surpassing supervised methods. They have also exhibited huge potential in cross-modal learning, such as the CLIP by Radford et al. [64].

## 6 Discussion, Limitations, and future direction

The success of most current state-of-the-art self-supervised representation learning methods relies on the assumption that the data has known stationarity and domain topology. In this work, we explore unsupervised representation learning under a more general assumption, where the stationarity and topology of the data are unknown to the machine learning model and its designers. We argue that this is a general and realistic assumption for high-dimensional data in modalities of natural science. We propose a novel unsupervised representation learning method that works under this assumption and demonstrates our method's effectiveness and generality on a synthetic biological vision dataset and two datasets from natural science that have diverse modalities. We also perform a step-by-step ablation study to show the effectiveness of the novel components in our model.

Unlike the self-organizing layer, the procedure for aggregating similar dimensions together is separate from the training of MAE. Given the effectiveness of the self-organizing layer, this design could be sub-optimal and could be improved. Learning the clusters end-to-end with the representation via back-propagation is worth investigating in the future. Additionally, the computational costs of computing pairwise mutual information and clustering both scales quadratically with the number of dimensions of the signal. Although each procedure only needs to be performed once, the computation could become too slow and infeasible for extremely high-dimensional data ($d > 10,000$) with our current implementation as shown in Appendix A.9. In the Appendix, we provide a benchmark of the runtime of our implementation. Nevertheless, we think this problem could be resolved by a GPU implementation for clustering and mutual information. Additionally, although natural science datasets such as TCGA and V1 calcium imaging are considered large datasets in their respective domains, they are still small compared to the datasets used in computer vision. We expect larger natural science datasets to emerge as measuring and imaging technology advances. Additionally, adapting URLOST to support contrastive learning objectives, such as SimCLR, presents another intriguing direction.

## 7 Acknowledgement

We would like to specially thank Bruno Olshausen for suggesting key prior work on computational models and biological evidence for recovering the 2D topology of natural images. We also appreciate the helpful suggestions from Surya Ganguli and Atsu Kotani.

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

# A APPENDIX

## A.1 MOTIVATION OF DENSITY ADJUSTED SPECTRAL CLUSTERING

Using the terminologies in functional analysis, the mutual information graph defined in section 2.2 corresponds to a compact Riemannian manifold $\mathcal{M}$ and the Laplacian matrix $L$ is a discrete analogous to the Laplace Beltrami operator $\mathcal{L}$ on $\mathcal{M}$. Minimizing the spectral embedding objective $tr(YLY^T)$ directly corresponds to the following optimization problem in function space:

$$\min_{||f||_{L^2(\mathcal{M})}} \int_{\mathcal{M}} ||\nabla f||^2 d\lambda \tag{5}$$

where $f(x) : \mathcal{M} \to [0, 1]$ is the normalized signal defined on $\mathcal{M}$. We particularly want to write out the continuous form of spectral embedding so we can adapt it to non-stationary signals. To do so, we assume the measure $\lambda$ is absolutely continuous with respect to standard measure $\mu$. By apply the Radon-Nikodym derivative to equation 5, we get:

$$\int_{\mathcal{M}} ||\nabla f||^2 d\lambda = \int_{\mathcal{M}} ||\nabla f||^2 \frac{d\lambda}{d\mu} d\mu$$

where the quantity $\frac{d\lambda}{d\mu}$ is called the Radon-Nikodym, which is some form of density function. Let $p(x) = \frac{d\lambda}{d\mu}$, we can rewrite the optimization problem as the following:

$$\min_{||f||_{L^2(\mathcal{M})}} \int_{\mathcal{M}} ||p(x)^{\frac{1}{2}} \nabla f(x)||^2 d\mu \tag{6}$$

The density function $p(x)$ on the manifold is analogous to the density adjustment matrix in equation 1. Standard approaches in equation 5 assume that nodes are uniformly distributed on the manifold, thereby treating $p(x)$ as a constant and excluding it from the optimization process. However, this assumption does not hold in our case involving non-stationary signals. Our work introduces a variable density function $p(x)$ for each signal, making it a pivotal component in building good representations for non-stationary signals. This component is referred to as *Density Adjusted Spectral Clustering*. Empirical evidence supporting this design is provided through visualization and ablation studies in the experimental section.

## A.2 SPECTRAL CLUSTERING ALGORITHM

Given a high dimensional dataset $S \in \mathbb{R}^{n \times m}$, Let $S_i$ be $i$th column of $S$, which represents the $i$th dimension of the signal. We create probability mass functions $P(S_i)$ and $P(S_j)$ and the joint distribution $P(S_i, S_j)$ for $S_i$ and $S_j$ using histogram. Let the number of bins be $K$. Then we measure the mutual information between $P(S_i)$ and $P(S_j)$ as:

$$I(S_i; S_j) = \sum_{l=1}^{K} \sum_{k=1}^{K} P(S_i, S_j)[l, k] \log_2 \left( \frac{P(S_i, S_j)[l, k]}{P(S_i)[l] P(S_j)[k]} \right)$$

Let $A_{ij} = I(S_i; S_j)$ be the affinity matrix, and let the density adjustment matrix be $P$ defined in 2.2. Correlation is used instead of mutual information when the dimension is really high since computing mutual information is expensive. We follow the steps from [51] to perform spectral clustering with a modification to adjust the density:

1. Define $D$ to be the diagonal matrix whose $(i,i)$-element is the sum of $A$'s $i$-th row. Construct the matrix $L = P^{\frac{1}{2}}D^{-\frac{1}{2}}AD^{-\frac{1}{2}}P^{\frac{1}{2}}$.

2. Find $x_1, x_2, \cdots, x_k$, the $k$ largest eigenvectors of $L$, and form the matrix $X = [x_1, x_2, \cdots, x_k] \in \mathbb{R}^{n \times k}$ by stacking the eigenvectors in columns.

3. Form the matrix Y from X by renormalizing each of $X$'s rows to have unit norms. (i.e. $Y_{ij} = X_{ij}/(\sum_i X_{ij}^2)^{\frac{1}{2}}$)

4. Treating each row of $Y$ as a point in $\mathbb{R}^k$, cluster them into $k$ clusters via K-means or other algorithms.

Some other interpretation of spectral embedding allows one to design a specific clustering algorithm in step 4. For example, [72] interprets the eigenvector problem in 6 as a relaxed continuous version of K-way normalized cuts problem, where they only allow $X$ to be binary, i.e. $X \in \{0,1\}^{N \times K}$. This is an NP-hard problem. Allowing $X$ to take on real value relaxed this problem but created a degeneracy solution. Given a solution $X^*$ and $Z = D^{-\frac{1}{2}}X^*$, for any orthonormal matrix $R$, $RZ$ is another solution to the optimization problem 6. Thus, [72] designed an algorithm to find the optimal orthonormal matrix $R$ that converts $X^*$ to discrete value in $\{0,1\}^{N \times K}$. From our experiment, [72] is more consistent than K-means and other clustering algorithms, so we stick to using it for our model.

### A.3 DATA SYNTHESIZE PROCESS

We followed the retina sampling approach described in [14] to achieve foveated imaging. Specifically, each retinal ganglion cell is represented using a Gaussian kernel. The kernel is parameterized by its center, denoted as $\vec{x}_i$, and its scalar variance, $\sigma_i'^2$, i.e. $\mathcal{N}(\vec{x}_i, \sigma_i'^2\mathbf{I})$, which is illustrated in Figure 5.A. The response of each cell, denoted as $G[i]$, is computed by the dot product between the pixel value and the corresponding discrete Gaussian kernel. This can be formulated as:

$$G[i] = \sum_n^N \sum_m^W K(\vec{x}_i, \sigma_i')[n, m]I[n, m]$$

where $N$ and $W$ are dimensions of the image, and $I$ represents the image pixels.

For foveated CIFAR-10, since the image is very low resolution, we first upsample it 3 times from $32 \times 32$ to $96 \times 96$, then use in total of 1038 Gaussian kernels to sample from the upsampled image. The location of each kernel is illustrated in Figure 5.B. The radius of the kernel scales proportionally to the eccentricity. Here, we use the distance from the kernel to the center to represent eccentricity. The relationship between the radius of the kernel and eccentricity is shown in Figure 5.C. As mentioned in the main paper, in the natural retina, retinal ganglion cell density decreases linearly with eccentricity, which makes the fovea much denser than the peripheral, unlike the simulated lattice we created. The size of the kernel should scale linearly with respect to eccentricity as well. However, for the low-resolution CIFAR-10 dataset, we reduce the simulated fovea's density to prevent redundant sampling. In this case, we pick the exponential scale for the relationship between the size of the kernel and eccentricity so the kernel visually covers the whole visual field. We also implemented a convolution version of the Gaussian sampling kernel to speed up data loading.

### A.4 DENSITY ADJUSTED SPECTRAL CLUSTERING ON FOVEATED CIFAR10 DATASET

We provide further intuition and visualization on why density-adjusted spectral clustering allows the model to learn a better representation of the foveated CIFAR-10 dataset.

As shown in Figure 5, the kernel at the center is much smaller in size than the kernel in the peripheral. This makes the kernel at the center more accurate but smaller, which means it summarizes less information. Spectral clustering with constant density will make each cluster have a similar number

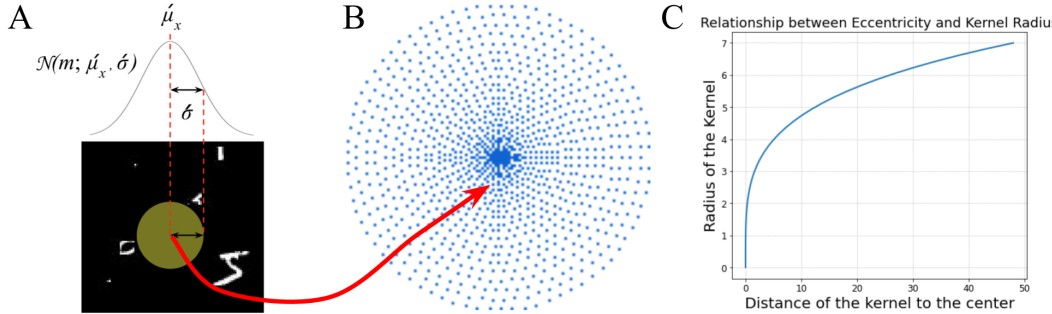

Figure 5: **Foveated retinal sampling** (A) Illustration of a Guassian kernel shown in [14]. Diagram of single kernel filter parameterized by a mean $\mu'$ and variance $\sigma'$. (B) the location of each Gaussian kernel is summarized as a point with 2D coordinate $\mu'$. In total, the locations of 1038 Gaussian kernels are plotted. (C) The relationship between eccentricity (distance of the kernel to the center) and radius of the kernel is shown.

Table 5: **Evaluation on foveated CIFAR-10 with varying hyperparameter for density function.** For each set of values of $\alpha$ and $\beta$, we perform density-adjusted spectral clustering and run URLOST with the corresponding cluster. The evaluation of each trial is provided in the table.

|  | beta = 0 | beta = 2 |
|---|---|---|
| alpha = 0 | 82.74 % | 84.24 % |
| alpha = 0.5 | 84.52 % | 85.43 % |
| alpha = 1.0 | 83.83 % | 81.62 % |

of elements in them. Since the kernel in the center is smaller, the cluster in the center will be visually smaller, than the cluster in the peripheral. The effect is shown in Figure 6. Moreover, since we're upsampling an already low-resolution image (CIFAR-10 image), even though the kernel at the center is more accurate, we're not getting more information. There, to make sure each cluster has similar information, the clusters in the center need to have more elements than the clusters in the peripheral. In order to make the clusters at the center have more elements, we need to weight the clusters in the center more with the density function. Since the sampling kernels at the center have small eccentricity and are more correlated to their neighbor, increasing $\alpha$ and $\beta$ will make sampling kernels at the center have higher density, which makes the cluster at the center larger. This is why URLOST with density-adjusted spectral clustering performs better than URLOST with constant density spectral clustering, which is shown in Table 5. Meanwhile, setting $\alpha$ and $\beta$ too large will also hurt the model's performance because it creates clusters that are too unbalanced.

## A.5 SELF-ORGANIZING LAYER LEARNS INVERSE PERMUTATION

For *locally-permuted CIFAR-10*, we divide each image into patches and locally permute all the patches. The $i$-th image patch is denoted by $x^{(i)}$, and its permuted version, permuted by the permutation matrix $E^{(i)}$, is expressed as $E^{(i)}x^{(i)}$. We use linear layers to parameterize the self-organizing layers. Let $g(x, W^{(i)}) = W^{(i)}x$ denotes the $i$th element of the self-organizing layer. We're providing the proof for the statement related to the visual evidence shown in Section 4.2

Statement: If the self-organizing layer effectively aligns the input sequence, then $E^{(i)T}w^{(i)}$ should exhibit visual similarities.

Proof: we first need to formally define what it means for the self-organizing layer to effectively align the input sequence. Let $\mathbf{e}_k$ denote the $k$th natural basis (one-hot vector at position $k$), which represents the pixel basis at location $k$. Permutation matrix $E^{(i)}$ will send $k$th pixel to some location accordingly. Mathematically, if the projection layer effectively aligns the input sequence, it means $g(E^{(j)}e_k, W^{(j)}) = g(E^{(i)}e_k, W^{(i)})$ for all $i, j, k$. We can further expand this property to get the

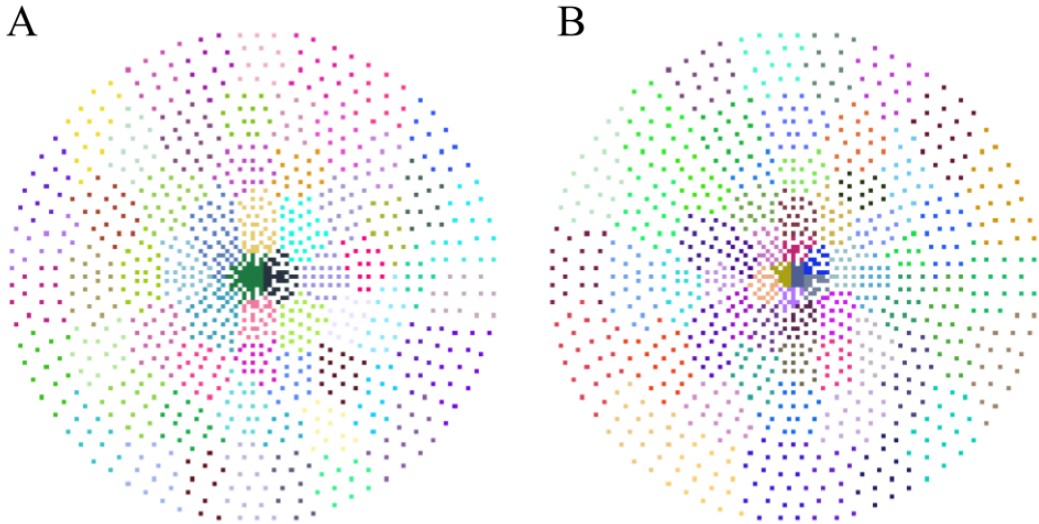

Figure 6: **Effect of density adjusted clustering.** Eccentricity-based sampling lattice. The center of the sampling lattice has more pixels which means higher resolution compared to the peripheral. (A) Result of density-adjusted spectral clustering ($\alpha = 0.5, \beta = 2$). Clusters in the center have more elements than clusters in the peripheral. But clusters look more visually similar in size than B. (B) Result of uniform density spectral clustering ($\alpha = 0, \beta = 0$). Each cluster has a similar number of elements in them but the clusters in the center are much smaller than the clusters in the periphery.

following two equations:

$$g(E^{(i)}e_k, W^{(i)}) = W^{(i)}E^{(i)}e_k$$
$$g(E^{(j)}e_k, W^{(j)}) = W^{(j)}E^{(j)}e_k$$

for all $i, j, k$. Since the above equation holds for all $e_k$, by linearity and the property of permutation matrix, we have:

$$W^{(i)}E^{(i)} = W^{(j)}E^{(j)}$$
$$E^{(i)T}W^{(i)} = E^{(j)T}W^{(j)}$$

This implies $E^{(i)T}w^{(i)}$ should exhibit visual similarities for all $i$.

### A.6    TRAINING AND EVALUATION DETAILS

$\beta$-**VAE**. $\beta$-VAE was trained for 1000 epochs and 300 epochs on the V1 neura l and TCGA gene expression respectively. We use the Adam optimizer with a learning rate of 0.001 and a cosine annealing learning rate scheduler. The encoder is composed of a 2-layer MLP with batch normalization and LeakyReLU activation. We use hidden dimensions 2048 and 1024 for V1 and Gene datasets respectively. Then, two linear layers are applied to get the mean and standard for reparameterization. The decoder also has a 2-layer MLP, symmetric to the encoder but using standard ReLU activation and no batch normalization. We tried out different hyperparameters and empirically found that this setting gave the best performance.

**MAE**. MAE follows the official implementation from the original paper. For CIFAR10, we ran our model for 10,000 epochs. We use Adam optimizer with a learning rate of 0.00015 and a cosine annealing. To fit in our tasks, we use 8 8-layer encoders and 4-layer decoders with hidden dimension 192. The ViT backbone can take different patch sizes and we indicated them accordingly in Table 1. ViT(Pixel) means treating each pixel as a patch, so essentially the patch size is 1. This is also used for the real-world high-dimensional dataset since no concept of patch is defined in the signal space.

For V1 neural recording and TCGA gene expression task, we use 4 layers encoder and 2 layers. We use hidden dimension 1380 for 1000 epochs and hidden dimension 384 with 3000 epochs for V1 neural recording and TCGA dataset task. The hidden dimension and the number of epochs we used for MAE are greater than $\beta$-VAE. However, when we use the same parameters on $\beta$-VAE, we did not seem to find a performance gain. Training Transformers usually requires a large number of data. For example, the original transformer on vision is pretrained over 14M images.

**URLOST MAE** Many hyperparameters are associated with the spectral clustering algorithms. The performance of the model depends on the cluster, which thus depends on these hyperparameters. In general, URLOST is not very sensitive to these hyperparameters. Density parameters (how nodes are normalized) and the number of clusters affects the shape of the clusters, thus, affects the performance of the model to some degree.

Intuition on hyperparameter selection: Both these hyperparameters are related to the size of each cluster, which relates to how difficult the task is. If a cluster is too big, then we need to predict a big missing area on the graph, making the unsupervised learning task too difficult. On the other hand, if the cluster is too small, for example, if the cluster shrinks down to one pixel for an image, then the prediction tasks become too easy. The solution is simply doing a low-pass filter on the graph to fill in the missing pixel. So the model will not learn high-level or semantic representation. Both the number of clusters and density factors are related to the cluster size. Increasing the number of clusters could make each cluster smaller. The density factor essentially defines how much we want to normalize each node on the graph. For an image dataset, clusters should contain semantic sub-structures of the image, like body parts or parts of objects. We use 64 clusters, which results in clusters of size roughly 4x4. In other words, the ratio between the size of clusters (part) and the number of dimensions (whole) is roughly 1:64. For other datasets (gene and V1), we roughly keep this ratio and perform a grid search over these hyperparameters. Similarly, for the density factors, we also perform a grid search centered at the density factor equal to 1. Empirically, URLOST is not very sensitive too these hyperparameters as shown in Figure 7.

We also provide the exact hyperparameters we used for the experiments in this paper: the parameter of URLOST MAE is the same as MAE except for the specific hyper-parameter in the method section. For CIFAR10, we use $K = 20$, $\alpha = 0.5$ and $\beta = 2$. We set the number of clusters to be 64. For V1 neural recording, we use $K = 15$, $\alpha = 0$ and $\beta = 1$. We set the number of clusters to 200. For TCGA dataset, we use $K = 10$, $\alpha = 0$ and $\beta = 1$. We set the number of clusters to 32.

### A.7 Effect of spectral clustering hyperparameters on performance

We perform an experiment doing a grid search over each hyperparameter in spectral clustering that affects the performance of URLOST. The grid search experiment is performed over the TCGA gene classification dataset. As shown in Figure 7, the model is not sensitive for most of the hyperparameters. For a large range of hyperparameters, the model performance stays the same.

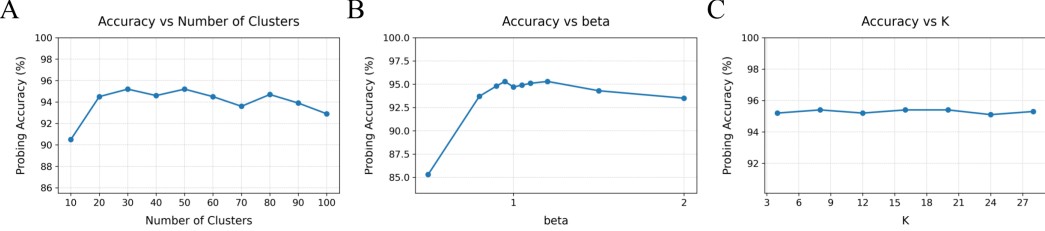

Figure 7: **Effect of spectral clustering hyperparameters on performance** (A) Effect of number of cluster on performance. (B) Effect of normalization parameter for spectral clustering ($\beta$ in Eq. 4) on performance. (C) Effect of numbers of neighbors from $K$ nearest neighbors on performance.

### A.8 Topology and Stationarity

We provide a formal definition of topology and stationarity used in the context of our work here:

**On topology of the domain of signals.** A signal is a function $X(s)$ defined on some domain $S$, where $s \in S$. Let's focus on $S$. While the definition of topology is a little abstract and relies on open sets, the intuition is that it defines a generalized sense of nearness, or whether two points are close to each other or far apart. A common way to give such a space the property of being near or far away is with an explicit distance function, or a metric. The distance function generates a class of objects—open sets. These spaces (with an associated metric) are called metric spaces. For example, the topology of image space is induced by the L2 norm as the metric function. However, not every space will have a natural metric on it. This is why we need to use the term "topology".

**Topological spaces are generalizations of metric spaces.** The definition of topology is any collection of subsets of $S$ that satisfies certain axioms: close under arbitrary union and finite intersection. In the practical setting, the domain $S$ is always discretized as a set of indices, and we can assume that we always deal with a signal defined on a graph [70]. Graphs can be embedded in $\mathbb{R}^3$. For a finite graph, the graph topology is just the subspace topology inherited from the usual topology in $\mathbb{R}^3$. Under this specific definition, the word "graph," "adjacency matrix," and "topology" can be used interchangeably. In this work, we assume we are not provided the topology. Thus, using pairwise mutual information as the adjacency matrix is a way to model the topology. Correlation is another way to model the topology, where we use only second-order pairwise statistics. This is slightly different from the reviewer's understanding "Topology appears to correspond to "correlation" between dimensions." More generally, graphs are topological spaces that are the simplicial 1-complexes or the 1-dimensional CW complexes.

**Example with stationarity but no topology:** Permuted Cifar10 is an example we had in the manuscript for this criterion (no topology but has stationarity). No topology refers to the fact that $n(i)$ is not defined after random permutation. "Has stationarity" means that the signal is stationary when the topology is recovered. Specifically, we use the mutual information graph to define neighbors $n(i)$. Under this particular definition of $n(i)$, there exists a meaningful invariant statistic $S_f(i)$ for the signal. Thus, the signal is stationary. This is because the topology recovered is very similar to the original 2d topology of images before permutation, and the sign before permutation is stationary. However, under an incorrect definition of $n(i)$, the signal is no longer stationary.

**Example with no topology but no stationarity:** One example would be foveated images simulated using convolutional kernels. One could make a different region of an image have a different resolution while preserving the 2d grid structure of the image by using a Laplace pyramid. As a result, the image is still represented as a 2d matrix, and the definition of $n(i)$ is the same as regular images. However, many invariant statistics before the foveation process are destroyed due to the imbalance resolution at different regions of the image.

**On stationarity.** In probability theory, a random process is a family of random variables. While the index was originally and frequently interpreted as time, it was generalized to random fields, where the index can be an arbitrary domain. That is, by modern definitions, a random field is a generalization of a stochastic process where the underlying parameter need no longer be real or integer valued "time" but can instead take values that are multidimensional vectors in $\mathbb{R}^n$, points on some manifolds and Lie Groups, or graphs [81; 50; 77; 42; 65; 5]. Let's consider a real-valued random field $X(s)$ defined on a homogeneous space $S = s$ of points $s$ equipped with a transitive transformation group $G = g$ of mappings of $S$ into itself, and having the property that the values of the statistical characteristics of this field do not change when elements of $G$ are applied to their arguments. In convention, we call a random field $X(s)$ a strict-sense stationary (or strict-sense homogeneous) random field if for all $n = 1, 2, \cdots$ and $g \in G$, the finite-dimensional probability distribution of its values at any $n$ points $s_1, \cdots, s_n$ coincides with that of its values at $gs_1, \cdots, gs_n$. We call $X(s)$ field wide-sense stationary (or wide-sense homogeneous) random field if $E|X(s)|^2 <$ inf and $EX(s) = EX(gs)$, $EX(s)X(s_1) = EX(gs)EX(gs_1)$ for all $s$, $s_1 \in S$ and $g \in G$. Or wide-sense stationarity means the first and second-order statistics do not change w.r.t. the group action. Further, we can intuitively generalize the wide-sense stationarity to higher-order statistics if we need to define any nth-order stationarity between strict-sense and wide-sense.

In intuitively, stationarity just means the statistics does not change w.r.t. transformations. Let's take the image as an example, it is a signal $X(s)$ defined on $S = \mathbb{Z}^2$. And let's define the group action as translation. If we use a convolution neural network and learn a set of $L \times L$ filters or use a VIT to learn a set of shared features on $L \times L$ patches. A hidden assumption is that the statistics of our image does not change for $L \times L$ region, or in order words the joint probability $P(X(1,1), \cdots, X(L,L))$ $= P(X(1+a, 1+b), \cdots, X(L+a), \cdots, X(L+b))$, where $(a, b) \in$ translation group $T$. While

Table 6: **Ratio between Parameters of Self-Organizing Layers and Parameters of the Entire Model**. We report the model used for each dataset.

| Dataset | Permuted Cifar10 | TCGA | V1 |
|---------|------------------|-------|------|
| Ratio | 6.7% | 15.6% | 5.9% |

we have initially cited [8], it might be a good idea to provide a more thorough explanation in the appendix, hence, we have included another five references to provide a more comprehensive point of view.

## A.9 COMPUTATIONAL COST

**Computation cost**. The computational cost and scalability of URLOST are nearly the same as MAE. The computational cost of MAE is analyzed in the original MAE paper. Its efficient design of MAE speeds up the training and evaluation time around 4x compared to a standard Transformer-based auto-encoder. The additional elements in URLOST are simply clustering and self-organizing layers. This effect is shown in figure 7, despite the encoder used in the experiment being 3 times as the decoder, the runtime of the encoder is still faster than the decoder. Moreover, the operation of clustering indexing and self organizing layer is very small relative to the overall runtime of the model. The mutual information and the clustering results are precomputed based on the dataset and thus do not introduce more cost for the model training. During training and inference, clustering is just a one-step indexing process that does not create significant overhead in additional to MAE. For all experiments performed in this paper, since clustering and mutual information are computed and saved for all datasets at once, their computational cost is negligible with respect to the training time of MAE. However, The runtime of these two procedure could be a problem for extremely high dimensional data.

With the current implementation, we perform a scalability benchmark as suggested by the reviewer. We resize an image dataset to different resolutions. At each resolution, we compute mutual information and clustering for a different number of pixels (dimensions), and we provide a plot of running time with respect to the number of pixels in Figure 8. Both computational cost of computing pairwise mutual information and clustering both scale quadratically with the number of dimensions of the signal. It takes aroudn 130 minutes to calculate mutual information for data with 10,000 dimensions. Clustering takes much shorter time (less than a minute).

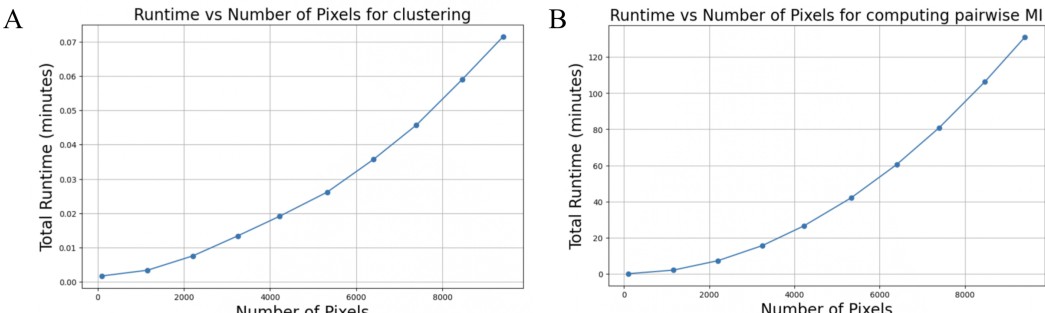

Figure 8: **Runtime for clustering and computation mutual information.** A. Benchmark of runtime for clustering for signal with different dimensions. B. Benchmark of runtime for computing pairwise mutual information with different dimensions. Both run time scale quadratically as the dimension of the signal.

**Amount of parameters**. The self-organizing layer introduces additional parameters compared to the original MAE model. We believe additional parameters are a reasonable trade-off for not knowing topology and stationarity, which are important inductive biases in the data. However, it doesn't increase the parameters drastically, we provide the ratio of the additional parameters in table 6.

Table 7: **Runtime of each module during the forward operation**. We report the runtime of each module during inference for a single example. We use the model trained on permuted CIFAR10 dataset. The experiment is performed using a single RTX 2080TI and is averaged over 500 trials.

| Module | Cluster Indexing | Self Organizing Layer | Encoder | Decoder |
|---|---|---|---|---|
| Runtime | 0.27 ms | 1.11 ms | 17.5 ms | 20.4 ms |

## A.10 SCALABILITY

Scalability is not the primary focus of this work. In general, unsupervised representation learning benefits from scaling to a larger amount of data, and the MAE model, which our method is built upon, demonstrates a level of scaling [34]. We believe URLOST inherits this potential for the fact that it shows stronger advantages on more challenging tasks, as seen with CIFAR-10 compared to simpler benchmarks. This suggests that scaling to larger and more complex datasets is a promising direction. Our study explores the novel settings of unsupervised representation learning without knowing the data topology and stationarity, demonstrating the effectiveness of our method across diverse data modalities with significant differences. This generalization is our key and positive contribution, highlighting the potential of our approach to inspire future work in this area.

## A.11 NOISE ROBUSTNESS OF MUTUAL INFORMATION RECOVERED TOPOLOGY

We tested the robustness of the topology recovery algorithm used in URLOST. Specifically, we add gaussian noise to CIFAR10 to generate data with a low signal-to-noise ratio.

We found out that while decreasing signal-to-noise ratio will lower the overall mutual information between pairs of pixels, the relationship between mutual information between pixels and image topology (distance between two pixels) does not change. We perform the following experiments: we add iid gaussian noise to each pixel with a fixed noise level, then we measure the mutual information between pairs of pixels as $MI((i,j),(k,l))$. For visualization, we aggregate all pairs by the distance between them and average their mutual information, i.e. let $A(a) = (i,j),(k,l)s.t.\sqrt{(i-k)^2+(j-l)^2}$. We have $MI(a) = \frac{1}{|A(a)|}\sum_{i,j,k,l\in A(a)} MI((i,j),(k,l))$. $MI(a)$ denotes the mutual information between pixels of different distances.

As shown in Figure 9 below, although the overall mutual information decreases a lot as the noise level increases, the order of MI(a) is still the same. In other words, the mutual information is still the biggest when the two pixels are nearby and decays when the mutual information decreases. This implies the topology recovered by pairwise mutual information is robust even when the signal has a lower signal-to-noise ratio. It only breaks when you add too much noise ($\sigma = 0.9$). For this amount of noise, the image is not distinguishable.

## A.12 SELF-ORGANIZING MAP (SOM)

Kohonen in 1982 presents a fascinating thought experiment [43]: "Suppose you woke up one day to find someone rewired your optic nerve (or you have been implanted with a prosthetic retina). The signals from the retina to the brain are intact, but the wires are all mixed up, projecting to the wrong places. Can the brain learn to "descramble" the image?" Kohonen proposes the "self-organizing map" (SOM) algorithm to address this problem. SOM is an unsupervised machine learning technique used to produce a low-dimensional (typically two-dimensional) representation of a higher-dimensional dataset while preserving the topological structure of the data. In the task of "descrambling" pixels, SOM maps each pixel into a 2D index set by leveraging the mutual information between pixels at different locations. The parameters of SOM include a set of neurons, where each neuron has a location index. Let $\mathbf{W}_v$ denote the weight vector of the $v$th neuron and $\mathbf{D}$ the input vector. $\theta(u,v,s)$ is the interaction function that determines the influence between the $u$th and $v$th neurons based on their distance. The update equation for SOM is given by:

$$\mathbf{W}_v(s+1) = \mathbf{W}_v(s) + \theta(u,v,s) \cdot \alpha(s) \cdot (\mathbf{D}(t) - \mathbf{W}_v(s))$$

In the task of "descrambling" pixels, the input consists of pixel values. The indices will be two-dimensional, such as $s = (i,j)$, and $\mathbf{W}_v(s)$ will learn to represent a set of "descrambled" pixels,

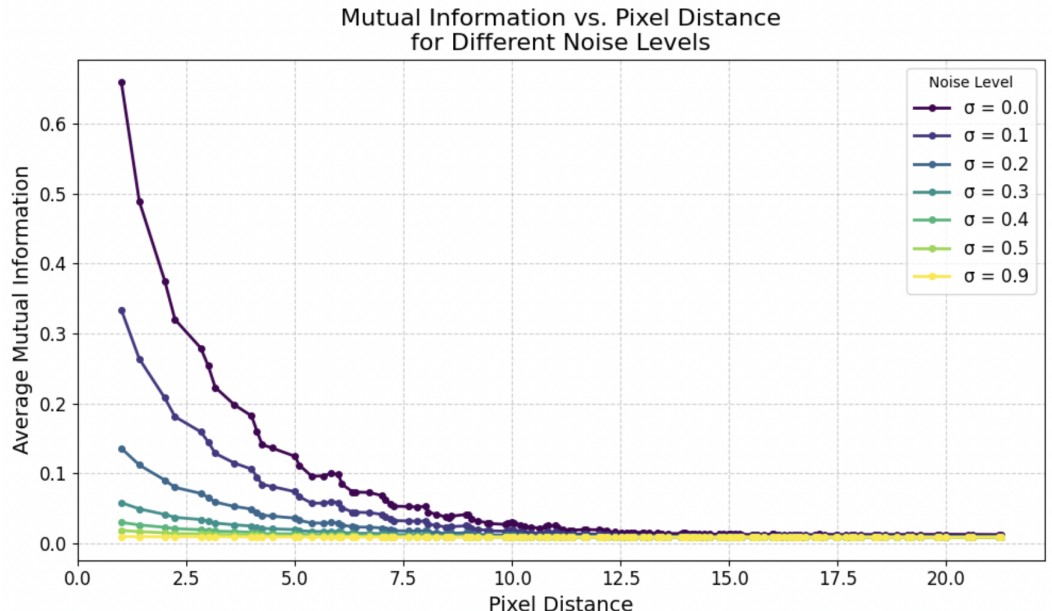

Figure 9: **The effect of additive noise on pairwise mutual information.** We plot averaged mutual information between pairs of pixels. We aggregate all pairs with the same distance. Each curve represents a specific noise level. Although the curve becomes lower as noise increases, the decaying features of the curve remain the same.

where $s$ represents the correct indices of the pixels. In other words, the index set defines the correct topology of the image as a 2D grid, and SOM maps the pixels onto this 2D topology by leveraging the mutual information between pixels at different locations. Other methods such as [67] use manifold learning to uncovered topology 2-d topology from natural images. However, when the intrinsic dimension of the manifold is larger than 2d, it is difficult to integrate the "uncovered" topology with state-of-the-art self-supervised learning algorithms.

## A.13    VISUALIZING THE WEIGHT OF SELF-ORGANIZING

As explained in the previous section (Appendix A.5) and visualized in Figure 10, we can visualize the weights of the learned self-organizing layer when trained on the locally-permuted CIFAR-10 dataset. If we apply the corresponding inverse permutation $E^{(i)T}$ to its learned filter $W^{(i)}$ at position $i$, the pattern should show similarity across all position $i$. This is because the model is trying to align all the input clusters. We have shown this is the case when the model converges to a good representation. On the other hand, what if we visualize the weight $E^{(i)T}W^{(i)}$ as training goes on? If the model learns to align the clusters as it is trained for the mask prediction task, $E^{(i)T}W^{(i)}$ should become more and more consistent as training goes on. We show this visualization in Figure 10, which confirms our hypothesis. As training goes on, the pattern $E^{(i)T}W^{(i)}$ becomes more and more visually similar, which implies the model learns to gradually learn to align the input clusters.

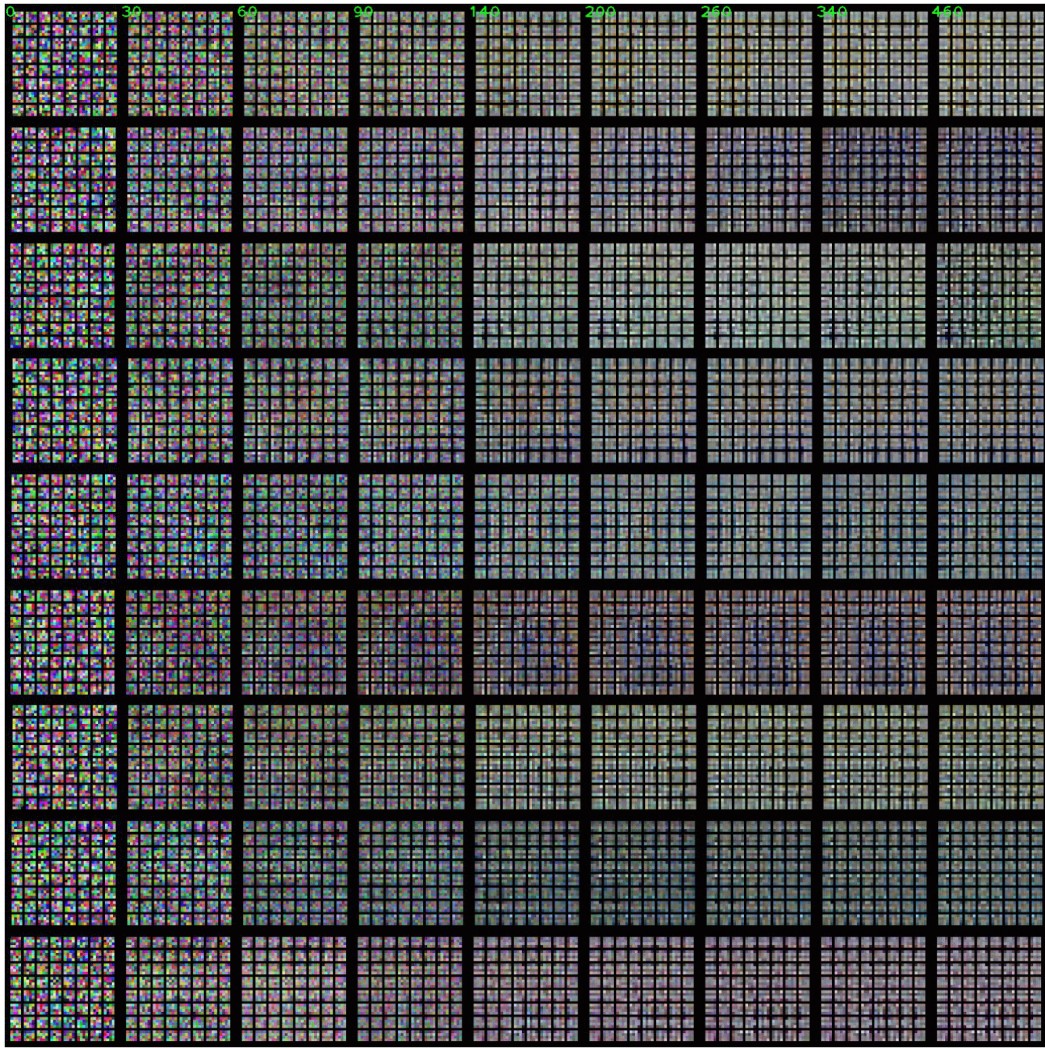

Figure 10: **Visualize the weight of the self-organizing layer after applying inverse permutation.** A snapshot of $E^{(i)T}W^{(i)}$ is shown at different training epoch. The number of epochs is shown on the top row. Each figure shows one column of the weight of the self-organizing layer, at different positions, i.e. $W_{:,k}^{(1)}$, where $k$ is the column number and $i$ is the position index. In total, 9 columns are shown.

