# OpenReview forum: "URLOST: Unsupervised Representation Learning without Stationarity or Topology"
_ICLR.cc/2025/Conference — ICLR 2025 Poster_

### Official Review · Reviewer_t33q · 2024-10-17

**Soundness:** 2
**Presentation:** 2
**Contribution:** 2
**Rating:** 8
**Confidence:** 3

**Summary:**

The paper suggests a novel way unsupervised representation learning pipeline, which consists of three stages: clustering, self-organising layers and masked autoencoder training. The authors try to tackle the assumptions of stationarity and topological ordering of the data for self-supervised learning.

**Strengths:**

* great motivation: important problem of topological ordering and stationarity assumptions is raised. Biological inspiration is also interesting.
* novel ideas, especially, when it comes to clustering and self-organized layers instead of patching. This potentially allows to aggregate features on different scales, which could be highly beneficial in many cases (for example, action tracking could be composed of motions tracking)
* making the code available is great for reproducibility

**Weaknesses:**

* While the authors criticise graph neural networks for not being scalable and the paper is proposed as a new self-supervised approach, the scalability aspect is not addressed, both from the perspective of time (and spectral clustering can become slow) or performance . The datasets used in the paper are fairly small, while even for transcriptomics or neuroscience there are quite big datasets publicly available now.
* Topology is mentioned a lot, however, I do not fully understand, where exactly authors show that their method is not relying on topology. I would suggest authors to take a point cloud dataset (for example, 3D Point Cloud Classification on ModelNet40 ) and perform a comparison on it. The transcriptomics vectors are still ordered (like each position corresponds to a specific gene), which could be also thought as topological organisation (like a flattened picture).
* The experiments are missing the error bars or confidence intervals, which makes it a bit hard to understand how robust is the method and how significant are the improvements. I would inspire the authors to add the error bars for all of the experiments
*  while the paper motivation is highly biologically grounded, the actual image dataset has very low signal-to-noise ratio, eg  a lot of natural scenes do not so nicely isolate a single object in the center of the image such as CIFAR. I would inspire the authors to use a more noisy multi-object classification or segmentation dataset
* while avoiding stationarity assumption is one of the main paper motivations, I felt like a textbook case of non-stationary data, eg time series, is missing. I would suggest to include some non-stationary timeseries classification example, for example, for epilepsy, to prove illustrate this point in a much more convincing manner.
* I find the biological motivation sometimes more confusing than helping, for instance, lines 471-476, mention self-organising maps, without giving a brief introduction on what they are.

**Questions:**

* if I permute all images in the same way, why would it ruin the topology? As far as I understand it, I will just transform it but the ordering is still preserved

---

> ### Author Response · Authors · 2024-11-23
> **Author Response to Reviewer t33q (1/3)**
>
> Dear Reviewer t33q,
>
> Thank you for your time and your valuable reviews and questions. We have responses to each question individually in the following and we hope we have addressed your concerns and provided a clearer understanding of our work. We look forward to the continued discussion should there be any additional questions and concerns.
>
> __Q1__: While the authors criticize graph neural networks for not being scalable the paper is proposed as a new self-supervised approach.
>
> __A1__: To clarify, we did not criticize graph neural networks for not being scalable. On the contrary, URLOST is not contradict with GNN in fact, we cited many graph neural networks papers and even included one as a baseline model (ViG). We want to emphasize that our model is essentially solving a different problem than graph neural networks. Graph neural networks assume each data point comes in as a graph. Nevertheless, we assume that the topology of the underlying domain of the data is a graph, which can be recovered by pairwise mutual information. In other words, we assume all the data points use the same graph except the signal varies as a function of the graph. Although GNN has a more general formulation, URLOST has a much simpler design that allows it to utilize powerful neural architecture and unsupervised learning tasks: Transformer and masked prediction. Our goal is never to downplay the importance of GNN, but rather to provide a simple tool for scientists to do learning on high dimensional data without labels.
>
> __Q2__: The scalability aspect is not addressed, both from the perspective of time (and spectral clustering can become slow) or performance.
>
> __A2__: We provide an additional section in the appendix A.10 to discuss the computational cost and scalability. In short, the computational cost and scalability of URLOST is nearly the same as MAE.
>
> __Q3__: The datasets used in the paper are fairly small, while even for transcriptomics or neuroscience there are quite big datasets publicly available now.
>
> __A3__: We use the Foveated Cifar, TCGA, and V1 recording datasets because they are all good representations of the problem we are addressing. They are all very high dimensional data, without explicit topology or stationarity, which is challenging for existing SSL methods while URLOST works effectively. The amount of data should not downweight this contribution. Additionally, they also come from completely different domains in natural science. So it demonstrates URLOST’s generalization capability. Another important factor is that they both have some labels so that we can use them for evaluation after the unsupervised pre-training phase. As a defense, we also tried to find more transcriptomics or neuroscience data. The problem is that patient data is expensive and rare. The TCGA is considered a fairly large dataset and is used in many pieces of literature. Some large neuroscience datasets are lower dimensional or contain explicit topology. For example, large amounts of neuroscience data are fMRI data, which is typically in the format of voxel. Since fMRI has an inherent topological structure, it’s not necessary to use URLOST as opposed to standard MAE. The ideal dataset from neuroscience would be a large populational neural response recording. This type of data only starts to emerge in recent years with accurate imaging techniques. The dataset we used is relatively recent. However, we expect high dimensional data would be more prevalent in neuroscience as measuring techniques advance. And we expect URLOST to be a popular unsupervised data analysis tool for these high-dimensional datasets.
>
> Both reviewer 1 and reviewer 3 agree that our “experimental evidence to support the proposed method is quite convincing,” and “the experiments cover a good range of complexity. ” We would love to try our model on a larger dataset in this domain if the reviewers or other reviewers could suggest additional datasets.
>
> __Q4__: “I do not fully understand, where exactly authors show that their method is not relying on topology.”
>
> __A4__: Take image as an example, for each pixel $x_i$, let $n(i)$ denotes the set of index of neighboring pixels of $x_i$, and $\mathbf{x}_{n(i)}$ denote the set of pixel values in its neighborhood $n(i)$. This topology is clearly defined in the perfectly grid-shaped images. Existing popular models rely on this as an important inductive bias (patchify in ViT, ViG, Convolutional kernel in CNN). However, in many cases of new data, they are not known beforehand. “Our method is not relying on topology” refers to the fact that even though the $n(i)$ is not defined, URLOST is able to recover the topology using the mutual information graph to define neighbors $n(i)$. We show that this enables the Transformer model and the masked-prediction task to become effective again.
>
> **Response continues in the comment**.

---

> ### Author Response · Authors · 2024-11-23
> **Author Response to Reviewer t33q (2/3)**
>
> __Q5__: “The transcriptomics vectors are still ordered (like each position corresponds to a specific gene), which could be also thought as topological organisation (like a flattened picture)”
>
> __A5__: Good point! Other works also attempted to leverage the sequential topological structure by ordering the transcriptomics vectors. They order dimensions by the location of the target chromosome, and then apply CNN-based neural networks. However, the performance of the CNN-type neural network is not as good as the fully connected neural network. The topological structure of transcriptomics vectors is highly complicated, which can not be treated as simple 1-d or 2-d structures. The author in [1] also suspects that “gene expression does not meet the translation equivariance assumption of CNNs.” This is likely why URLOST outperforms the CNN based method in [1].
>
> __Q6__: “The experiments are missing the error bars or confidence intervals, which makes it a bit hard to understand how robust is the method and how significant are the improvements. I would inspire the authors to add the error bars for all of the experiments”
>
> __A6__: We provide error bars for all experiments on the natural science dataset (gene and V1). For all CIFAR10 experiments, we don’t think it’s necessary to add error bars because the performance gain of URLOST vs other methods is significant.
>
> __Q7__:”I would suggest authors to take a point cloud dataset (for example, 3D Point Cloud Classification on ModelNet40 ) and perform a comparison on it [...] including some non-stationary time series classification example.”
>
> __A7__: The goal of URLOST is to apply to high dimensional data without explicit topology or stationarity from many natural science domains. Gene expression and V1 neural response are great examples because the topological structure of these domains is not even well-understood. Standard neural network structures and unsupervised learning methods do not perform that well on these dataset. Unlike gene and V1, 3d Point Cloud and time series are both dataset with regular topology. MAE can be applied to these dataset without too many additional changes. Pachification, as the common practice for ViT and ViG, can be achieved by using inherent topology for the respective data domain. There also exists other methods that apply MAE to these dataset [2][3]. In light of this, we do not think it is URLOST’s mission to work on point cloud and time series.
>
> __Q8__: while the paper motivation is highly biologically grounded, the actual image dataset has very low signal-to-noise ratio, eg a lot of natural scenes do not so nicely isolate a single object in the center of the image such as CIFAR. I would inspire the authors to use a more noisy multi-object classification or segmentation dataset
>
> __A8__: We want to emphasize the goal of this work is to develop an unsupervised learning method for high dimensional data with no topology or stationarity. Our model is motivated by the real-world biological vision signals. We could indeed run larger computer vision dataset with a foveated process. However, for either cifar10 or larger CV dataset, we will intentionally remove topological structure and stationarity from image data via the foveation process. Ideally, we want to apply URLOST to tasks where the data doesn’t have topology or stationarity to work with in the first place. Biologically visual signals recorded at V1 neurons are a great example. The goal is not to show how to URLOST and can solve object classification or segmentation tasks. Rather, we want to show URLOST performs better with the extra unique design compared to standard SSL methods. Despite the low signal-to-noise ratio in CIFAR10, we use the same dataset for all the baselines.
>
> We also note that single-object centric datasets such as CIFAR and ImageNet are used intensively in SSL literature and contribute a lot to the development of representation learning. SSL methods developed on them have proven to be effective in many downstream tasks. So we believe they are reasonable dataset choices for URLOST.
>
> **Response continues in the comment**.

---

> ### Author Response · Authors · 2024-11-23
> **Author Response to Reviewer t33q (3/3)**
>
> __Q9__: I find the biological motivation sometimes more confusing than helping, for instance, lines 471-476, mention self-organizing maps, without giving a brief introduction on what they are.
>
> __A9__: We kept the biological motivation concise in the introduction for the general audience. In short, the biological motivation for URLOST is: that even though the sensors of our visual system are in highly irregular patterns and even vary among individuals, we perceive regular and smooth perception. The summary of motivation of self-organizing layers is that we are born with random connections as if all the sensors in our eyes are permuted, we learn the correct connection by co-activation of nearby neurons from spontaneous neural activities.
>
> The self-organizing layers in URLOST are extensively introduced and discussed in the paper before line 471. The self-organizing map is a similar concept in the literature. We follow your suggestion by including a brief introduction and motivation for the self-organizing map in the relevant work section. We found this story truly fascinating, and this is what largely motivates us to work on this method. We hope you will enjoy it!
>
> Kohonen in 1982 presented a fascinating thought experiment[4]:
>
> > "*Suppose you woke up one day to find someone rewired your optic nerve (or you have been implanted with a prosthetic retina). The signals from the retina to the brain are intact, but the wires are all mixed up, projecting to the wrong places. Can the brain learn to "descramble" the image?*"
>
> Kohonen proposes the "self-organizing map" (SOM) algorithm to address this problem. SOM is an unsupervised machine learning technique used to produce a low-dimensional (typically two-dimensional) representation of a higher-dimensional dataset while preserving the topological structure of the data.
>
> The parameters of SOM include a set of neurons, where each neuron has a location index. Let **W_v** denote the weight vector of the *v*th neuron and **D** the input vector. **θ(u, v, s)** is the interaction function that determines the influence between the *u*th and *v*th neurons based on their distance. Through some simple learning rule, **W_v(s)** will learn to represent a set of "descrambled" pixels, where *s* represents the correct indices of the pixels.
>
> In other words, the index set defines the correct topology of the image as a 2D grid, and SOM maps the pixels onto this 2D topology by leveraging the mutual information between pixels at different locations.
>
> __Q10__: if I permute all images in the same way, why would it ruin the topology? As far as I understand it, I will just transform it but the ordering is still preserved
>
> __A10__: This is correct. A regular permute applied to all images only transforms the topology without destroying it. In other words, one would still know for certain the locations where a pixel’s original neighboring pixels are transformed. A random permutation to every image will destroy (or hide) the topology. This is what our experiment did.
>
> [1] Xiaoyu, Zhang. et al. 2021. OmiEmbed: A Unified Multi-Task Deep Learning Framework for Multi-Omics Data
>
> [2] Guo, Meng-Hao. et al. 2020. PCT: Point cloud transformer
>
> [3] Nie, Yuqi. et al. 2023 A Time Series is Worth 64 Words: Long-term Forecasting with Transformers
>
> [4] Teuvo Kohonen. Self-organized formation of topologically correct feature maps. Biological
> cybernetics, 43(1):59–69, 1982.

---

> ### Comment · Reviewer_t33q · 2024-11-23
>
> Thanks a lot for your clarifications.
> I guess, it would be nice to streamline a paper a bit based on the explanations above (details like in Q10).
>
> For Q3, the possible more recent dataset could be [1], which contains more than 5000 images, and several animals, which could be an interesting generalisation dimension. [2] could be another interesting dataset but it rather has dynamic stimuli (videos not images), however, decoding on videos is done as well [3].
>
> For Q2, it's a bit hard to respond as the appendix A10 is not attached and not in the original draft. I would prefer if you would share the details here.
>
> For Q8, as far I understand the paper so far, URLOST tries to reconstruct the topology with the help of clustering and further non-shared linear layer per cluster, which you call a 'self-organised layer'. My comment with signal-to-noise images was inspired by the fact that the whole idea highly relies on clustering and if the original vectors are noisy the clustering might be not helping much. Same for multiple object in a scene, if there are 2 objects in the image (lets say 2 rabbits), they are likely to end up in the same cluster, as the vectors are going to be close to each other and collapse two object in one. While, this might indeed not be the goal of this architecture, this limitations are not addressed in the paper
>
> [1] Willeke, Konstantin F., et al. "The sensorium competition on predicting large-scale mouse primary visual cortex activity." arXiv preprint arXiv:2206.08666 (2022).
> [2] de Vries, Saskia EJ, et al. "A large-scale standardized physiological survey reveals functional organization of the mouse visual cortex." Nature neuroscience 23.1 (2020): 138-151.
> [3] Bauer, Joel, Troy W. Margrie, and Claudia Clopath. "Movie reconstruction from mouse visual cortex activity." bioRxiv (2024): 2024-06.

---

> ### Author Response · Authors · 2024-11-24
> **Author Follow-up Response to Reviewer t33q - Part 1/2**
>
> Dear reviewer t33q,
>
> We really appreciate your quick reply and the additional comments. We modified the shuffling pixels idea in the introduction to streamline the paper a bit as you suggested. We want to thank you again for helping us to make our work clearer and easier to understand by the general ML audience. Please refer to line 93 for the change. The revised manuscript was not immediately uploaded last night. And now it is updated. Thank you for the reminder. We use the color blue to indicate all the additional content. Below is the follow-up response for Q3, we will come back for the scalability and signal-to-noise questions shortly.
>
> **For Q3**
> We acknowledge and value the reviewer's suggestion on the additional datasets. We cite the additional three datasets. We provide our observations on them in the following:
>
>  - **Data domain:** the modality of the data in the suggested datasets is very similar to the one we used for V1 (neural recording of mice), where we show URLOST is effective.
>
>  - **Data scale:** The competition dataset in [1] contains “a total of 25,200 unique images, with 6,000–7,000 image presentations per recording.” Each recording is performed on a different subject, and the model in the competition is trained and tested on each recording independently. [3] has around 1800 stimuli for each recording. The dataset we used has ~5600 stimuli from one subject, which is on the same scale as the proposed dataset. In comparison to the dataset in [1] and the dataset used in our work, the dataset in [2] is too small and contains only 118 natural image stimuli.
>
>  - **Adapting the task:** Although sharing the same data modality with our V1 experiment, the task in [1] and [3] are actually different from the tasks tackled by URLOST. Specifically in [1] and [3], the task is training on stimulus for predicting neural response. The input of the model has regular topology and stationarity (standard images or videos). The model is only trained in a supervised learning fashion. Whereas, for URLOST, the model is pretrained on the neural response (stationarity and topology not known) in an unsupervised fashion. We’re afraid we cannot use the starter kit provided in [1] and [3]. To adapt the task to a similar setting in our work, we will have the model pretrained on the neural response without providing the stimulus, then test if we can decode the stimulus or match the representation of neural response if they come from the same stimulus. The dataset in [3] contains video as the stimulus, which means the neural response has an additional temporal dimension. Although the neural response from each frame is a high-dimensional signal without topology or stationarity, the temporal dimension does have a linear structure. Since this additional structure is known and is a strong inductive bias, it is more fair and reasonable to further extend URLOST’s architecture in a similar way as extending MAE from images to videos[4][5]. Given this change to the model architecture. It will be an interesting future direction to adapt URLOST for high dimensional time series, where the high dimensional signal at each frame has no topology. We would like to thank the reviewer for suggesting the dataset [3] and providing future directions of building a video MAE URLOST.
>
> In summary, we find that these datasets are in a similar data domain as our existing task, of similar scale in terms of data amount, and require non-trivial effort to reformulate their task and data to our task. Therefore, reporting the results of URLOST on them is probably out of the scope of rebuttal of the original paper but more of an extension to the current work.
> That being said,  we are willing to put in some effort to adapt the data from [1][3] to a setting of unsupervised learning without topology or stationarity, the reviewer still thinks this is necessary.
>
> In the meantime, we would like to note that, during the rebuttal, we have presented several other experiments and ablation studies on our existing tasks to support our discovery and address some of the common questions raised by multiple reviewers, we hope that can also lift your concerns a little bit about the strength of URLOST’s empirical results.
>
> [1] Willeke, Konstantin F., et al. "The sensorium competition on predicting large-scale mouse primary visual cortex activity." arXiv preprint arXiv:2206.08666 (2022).
>
> [2] de Vries, Saskia EJ, et al. "A large-scale standardized physiological survey reveals functional organization of the mouse visual cortex." Nature neuroscience 23.1 (2020).
>
> [3] Bauer, Joel, Troy W. Margrie, and Claudia Clopath. "Movie reconstruction from mouse visual cortex activity." bioRxiv (2024): 2024-06.
>
> [4] Tong, Zhan, et al. "Videomae: Masked autoencoders are data-efficient learners for self-supervised video pre-training." NeurIPS (2022).
>
> [5] Feichtenhofer, Christoph, Yanghao Li, and Kaiming He. "Masked autoencoders as spatiotemporal learners." NeurIPS (2022).

---

> ### Author Response · Authors · 2024-11-24
> **Author Follow-up Response to Reviewer t33q - Part 2/2**
>
> Additionally, for your specific concern regarding “actual image dataset has very low signal-to-noise ratio” compared to CIFAR10, we perform an experiment by adding gaussian noise to CIFAR10 to specifically generate a dataset with low signal-to-noise ratio. We found out that while decreasing signal-to-noise ratio will lower the overall mutual information between pairs of pixels, the relationship between mutual information between pixels and image topology (distance between two pixels) does not change. We perform the following experiments: we add iid gaussian noise to each pixel with a fixed noise level, then we measure the mutual information between pairs of pixels as $MI((i,j),(k,l))$. For visualization, we aggregate all pairs by the distance between them and average their mutual information, i.e. let $A(a) = {(i,j),(k,l) s.t. \sqrt{(i-k)^2 + (j-l)^2}}$. We have $MI(a) = \frac{1}{|A(a)|} \sum_{i,j,k,l \in A(a)} MI((i,j),(k,l))$. $MI(a)$ denotes the mutual information between pixels of different distances.
>
> As shown in the table below, although the overall mutual information decreases a lot as noise level increases, the order of MI(a) is still the same. In other words, the mutual information is still the biggest when the two pixels are nearby, and decay when mutual information decreases. This implies the topology recovered by pairwise mutual information is robust even when signal has signal-to-noise ratio. It only breaks when you add too much noise ($\sigma = 0.9$). For this amount of noise, the image is not even distinguishable. This experiment strengthens our paper because it suggests it’s noise robust. We want to acknowledge the reviewer for the comment that leads to this experiment. We included this in Appendix.
>
>
> | d =       | 1.0    | 1.41   | 2.0    | 2.24   | 2.83   | 3.0    | 3.16   | 3.61   | 4.0    | 4.12   |
> |-----------|--------|--------|--------|--------|--------|--------|--------|--------|--------|--------|
> | $\sigma=0.0$ | 0.6596 | 0.4887 | 0.3752 | 0.3198 | 0.2792 | 0.2539 | 0.2229 | 0.1978 | 0.1819 | 0.1603 |
> | $\sigma=0.1$ | 0.3360 | 0.2656 | 0.2092 | 0.1816 | 0.1597 | 0.1457 | 0.1291 | 0.1150 | 0.1059 | 0.0942 |
> | $\sigma=0.2$ | 0.1342 | 0.1097 | 0.0889 | 0.0784 | 0.0700 | 0.0643 | 0.0574 | 0.0516 | 0.0477 | 0.0435 |
> | $\sigma=0.3$ | 0.0580 | 0.0491 | 0.0411 | 0.0369 | 0.0334 | 0.0309 | 0.0283 | 0.0261 | 0.0246 | 0.0229 |
> | $\sigma=0.4$ | 0.0297 | 0.0258 | 0.0223 | 0.0205 | 0.0191 | 0.0182 | 0.0168 | 0.0159 | 0.0153 | 0.0146 |
> | $\sigma=0.5$ | 0.0189 | 0.0172 | 0.0153 | 0.0143 | 0.0135 | 0.0131 | 0.0127 | 0.0123 | 0.0120 | 0.0116 |
> | $\sigma=0.9$ | 0.00954| 0.00919| 0.0090 | 0.00903| 0.00905| 0.00915| 0.00911| 0.0090 | 0.00904| 0.00899|

---

> ### Comment · Reviewer_t33q · 2024-11-26
>
> Thanks a lot for the updates.
> I would like to thank the authors for the engaged rebuttal.
> The paper is an interesting proof-of-concept. However, I am still dissatisfied with the way the authors addressed the scalability and I do believe that clustering many (10^5, as a 512x512 image for the sake of example) high-dimensional points might be slow and requires scalability benchmarking. For a proof-of-concept work it might not be crucial to have such a benchmark but I believe it is crucial to mention the possible limitations. Other limitations of the suggested are not clearly stated in the paper as well, while I would expect them in the discussion section.
> I would kindly ask the authors to adjust the writing based on it, such that I could improve my score.
>
> Minor comment:
> Appendix A9 on the computational costs is also not what I would have expected as instead of providing a table with the time consumed by the different parts of the model (numerical evidence) and used hardware it provides a lot of text not supported by the evidence. (table 6 is about number of parameters but relative not absolute, why? does not this ratio change between the models?).

---

> ### Author Response · Authors · 2024-11-27
> **Author Follow-up Response to Reviewer t33q Part 3 (1/3)**
>
> We thank the reviewer’s engagement during the rebuttal as well! We could see the reviewer truly helped us shape our work better during the review process. We are happy to reflect on our discussion in the discussion and limitation sections of the paper. The following is our detailed response:
>
> We didn’t worry about the computational cost of clustering and pairwise mutual information because they are computed offline and not part of training the neural networks. Both clustering and mutual information are computed and saved for all datasets at once (in less than a day), and we never come back to re-compute it. During MAE training, the clustering results are directly used to group the dimensions of the input, which only causes a small computation overhead compared to the forward and backward computation of deep networks. Therefore, in table 9 we cared about the relative percentage of increased parameters compared to the original MAE, rather than paying attention to the actual wall-clock time of each module.
>
> That being said, we agree with the reviewer that introducing URLOST induces computation overhead in two folds:
>  - Precompute the mutual information and spectral clustering on the dataset. This is quadratic to the input dimensions as the reviewer’s insight, but the computation is not involved in training, so we can think of it as being amortized by the comparatively longer training.
>  - On the fly with the model, cluster the input dimensions and compute self-organizing. This computation happens for every data point in every iteration during training and inference, but the overhead is small compared to the computation of the original deep network.
> We would like to acknowledge the reviewer for rigorously paying attention to this aspect, and we include the discussion of this in the limitation section.
>
> We highly value the reviewer’s suggestion and perform a **scalability benchmark** with the current implementation. We resize an image dataset to different resolutions (i.e. different number of dimensions). At each resolution, we compute mutual information and clustering for a different number of pixels (dimensions). We provide benchmarks for computing clustering and mutual information separately. We update Appendix A.9 with these runtime plots. As expected, both the computational cost of computing pairwise mutual information and clustering both scales quadratically with the number of dimensions of the signal. It appears that clustering is not the bottleneck of the computation. For clustering, images of size 100x100 (10,000 pixels) only take 0.1 mins and images of size 224x224 (50,176) pixels take 2.2 mins. However, computation of pairwise mutual information is much slower than clustering as shown in the figure, for example, 100x100 (10,000 pixels) takes 152 mins. Although the computational approach is still feasible for all the datasets we used and the dataset mentioned by the reviewer (d < 10,000), it will become an issue for extremely high dimensional data such as 10^5 dimensions. Although it is a pre-computation before model training, It will definitely take a long time to compute with the current implementation. We included this issue in the additional limitation section in the manuscript.
>
> **On accelerating the operators.** We also want to reassure the reviewer that these limitations can be improved by implementing accelerated CUDA kernels specialized for clustering and pairwise mutual information. The key computational module of spectral clustering is eigendecomposition and similarity search, which have GPU implementation from pytorch and Faiss, respectively. Efficient computation of mutual information requires the “bincount” operator to be implemented on GPU. Although pytorch has a “bincount” operator implemented, it does not currently support batchwise computation, which could be a bottleneck of the computation. Still, having a batchwise “bincount” operator on GPU in the future seems feasible.
>
> **Overall**, we sincerely appreciate the reviewer for being responsible and pointing out the limitations of our work. Although we did initially include some limitations in the discussion section, they are not made to be very clear. We agree with the reviewer and now dedicate an entire section to discussing the limitations of our work. We also love this format because it makes our work more complete. *We made the following changes*:
>
>
> Include an additional section in the main manuscript to discuss limitations.
>
> Included runtime benchmark for mutual information calculation in Appendix A9.
>
> Included runtime benchmark for clustering in Appendix A9.

---

> ### Author Response · Authors · 2024-11-27
> **Author Follow-up Response to Reviewer t33q Part 3 (2/3)**
>
> As the reviewer suggested, we added a new section dedicated to discussing limitations starting at line 527, Section 6:
>
> During experiments, we found that clustering is crucial for the quality of representation learning. Unlike the self-organizing layer, the procedure for aggregating similar dimensions together is separate from the training of MAE. Given the effectiveness of the self-organizing layer, this separation could be sub-optimal and could be improved. Learning the clusters end-to-end with the representation via back-propagation is worth future investigation. Additionally, the computational costs of computing pairwise mutual information and clustering both scales quadratically with the number of dimensions of the signal. Although each procedure only needs to be performed once, the computation could become too slow and infeasible for extremely high-dimensional data ($d > 10,000$) with our current implementation. We provide a benchmark on the runtime of our implementation in the Appendix. Nevertheless, we think this problem could be resolved by a GPU implementation for clustering and mutual information. Additionally, although natural science datasets such as TCGA and V1 calcium imaging are considered large datasets in their respective domains, they are not even close to the scale of popular domain data like computer vision. We expect larger natural science datasets will emerge as measuring and imaging technology advances. It's a great direction to perform URLOST on even larger datasets.

---

> ### Author Response · Authors · 2024-11-27
> **Author Follow-up Response to Reviewer t33q Part 3 (3/3)**
>
> Regarding the minor comments about the computation costs and evidence, we value the reviewer's comment and made the following revisions by **updating and adding figures and tables** with numerical evidence, in addition to the discussion posted above:
>  - In Table 6, we provide the ratio between the parameters of self-organizing layers and the parameters of the entire model.
>  - In Figure 9 we provide the runtime plots for computing spectral clustering and mutual information given different number of dimensions.
>  - In Table 7, we profile the detailed running time of the individual modules (clustering, self-organizing layer, and Transformer blocks).
>
> Please kindly refer to the updated manuscript for the details in the tables and the figures.
> In addition, we made this reasoning more clear in the limitation section.

---

> > ### Comment · Reviewer_t33q · 2024-11-27
> >
> > Great, thanks for addressing the comments. I believe the updated discussion section and figures in the appendix improved the paper.
> > I believe a score of seven would suit me better but as it is not there, I used eight.

---

> > > ### Author Response · Authors · 2024-11-27
> > >
> > > Thank you for your encouraging feedback. We also believe the additional limitation section and runtime experiment improved the quality of the paper! Thank you for all the suggestions!

---

### Official Review · Reviewer_u6uR · 2024-11-01

**Soundness:** 2
**Presentation:** 2
**Contribution:** 2
**Rating:** 3
**Confidence:** 4

**Summary:**

The paper proposes a method to learn useful representations of data without assuming the relationships between dimensions of topological structure, translation invariance, as in images/sounds.

**Strengths:**

The paper is well motivated and the task tackled is important and relevant to the community.

**Weaknesses:**

A lot of work has gone in to the paper, but unfortunately it is quite difficult to follow; and the proposed method seems very expensive (computing entropy, eigenvalues, clustering, etc). On the first point, to reach a broad audience, specific effort should be made regarding aspects less familiar to the general ML community. I find the method and data transformations difficult to parse.

* Clarity
   * [036-041] Unclear references to topology and stationarity -  core aspects of the paper
      * Is the topology property that dimensions are ordered such that correlation (or similar) is expected between the signal at "nearby" dimensions?
      * Stationarity seems very entwined with topology, does stationarity require topology? If not how?
      * What assumptions *is* the model making?
   * [225] unclear how permuting pixels leaves stationarity intact
   * [259] unclear how reducing density prevents redundant sampling.
* [095] A fully connected NN doesn't assume topology/spatial invariance and seems a key benchmark, but is not considered/mentoined.
* [183] How does a vector denote a cluster? the cluster mean, indexes of cluster members?
* [337] Given $\beta$-VAE results in Table 2 are close on the real datasets, have optimal parameters/architecture been used? Why is this not compared to in Table 1?
* The proposed method seems expensive but no mention of this is made.

Minor
* [031] unsupervised representation learning (UL) and self-supervised representation learning (SS) are not the same. SSL is a subset of UL, e.g. a $\beta$-VAE performs UL but not SSL, whereas SimCLR performs SSL (and so UL).
* Several typos
   * [037] "arised", [055] "singal" etc

**Questions:**

see Weaknesses

---

> ### Author Response · Authors · 2024-11-23
> **Author Response to Reviewer u6uR (1/3)**
>
> Thank you for your thoughtful review and questions. It indeed requires non-trivial efforts to explain the content that is across multiple domains (SSL, ML&stats, bio) and we value your feedback and help in strengthening the paper. You suggest the main weakness of the paper is that it needs additional clarification. The clarification mainly involves two ideas: computational expense, and definition of topology and stationarity. We added a detailed definition of topology, stationarity, and the interplays between these two quantities in the revised manuscript. We also added a section for computational expenses. In short, the method is not that expensive. The runtime of calculating pairwise mutual information, spectral embedding, and clusterings is far less comparable to training MAE.
>
> Here’s a list of changes we made to the manuscript according to your suggestions:
>  - Added formal definition of stationarity and topology in the context of our work in Appendix A.9.
>  - Added an example of data with “topology but no stationarity in Appendix A.9.
>  - Added explanation for why permuted cifar10 has stationarity but no topology in Appendix A.9.
>  - We modified self-supervised learning baseline to unsupervised representation learning baseline in the experiment section.
>  - Added a section on computational cost in Appendix A.10.
>
>
> We also provide detailed answers to each of your questions.
>
> __Q1__: [036-041] “Is the topology property that dimensions are ordered such that correlation (or similar) is expected between the signal at "nearby" dimensions?”
>
> __A1__: Yes this is a correct understanding. URLOST essentially processes the high dimensional data to reconstruct a topology by placing similar dimensions to be nearby. Similarity is defined via mutual information between dimensions. We assume the unknown topology of the domain of the signal can be roughly recovered by the pairwise mutual information matrix (graph).
>
> __Q2__: [036-041]  “Stationarity seems very entwined with topology, does stationarity require topology? If not how? [225] unclear how permuting pixels leaves stationarity intact.”
>
> __A2__: Thanks for the good question. The definition of stationarity depends on the topology of the data. It is difficult to define stationarity without knowing the topology. When we refer to data that has no topology but is stationary, we mean the data is stationary with the “recovered topology” using URLOST. We want to emphasize that our model assumes we don’t know the underlying correct topology of data. Specifically, we assume the underlying topology can be recovered by the mutual information graph. Then the stationarity is defined on the recovered topology. We expanded the mathematical definition for stationarity and topology in the manuscript as suggested by the reviewer for better clarity. We provide an explanation for why “permuting Cifar” will destroy the topology but leave stationarity intact. We also provide another example for a dataset with “topology but no stationarity.” Please kindly refer to the Appendix A.9 in the newly uploaded revised manuscript for more details. For convenience, we copy it here:
>
> Take image as an example, for each pixel $x_i$, let $n(i)$ denotes the set of index of neighboring pixels of $I$, and **$x_{n(i)}$** denote the set of pixel values in its neighborhood $n(i)$. The distribution of a region in the image (e.g. image patches) can be modeled by the distribution $p([x_i | x_{n(i)} ])$. This distribution is complicated and is not invariant over space, but some statistics of this distribution could be invariant. Let $S_f (i) = E_{p({\bf x} \in [x_i | x_{n(i)} ]) } ({\bf f(x)})$ denotes the statistic of f at position i. For example, f could be mean or standard deviation. Or other higher-order statistics learnt by the neural network. “Topology” refers to how the neighborhood $n(i)$ is defined for each $i$. Stationarity refers to the property that $S_f(i)$ is the invariant for all $i$.
>
>
> __Q3__: Why “permuting Cifar” will destroy the topology but leave stationarity intact?
>
> __A3__: We provide the intuition in Appendix A.9. Permuted Cifar10 is an example we included in the manuscript for this criterion (no topology but has stationarity).
>
> "No topology" refers to the fact that $n(i)$ is not defined after random permutation. "Has stationarity" means that the signal is stationary when the topology is recovered. Specifically, we use the mutual information graph to define neighbors $n(i)$. Under this particular definition of $n(i)$, there exists a meaningful invariant statistic $S_f(i)$ for the signal. Thus, the signal is stationary.
>
> This is because the topology recovered is very similar to the original 2D topology of images before permutation, and the signal before permutation is stationary. However, under an incorrect definition of $n(i)$, the signal is no longer stationary.

---

> ### Author Response · Authors · 2024-11-23
> **Author Response to Reviewer u6uR (2/3)**
>
> __Q4__:Can we have an example of data with “topology but no stationarity.”
>
> __A4__: We provide the intuition in Appendix A.9. We didn’t provide an example for this in the manuscript. One example would be foveated images simulated using convolutional kernels. One could make a different region of an image have different resolution while preserving the 2d grid structure of the image by using laplace pyramid [1][2]. As a result, the image is still represented as a 2d matrix, the definition of $n(i)$ is the same as regular images. However, many invariant statistics before the foveation process are destroyed due to the imbalance resolution at different regions of the image.
>
> __Q5__: [259] unclear how reducing density prevents redundant sampling.
>
> __A5__: This density refers to how we simulate the foveation process. It is a different concept from the “density” of “density adjustment” in our spectral clustering. Here we provide a detailed explanation of the foveation process and how the density works in the process:
>
> We represent each ganglion cell (sensor) as a Gaussian kernel centered at a particular location on the image. Each Gaussian kernel has a position and width as described in Appendix A.3. Larger sensors sample a large region of the visual field but it is blurry. Smaller sensors sample a small region but have higher resolution. Foveated vision refers to the fact that the human retina has much more sensors in the center of the visual field, and much less sensors in the spherical. “Density” refers to how dense the sensors are in a particular region. For biological vision, the sensors are way denser in the center of the visual field than the peripheral.
>
> CIFAR10 has very low-resolution. To simulate the foveation process, we first upsample each CIFAR10 image to be bigger. If we place too many sensors in the center of the resampled CIFAR10 image, then sensors in these regions will sample redundant information due to the upsampling procedure. “Lower the density” refers to putting less sensors in the center of the image. Note that we still put more sensors in the center than in the peripheral for simulation. It’s just for the real retina, this imbalanced density between center and peripheral is more exaggerated. We acknowledge the reviewer that the description of simulated foveated sampling is hard to understand for the audience from general ML background. We have provided figures (Fig. 5) illustrating it in the original manuscript to aid understanding, and now we simplified it and provided better intuition in the updated manuscript.
>
> __Q6__: [095] A fully connected NN doesn't assume topology/spatial invariance and seems a key benchmark, but is not considered/mentioned.
>
> __A6__: A fully-connected network served as a baseline for V1 and gene data in table 2. The beta-VAE baseline uses a fully-connected network on both tasks with a similar number of hidden dimensions, number of layers, and extensive hyperparameter tuning. This is explained in the Appendix A.5. We made it more explicit in the experimental section as suggested by the reviewer. Fully-connected networks are known to be less competitive on vision tasks like CIFAR so we did not conduct a beta vae experiment.

---

> ### Author Response · Authors · 2024-11-23
> **Author Response to Reviewer u6uR (3/3)**
>
> __Q7__: [183] How does a vector denote a cluster? the cluster mean, indexes of cluster members?
>
> __A7__: It refers to a subset of dimensions indexed by each cluster member. Let n(i) denote the indexes of the $i$th cluster members. Let $x$ be the high dimensional data vector, $x^{(i)}$ = [$x[j]$ for $j$ in $n(i)$ ].
>
> __Q8__: [337] Given β-VAE results in Table 2 are close to the real datasets, have optimal parameters/architecture been used? Why is this not compared to in Table 1?
>
> __A8__: We believe we have optimal parameters and architecture for the beta-vae baseline. We put in a significant amount of effort for tuning the β-VAE baseline. The baseline we got actually outperformed the usage in the original paper on the same dataset [3]. We also make sure the number of layers and hidden dimensions we used for MAE and β-VAE are similar. This is explained in Appendix A.5.
>
> __Q9__: [031] unsupervised representation learning (UL) and self-supervised representation learning (SS) are not the same. SSL is a subset of UL, e.g. a β-VAE performs UL but not SSL, whereas SimCLR performs SSL (and so UL).
>
> __A9__: Thanks for the comments. We modified self-supervised learning baseline to unsupervised representation learning baseline in the experiment section.
>
> __Q10__: The proposed method seems expensive but no mention of this is made.
>
> __A10__: We provide an additional section in the appendix A.10 to discuss the computational cost and scalability. In short, the computational cost and scalability of URLOST is nearly the same as MAE.
>
> [1] Perry, Jeffrey S., and Wilson S. Geisler. "Gaze-contingent real-time simulation of arbitrary visual fields." Human vision and electronic imaging VII. Vol. 4662. International Society for Optics and Photonics, 2002.
>
> [2] Jiang, Ming, et al. "Salicon: Saliency in context." Proceedings of the IEEE conference on computer vision and pattern recognition. 2015.
>
> [3] Xiaoyu, Zhang. et al. 2021. OmiEmbed: A Unified Multi-Task Deep Learning Framework for Multi-Omics Data

---

> ### Author Response · Authors · 2024-11-27
> **Any Feedback?**
>
> Thank you once again for taking the time to evaluate our work and providing such insightful comments!
>
> We would like to check if you’ve had a chance to review our response and whether it has adequately addressed your major concerns. If you have any additional feedback, please let us know at your earliest convenience.
>
> Here are some important deadlines to keep in mind:
>
> **November 27th, 11:59 PM AoE: Last day for authors to upload a revised PDF.**
> After this deadline, no further updates to the manuscript will be possible, and authors will only be able to respond to comments on the forum. If you’d like any changes reflected in the revised manuscript, please inform us before this time.
>
> **December 2nd: Last day for reviewers to post messages to the authors (six-day extension).**
> This is the final opportunity to share any remaining concerns with us.
>
> **December 3rd: Last day for authors to post messages on the forum (six-day extension).**
> After this date, we will no longer be able to respond to any concerns or feedback.
>
> Once again, we truly appreciate your voluntary effort and valuable input in helping us refine our work!

---

> ### Author Response · Authors · 2024-11-28
> **Reminder for Feedback - 2**
>
> As the deadline to modify the PDF approaches in a few hours, it is unlikely that any new changes can be incorporated into the current revision. We apologize for this limitation in the final hours. However, we remain committed to addressing any additional concerns or questions before December 3rd.
>
> Here are some important deadlines to keep in mind:
>
> ~~**November 27th, 11:59 PM AoE: Last day for authors to upload a revised PDF.** After this deadline, no further updates to the manuscript will be possible, and authors will only be able to respond to comments on the forum. If you’d like any changes reflected in the revised manuscript, please inform us before this time.~~
>
> **December 2nd: Last day for reviewers to post messages to the authors (six-day extension).** This is the final opportunity to share any remaining concerns with us.
>
> **December 3rd: Last day for authors to post messages on the forum (six-day extension).** After this date, we will no longer be able to respond to any concerns or feedback.
>
> Once again, we truly appreciate your voluntary effort and valuable input in helping us refine our work!

---

> ### Comment · Reviewer_u6uR · 2024-11-28
> **Apologies for late response, score lowered**
>
> **TL:DR**: key concerns not addressed, related work not compared to
>
> Thanks you for your reply and genuine apologies for my late response. I drafted a response some time ago and thought it had been submitted and I have just realised it was not.
>
> Unfortunately, the changes to the paper do not address my main concerns. In particular
> * **Key definitions**: the definitions added in the appendix of **the key terms** in the paper (topology and stationarity) are neither "formal" nor clear. Furthermore the wording seems confused: S is an *expectation* of a *statistic* f (not S) of the data x. And there appears to be literally no attempt at a "formal definition", i.e. a clear mathematical description, of what these central terms mean.
>   - Topology appears to correspond to "correlation" between dimensions (e.g. neighbouring pixels or nearby timesteps), which is closely related to mutual information.
>   - Stationarity appears to correspond to what is typically referred to as invariance, e.g. spatial invariance, time invariance.
>
>    Further, it is unclear why standard mathematical terms are being redefined while more appropriate standard terms not used.
> * **Related works**: this work seems very related to [1,2], which aim to be domain agnostic and use masked auto-encoders. Comment and empirical comparison to those seems necessary.
> * **Computational cost**: The computational cost analysis seems flawed as parts that appear expensive (computing pairwise mutual information between dimensions and clustering) are effectively dismissed as "pre-computed" ("mutual information and the clustering results are precomputed... thus do not introduce more cost"). By this logic some algorithms would be free. For fair comparison between algorithms, the analysis should include all aspects of the algorithm necessary given a new data set.
> * **Typos**: those previously pointed out remain.
>
> I apologies again that this response is late as I genuinely thought this had been posted. However, my previous score was tentative subject to clarifications, and I am now more confident that the paper is unready for publication, so I am updating my score to reflect that $5 \to 3$.
>
> [1] Self-Guided Masked Autoencoders for Domain-Agnostic Self-Supervised Learning
>
> [2] Perceiver: General Perception with Iterative Attention

---

> ### Author Response · Authors · 2024-11-29
> **Author Follow-up Response to Reviewer u6uR - Part 1/8**
>
> > "Thanks you for your reply and genuine apologies for my late response. I drafted a response some time ago and thought it had been submitted and I have just realised it was not .. I apologies again that this response is late as I genuinely thought this had been posted."
>
> Thanks for responding to our reply!
>
> We provided the answers to address your previous concerns and comments a week ago and were hoping to start the discussion much earlier. Since your response came right after the PDF revision portal has closed, we are no longer able to make additional modifications in the PDF. However, we will try our best to address your new comments and remaining concerns. These efforts will be reflected in the next revision whenever possible.

---

> ### Author Response · Authors · 2024-11-29
> **Author Follow-up Response to Reviewer u6uR - Part 2/8**
>
> > "Key definitions .. Furthermore the wording seems confused .. Further, it is unclear why standard mathematical terms are being redefined while more appropriate standard terms not used."
>
> **On topology of the domain of signals.** A signal is a function $X(s)$ defined on some domain $S$, where $s\in S$. Let’s focus on $S$. While the definition of topology is a little abstract and relies on open sets, the intuition is that it defines a generalized sense of nearness, or whether two points are close to each other or far apart. A common way to give such a space the property of being near or far away is with an explicit distance function, or a metric. The distance function generates a class of objects—open sets. These spaces (with an associated metric) are called metric spaces. For example, the topology of image space is induced by the L2 norm as the metric function. However, not every space will have a natural metric on it. This is why we need to use the term “topology”.
>
> **Topological spaces are generalizations of metric spaces.** The definition of topology is any collection of subsets of $S$ that satisfies certain axioms: close under arbitrary union and finite intersection. In the practical setting, the domain $S$ is always discretized as a set of indices, and we can assume that we always deal with a signal defined on a graph [11]. Graphs can be embedded in $\mathbb{R}^3$. For a finite graph, the graph topology is just the subspace topology inherited from the usual topology in $\mathbb{R}^3$. Under this specific definition, the word “graph,” “adjacency matrix,” and “topology” can be used interchangeably. In this work, we assume we are not provided the topology. Thus, using pairwise mutual information as the adjacency matrix is a way to model the topology. Correlation is another way to model the topology, where we use only second-order pairwise statistics. This is slightly different from the reviewer’s understanding “Topology appears to correspond to "correlation" between dimensions.” More generally, graphs are topological spaces that are the simplicial 1-complexes or the 1-dimensional CW complexes.
>
> Using the term topology is to keep the most general conceptual setting. Intuitively, for a high-dimensional signal, we need to find the neighbors of each of the dimensions and so that the closely dependent dimensions are processed together. We acknowledge using topology here is an unnecessary but preferred choice by the authors. But this slight abuse of the terminology is no more than using “image manifold” in many papers even though the authors must not mean that images are from one manifold. And, here we don't mean that we will recover the open sets, rather, we find the nearness between different dimensions.

---

> ### Author Response · Authors · 2024-11-29
> **Author Follow-up Response to Reviewer u6uR - Part 3/8**
>
> In the following we will provide a formal and abstract definition of the stationarity and an intuitive understanding.
>
> **On stationarity.** In probability theory, a random process is a family of random variables. While the index was originally and frequently interpreted as time, it was generalized to random fields, where the index can be an arbitrary domain. That is, by modern definitions, a random field is a generalization of a stochastic process where the underlying parameter need no longer be real or integer valued "time" but can instead take values that are multidimensional vectors in $\mathbb{R}^n$, points on some manifolds and Lie Groups, or graphs [3, 4, 5, 6, 7, 8]. Let’s consider a real-valued random field $X(s)$ defined on a homogeneous space $S=${$s$} of points $s$ equipped with a transitive transformation group $G=${$g$} of mappings of $S$ into itself, and having the property that the values of the statistical characteristics of this field do not change when elements of $G$ are applied to their arguments. In convention, we call a random field $X(s)$ a strict-sense stationary (or strict-sense homogeneous) random field if for all $n = 1, 2, \cdots $ and $g \in G$, the finite-dimensional probability distribution of its values at any $n$ points $s_1, \cdots, s_n$ coincides with that of its values at $gs_1, \cdots, gs_n$. We call $X(s)$ field wide-sense stationary (or wide-sense homogeneous) random field if $E|X(s)|^2<\inf$ and $EX(s) = EX(gs)$, $EX(s)X(s_1) = EX(gs)EX(gs_1)$ for all $s$, $s_1$ $\in S$ and $g \in G$. Or wide-sense stationarity means the first and second-order statistics do not change w.r.t. the group action. Further, we can intuitively generalize the wide-sense stationarity to higher-order statistics if we need to define any nth-order stationarity between strict-sense and wide-sense.
>
> Intuitively, stationarity just means the statistics does not change w.r.t. transformations. Let's take the image as an example, it is a signal $X(s)$ defined on $S=\mathbb{Z}^2$. And let's define the group action as translation. If we use a convolution neural network and learn a set of $L\times L$ filters or use a VIT to learn a set of shared features on $L\times L$ patches. A hidden assumption is that the statistics of our image does not change for $L\times L$ region, or in order words the joint probability $P( X(1,1), \cdots, X(L,L) )$ = $P( X(1+a,1+b), \cdots, X(L+a), \cdots, X(L+b) )$, where $(a, b) \in$ translation group $T$.
> While we have initially cited [8], it might be a good idea to provide a more thorough explanation in the appendix, hence, we have included another five references to provide a more comprehensive point of view.
>
> So, essentially, stationarity is invariance of probability characteristics w.r.t. group actions.

---

> ### Author Response · Authors · 2024-11-29
> **Author Follow-up Response to Reviewer u6uR - Part 4/8**
>
> What is considered appropriate or standard terms highly depends on the background of the audience. The concepts and terms we use are standard and ordinary to some of the audience, and potentially alien to the rest. Many of the ‘new’ ideas in this field have been discussed in the past, however, one would be surprised how many classical references were omitted. While we can maintain different opinions on what terms to be used, it would be strange to dismiss a paper on a minor vocabulary disagreement.
>
> Since these terminologies can be opaque to some of the audience, in order to be more inclusive, we will provide a more thorough introduction to the definition of stationarity and topology in the appendix, with both formal and an intuitive version. But after all, we aim to keep the concepts intuitive: stationarity means the probability of the signal does not change with respect to group action on the domain of the signal, say, a translation. Topology intuitively means the connection between different dimensions and thus some of them should be processed together.
>
> Please let us know if we have addressed your concern.

---

> ### Author Response · Authors · 2024-11-29
> **Author Follow-up Response to Reviewer u6uR - Part 5/8**
>
> > "*Related works: this work seems very related to [1,2], which aim to be domain agnostic and use masked auto-encoders. Comment and empirical comparison to those seems necessary.*”
>
> We thank the reviewer for pointing out these two related works. They are related to our work in that we share the same goal and similar intuition in extending the current SSL architecture to being less dependent on the data domain. Additionally, all of us also utilize the Transformer architecture. However, our overall architecture is still different from theirs. We want to point out that this problem is not new. It is known as the Magic TV problem proposed by Kohonen in 1982 in the literature of early computational neuroscience [10]. Due to the insufficient tools in machine learning at the time, the proposed solution by Kohonen is limited. The module we designed in URLOST is strongly motivated and connected to the original problem. They are designed to explicitly recover the topology of the underlying signal. On the other hand, while [1][2] are solving the same problem, the reasons behind the design of the “Perceiver architecture” are also implicitly recovering the underlying topology of a signal. This important connection is not realized yet in [1][2]. We do not claim that recovering the topology explicitly is better than the implicit approach in [1][2]. They have their different advantages. Therefore, we believe our work is in complement with theirs, forming a joint effort in unsupervised representation learning for data without foreknown structure, and with a deeper connection and motivation from the research in the field of neuroAI. We will add a section to further discuss the relevance and connections between our work and the related work. Due to the late reply from the reviewer, we will have to include a short discussion in the next revision version together with a prolonged version in the appendix. Below, we provide further discussion on the connections and differences between our work and the related ones:
>
> **Connection to self-organizing map (SOM)**: We would like to first point out that both [1][2] are very similar to the very early topology recovering algorithm for high dimensional data, SOM. The SOM starts out with a set of neurons. Each neuron is attached to an index. These indexes define the topology of these neurons. The weights of each neuron is represented by a vector of the same size of input. The weights are then fitted to the data. Each neuron essentially acts like the centroid of a cluster for clustering different dimensions of the data. In [1], the essence of novel architecture of the Perceiver lies in the first layer of the model. The model has a “latent array” that is essentially the same as the neurons used in SOM, then learn to be the cluster centroid in the input signal. There is a slight difference between SOM and the Perceiver. Unlike self-organizing maps, “latent array” and input array in [1] are different dimensions, they are mapped to the same dimension via key, query and value matrix. [2] is a followup work of [1] which combines the Perceiver architecture and masked-prediction objective. It utilizes the “implicit” clustering learnt in the first layer of Perceiver to mask the input.

---

> ### Author Response · Authors · 2024-11-29
> **Author Follow-up Response to Reviewer u6uR - Part 6/8**
>
> **Difference between our work and related works**: we’re highly motivated by SOM. But instead of implicitly learning to recover the topology of the underlying signal like in [1][2] and SOM, we use mutual information to explicitly recover the topology and use spectral clustering to explicitly make clusters to aggregate similar dimensions of the data. We favor the explicit formulation and also believe in the contribution of this work. We do not claim our way of doing it explicitly is always better than doing it implicitly like in [1][2]. They are two approaches for solving similar problems. The explicit approach has several advantages. It makes the whole process more explainable. Unlike in [1][2], it makes the entire clustering and learning mask policy procedure offline. One only needs to compute mutual information and spectral clustering once for each dataset. Including the learning masking policy as part of the end-to-end process could be advantageous as discussed in the future direction section of our manuscript. However, it is not necessarily better because it could slow down the training, or the model could learn simple masking policy results in lower mask prediction loss but encourage the model to learn trivial representation. “KeepTheTopk” operator in [2] is a design to alleviate this problem. Moreover, both intuitively and empirically, we show the projection layer or the first attention layer in the Perceiver should not have shared weights. This is the key motivation for the “without stationarity” part of the paper and the proposed “self-organizing layer.” This is also a novel part of our work that differs from SOM and Perceiver. Also, the motivation between our work and [2] is completely different, as stated in [2], it is mainly motivated by Perceiver. [2] stems from the observation that the first layer of Perceiver learns to attend each vector in the latent array sequence to “a continuous group of highly correlated tokens” in the input sequence. They use this idea to form a good masking policy for masked autoencoders. Although this is a clever observation, throughout the entire paper, they did not explain the motivation behind why Perceivers learn such a good masking policy. As explained earlier, the first layer of Perceiver acts very similar to SOM and is implicitly recovering the topology of the signal. This connection is not made in [2].
>
> Our approach and the related work can be thought as complementary to each other and could potentially be combined. Additionally, [2] did not discuss other related work like flexiVIT and patchGT mentioned by reviewer sHq9. We also found similar recent work [9] around the same time. We believe all these works should count as concurrent parallel works. Given none of these works cites SOM, and they do not cite each other, we don’t think it’s fair to reject our work purely based on it misses two related works. Nevertheless, we will indeed include these related works and the connection between them in our manuscript. We believe making these connections is a contribution and further strengthens our work. Due to the late reply from the reviewer, we cannot modify the manuscript, but we will add them for the camera-ready version of the manuscript.
>
>
> Finally, we want to point out that this problem of learning the underlying topology of signals is a very important problem, it started as early as in 1982. Through this rebuttal process, we learn that there are many emergent approaches to solve the similar problem. This further shows it is an important problem. We think the beauty of the rebuttal process is to learn from other experts in the field, making connections, and discover unknown approaches for the same important problem. We would love to discuss with the reviewer and acknowledge the suggestion by the reviewer. We kindly ask the reviewer to reply to us timely during rebuttal if the reviewer truly wants to help us to make our paper and the community better. We will address these concerns for the camera ready paper.

---

> ### Author Response · Authors · 2024-11-29
> **Author Follow-up Response to Reviewer u6uR - Part 7/8**
>
> [1] Andrew Jaegle et al. Perceiver: General Perception with Iterative Attention
>
> [2] Johnathan Xie et al. Self-Guided Masked Autoencoders for Domain-Agnostic Self-Supervised Learning
>
> [3] Andrew Jaegle et al. Perceiver IO: A General Architecture for Structured Inputs & Outputs
>
> [3] Erik Vanmarcke, Random Fields: Analysis and Synthesis
>
> [4] David Mumford, Agnes Desolneux, Pattern Theory: The Stochastic Analysis of Real-World Signals
>
> [5] Michael Taylor, Random Fields: Stationarity, Ergodicity, and Spectral Behavior
>
> [6] Michael Jordan, Zoubin Ghahramani, Tommi S. Jaakkola, Lawrence Saul, AnIntroduction to Variational Methods for Graphical Models
>
> [7] A.G. Ramm, Random fields estimation theory
>
> [8] Hermine Bierme, Introduction to random fields and scale invariance
>
> [9] Self-Guided Masked Autoencoder
>
> [10] Kohonen, T. The self-organizing map.
>
> [11] Shuman et al., The Emerging Field of Signal Processing on Graphs: Extending High-Dimensional Data Analysis to Networks and Other Irregular Domains

---

> ### Author Response · Authors · 2024-11-29
> **Author Follow-up Response to Reviewer u6uR - Part 8/8**
>
> > “Computational cost: The computational cost analysis seems flawed as parts that appear expensive (computing pairwise mutual information between dimensions and clustering) are effectively dismissed as "pre-computed" ("mutual information and the clustering results are precomputed... thus do not introduce more cost"). By this logic some algorithms would be free. For fair comparison between algorithms, the analysis should include all aspects of the algorithm necessary given a new data set.”
>
> This misunderstanding arises because the reviewer cuts off our sentence in the middle and quotes it out of context. We explained that this pre-computation happens before training, “thus do not introduce more cost *for the model training*”, by which we do not mean it can be considered free at all. Training a model for self-supervised learning generally takes a much longer time and many iterations on the data, while this pre-computation happens once, before the training. Therefore, in comparison, the pre-computation is not an unacceptable cost, and can be considered amortized by the subsequent training time.  Additionally, it is wrong and unfair to claim that we are trying to “effectively dismiss” the point. We discuss the computational cost issue deeply with other viewers. We take this point seriously and conduct a runtime benchmarking experiment. We profiled the wall-clock running time for all modules. In addition, we analyzed the cost when scaling up the data dimension (up to 10,000 dimensions). We had an extensive conversation with reviewer t33q to come up with these experiments. These results are provided in the revised appendix (see Figure 9, Table 6 and 7), where we acknowledged it as a limitation of the current work in the last section of the paper. Specifically, we wrote “the runtime of these two procedures could be a problem for extremely high dimensional data” and provided numerical support in Figure 9. We also acknowledged all reviewers for commenting on the computational cost, which helped us give the paper a more complete presentation. We can further polish the writing in the next revision if the reviewer believes the misunderstanding still persists.
>
> >“Typos: those previously pointed out remain.”
>
> We addressed the first typo you mentioned “[031] unsupervised representation learning (UL) and self-supervised representation learning (SS) are not the same.” We removed “self-supervised representation learning” in line [031]. You could compare it with the submission history. Additionally, we add self-supervised representation learning in front of “contrastive learning and masked autoencoding” at line [35]. Please let us know if this address your concern.
> We forgot to correct the minor typo you mentioned “arised -> arisen; singal -> signal.” We have now corrected it and they will be corrected in the revision.

---

> ### Author Response · Authors · 2024-12-01
> **Follow-Up on Your Comments and Questions**
>
> Thank you again for your valuable comments!
>
> We’ve carefully considered your questions and concerns, and they were not too challenging to address. To provide clarity, we’ve included detailed background information and explanations. Please don’t hesitate to let us know if you find any gaps or have further questions—we’ll do our best to address them promptly.

---

> > ### Comment · Reviewer_u6uR · 2024-12-01
> >
> > * **Topology/Stationarity** - these terms were not well explained in the original submission or in the revision after this point was raised. Terms **in the title of the paper** should be clear from the outset and not need a third or more revision. As previously, these terms are incorrectly used and the authors should be more respectful of well-established mathematical terminology and its prior usage.
> >     - topology (of course) relates to neighbourhoods, and data is often discretised from a continuous medium (space/time) with an inherent topology under which "close" dimensions are often more correlated, as many ML models exploit (e.g. convolution, auto-regression). That is, "closeness" implies correlation, which **of course** does not mean correlation implies closeness (in general). The model here identifies correlation (/mutual information), which therefore *may* relate to topology, but equally may not. This logical relationship and the assumption of an underlying topology that induces correlation should be more rigorously considered and discussed.
> >    - stationarity does not seem the correct term and graph neural networks and so-called "geometric deep learning" has been developed around the concept of invariances in the data. The paper does not adequately discuss/fit in with prior work in this respect.
> >
> >    In summary, the paper is not clear on these fundamental terms, which is a shame as, in my view, it lets down contributions the work may otherwise bring. In this respect alone, the paper does not seem of publishable quality.
> >
> > * **Prior works** - the authors acknowledge those prior works are very related and address the same problem but are not compared to in the paper. Results of the proposed method are therefore not framed fairly relative to prior work and comparison to a $\beta$-VAE (from 2017) seems wholly insufficient. The authors now point out more related works that are not in the paper. It would not be surprising if some of these prior works materially outperform the current work. The proposed work may yet be of interest, having closer links to biological processes etc, but they need to be compared and contrasted.
> >
> >    In this respect alone, the paper does not seem ready for publication.
> >
> > * **Computational cost** - I did not "quote the paper out of context", rather pick out the words to clarify my point. The paper still says "The mutual information and the clustering results are precomputed based on the dataset and thus **do not introduce more cost for the model training**". This is a meaningless statement. The authors propose *an algorithm* to solve the task which can be viewed in 2 stages, branding one stage "the model" does not make the other less relevant or mean its cost should not be quantified. I acknowledge the authors now state this in limitations, but this raises a question over the value of the method as scalability is a fundamental aspect of machine learning up front, not an afterthought.
> >
> > * **wording** - at this point reviewers should see a (practically) final version of the paper (subject to cosmetic points). Strangely, the paper contains wording such as "we perform a scalability benchmark **as suggested by the reviewer.**", which is not camera-ready wording and the authors appear to misunderstand what should be written in the rebuttal and what should appear in the paper. This does not affect my current score (since I don't believe the paper is otherwise ready for publication) so I point this out for information as the authors should be more cognisant.
> >
> > All points here are significant. All but one were raised in my initial review and inadequately addressed (titular terms) or are fundamental to the algorithm design (cost). References [1,2] have so much in common with the proposed method (the problem addressed, architecture used), it was really for the authors to be aware of them and discuss them from the outset. The authors have since found yet more works that should have been compared to.
> >
> > I apologise once again for a delayed response to the authors' rebuttal, but I would not expect that to mean a work unready for publication is published at a top conference.

---

> ### Author Response · Authors · 2024-12-02
> **Who Neglects Classic Literature?**
>
> "**Topology/Stationarity** - these terms were not well explained in the original submission or in the revision after this point was raised. Terms in the title of the paper should be clear from the outset and not need a third or more revision. As previously, these terms are incorrectly used and the authors should be more respectful of well-established mathematical terminology and its prior usage."
>
> We disagree. The terms we used are correct and reviewer u6uR is not familiar with the classical literature and thus made such a statement. In fact, given the literature we have provided, it looks like review u6uR does not respect the well-established mathematical terminology and its prior usage. If u6uR has sufficient technical preparation and believes our previously provided response is technically wrong, please check and provide a more in-depth discussion and stop using high-level and vague terms.
>
>
> **The reviewer u6uR’s bias and insufficient technical preparation already made the discussion irritating for the authors. But to be professional, let us provide you with two more references to support the correctness of the terms we are using: the first is a classical reference in natural signal processing [1] and the second one is AlexNet [2]. One can find the usage of stationarity in the Introduction of AlexNet.**
>
> To make it convenient for the Review u6uR, we will quote the usage from both references here:
>
> In the highly cited review of natural image statistics [1]:
> > [...], it is commonly assumed that the statistical properties of the image are translation invariant (also known as **stationary**).
>
> In the Introduction of AlexNet[2]:
> >(CNN) … Their capacity can be controlled by varying their depth and breadth, and they also make strong and mostly correct assumptions about the nature of images (namely, **stationarity** of statistics and locality of pixel dependencies).
>
> [1] Eero P Simoncelli and Bruno A Olshausen,  Nature Image statistics and neural representation, 2001
>
> [2] Alex Krizhevsky, Ilya Sutskever, and Geoffrey E. Hinton, ImageNet Classification with Deep Convolutional Neural Networks, 2012
>
>
> Let us make this statement again: while we can maintain different opinions on what terms to use, it would be strange to dismiss a paper on a minor vocabulary disagreement. Why does the reviewer u6uR have to be this aggressive about this vocabulary disagreement?

---

> > ### Comment · Reviewer_u6uR · 2024-12-02
> >
> > The purpose of the review process is to clarify misunderstanding/misinterpretation, which is expected to occur, and should obviously be polite and respectful. The authors have chosen otherwise [1] and I will engage no further with them. The point made above should be clear **in the paper**. In future, the authors would be well advised to make rebuttal arguments far more *concise* and, needless to say, respectful.
> >
> > [1] "The reviewer u6uR’s bias and insufficient technical preparation already made the discussion irritating for the authors. But to be professional ..."

---

> > > ### Comment · Reviewer_t33q · 2024-12-02
> > >
> > > I would highly agree with the reviewer u6uR, that I would highly prefer more concise and respectful rebuttal.

---

> ### Author Response · Authors · 2024-12-02
> **Agreed**
>
> Dear Reviewer t33q,
>
> We fully agree with you that the whole process shall be respectful .. As the technical argument became repetitive and the reviewer insists to ignore the literature support we provided, it is definitely challenging for everyone.
>
> Getting into such a technical argument was not our intention or expected .. however, we maintain our technical disagreement with Reviewer u6uR.
>
> We apologize for the extended discussion; in fact, we dedicated our entire Thanksgiving holiday to addressing this. We kindly ask for your patience as we work through the final few points.
>
> URLOST authors

---

> ### Author Response · Authors · 2024-12-02
> **Author Follow-up Response to Reviewer u6uR - Part 1/5**
>
> >“topology (of course) relates to, and data is often discretised from a continuous medium (space/time) with an inherent topology under which "close" dimensions are often more correlated, as many ML models exploit (e.g. convolution, auto-regression). That is, "closeness" implies correlation, which **of course** does not mean correlation implies closeness (in general). The model here identifies correlation (/mutual information), which therefore may relate to topology, but equally may not. ”
>
> This statement from Reviewer u6uR has several technical flaws. First, the “closeness” of the signal domain does not necessarily imply correlation. For example, given signals formed by IID random variables, the “closeness” in the domain of the signal does not imply correlation. Second, mutual information is replaceable by correlation. Correlation is only second-order statistics. Independence implies zero correlation and zero correlation doesn’t imply independence in general. For Gaussian random variables, zero correlation implies independence. So, correlation is a weaker measurement than mutual information. ML models in general leverage higher-order statistics from the signal way beyond second-order correlation. In general, neither the “closeness” of the signal domain implies higher value dependence nor a high-value dependence implies the “closeness” in the signal domain is true. However, for natural signals, these two things are strongly coupled with each other as have been studied by many classical literature in natural signal statistics. If the reviewer u6uR wishes to make a strong exception, please provide a representative example to stimulate further discussion. But please be careful about the difference between correlation and dependence. One can whiten a signal with the second-order statistics and yet all of the high-order statistics remain, and this was the classical idea in independent component analysis.
>
> One more technical note, the point of this work is not to fully recover the signal domain but rather to build a representation for the signal. Why shall we process dimensions with stronger dependence? That's because the stronger the dependence is, the more structure can be leveraged or a stronger compressibility there is. If random variables are independent, there are no structures to be learned between them!
>
> During our previous reply, we decided to provide a deeper discussion about stationarity and topology to Reviewer u6uR, but the reviewer u6uR failed to engage with these more technical statements. This might be due to the fact that the reviewer u6uR does not have sufficient technical preparation related to these subjects. However, in our opinion, it is unreasonable and unnecessary to require the generic audience to engage these subjects in such a rigorous way. Though we will provide these rigorous details in the appendix, one has to realize that being too technical may equally reject the audience. We still aim to maintain the intuitiveness.

---

> ### Author Response · Authors · 2024-12-02
> **Author Follow-up Response to Reviewer u6uR - Part 2/5**
>
> > “The assumption of an underlying topology that induces correlation should be more rigorously considered and discussed.”
>
> This is a well-studied problem in many domains like images, audio, and television signal, see [1,3,4,5,6,7]. Our result shows it could be generalized to other domains like neural recording in visual systems and genes.
>
> [1] Eero P Simoncelli and Bruno A Olshausen, Nature Image statistics and neural representation, 2001
>
> [3] Hagai Attias, Attias Temporal Low-Order Statistics of Natural Sounds, 1996
>
> [4] Nicolas Roux, et al. Learning the 2-D Topology of Images, 2007
>
> [5] E.R. Kretzmer, Statistics of Television Signals, 1952.
>
> [6] Daniel L. Ruderman, Origins of scaling in natural images, 1997
>
> [7] Daniel Zoran, Natural Image Statistics for Human and Computer Vision, 2013

---

> ### Author Response · Authors · 2024-12-02
> **Author Follow-up Response to Reviewer u6uR - Part 3/5**
>
> > “stationarity does not seem the correct term and graph neural networks and so-called "geometric deep learning" has been developed around the concept of invariances in the data. The paper does not adequately discuss/fit in with prior work in this respect.”
>
> We disagree.
>
> Again, as we have provided the reviewer u6uR with sufficient literature.  The reviewer u6uR failed to engage with these more technical statements. However, in our opinion, it is unreasonable to expect the generic audience will engage these concepts in such a rigorous way.
>
> First, the terms we used are not wrong and we have provided sufficient literature support. They were well developed and rigorous. All of these literatures predate geometric deep learning. However, we have also adequately cited literature from geometric deep learning.
>
> Using invariance here is not wrong either. As we said, this is a preference and there is no right or wrong in this context. If the reviewer is willing to penetrate the classical literature, we believe this disagreement will disappear. The idea of invariance and stationery is used interchangeably in many fields like signal processing, machine learning, computational neuroscience and computer vision literature [1][2]. In order to be more inclusive and satisfy reviewer u6uR, we will provide additional explanation in the next revision. As we put it earlier, we can maintain different opinions on what terms to be used.
>
> [1] Eero P Simoncelli and Bruno A Olshausen,  Nature Image statistics and neural representation, 2001
>
> [2] Alex Krizhevsky, Ilya Sutskever, and Geoffrey E. Hinton, ImageNet Classification with Deep Convolutional Neural Networks, 2012

---

> ### Author Response · Authors · 2024-12-02
> **Author Follow-up Response to Reviewer u6uR - Part 4/5**
>
> > “Prior works - the authors acknowledge those prior works are very related and address the same problem but are not compared to in the paper. Results of the proposed method are therefore not framed fairly relative to prior work and comparison to a β-VAE (from 2017) seems wholly insufficient. The authors now point out more related works that are not in the paper. It would not be surprising if some of these prior works materially outperform the current work. The proposed work may yet be of interest, having closer links to biological processes etc, but they need to be compared and contrasted.”
>
> Reviewer suggested [1][2] right after the PDF revision portal had closed, which rules out the possibility to put these references into the revision. While we acknowledge the connection between URLOST and [1, 2], the development of URLOST does not rely on any of these works.  The development of URLOST is motivated entirely from SOM and the more classical literatures, and it is explicitly modeling the dependency between different dimensions. The idea we sufficiently motivated and the empirical results we provide is able to support the claimed advancements. Based on our interpretation, [1] and [2] can be treated as implicit SOM methods, though such a connection is proposed by us rather than the original authors of these works.
>
> **The result in [2] is not comparable to our model. The evaluation used in [2] is fine-tuning but not linear probing.** This is not starkly different from how a self-supervised learning model or unsupervised learning model should be evaluated. Further, the results from [2] are almost exclusively fine-tuning results, which is not sufficient to show the quality of self-supervised representation. Following the evaluation standard in SSL [3-7], one should either show the linear-probing results or the nearest neighbor classification results. When we tried to compare our results with [2], we found that the evidence shown from [2] is insufficient to support its self-supervised representation. As we dig into the paper [2] and its open-sourced code, we can not find either linear probing results or nearest neighbor results. ~~If this is true~~ **We have confirmed that [2] does not have linear probing or KNN evaluation.** Since [2] focuses on fine-tuning, it is almost impossible to evaluate the representation quality by SSL convention [3-7]. ~~While we can put in further effort to help the authors in [2] debug their work, it is really not our responsibility to do so.~~ Additionally, although the reviewer frames the narrative as “beta-vae is a weak model from 2017”, we want to remind the reviewer that beta-vae is considered the state of the art unsupervised learning model for the particular dataset we used [8,9]. We cited the more recent reference in our manuscript.
>
> Moreover, our work is fundamentally different from [2]. It is similar that both work leverage masked autoencoders. However, the key contribution of both papers is how to process the signals for learning useful representations. The way we proposed to process the signal is significantly different from [2]. In [2], similar tokens in a sequence are processed together. However, the similarity is purely defined by the value of each token in a particular sequence, which does not depend on the statistic of each token. In the example of natural image data, this means pixels of the same color will be aggregated together. Each token in the latent array will attend a different set of pixels in a different image. Thus, the way pixels are clustered does not depend on the underlying topology of natural images. On the other hand, in URLOST, instead of computing similarity between pixels on a single image, we collect many images and compute the joint statistics between pairs of pixels. In other words, URLOST explores the joint statistics of pixels at different locations while [2] explores the marginal distribution of pixels. This distinction is crucial and allows our model to effectively recover the underlying topology of natural images, but not [2].
>
> [1] Andrew Jaegle et al. Perceiver: General Perception with Iterative Attention
>
> [2] Johnathan Xie et al. Self-Guided Masked Autoencoders for Domain-Agnostic Self-Supervised Learning
>
> [3] Chen et al., A Simple Framework for Contrastive Learning of Visual Representations, 2020
>
> [4] He et al., Momentum Contrast for Unsupervised Visual Representation Learning, 2019
>
> [5] Wu et al., Unsupervised feature learning via non-parametric instance discrimination, 2018
>
> [6] Zbontar et al., Barlow Twins: Self-Supervised Learning via Redundancy Reduction
>
> [7] Bardes et al., VICReg: Variance-Invariance-Covariance Regularization for Self-Supervised Learning
>
> [8] Zhang, Xiaoyu, OmiEmbed: A Unified Multi-Task Deep Learning Framework for Multi-Omics Data, 2021
>
> [9] Zhang, Xiaoyu, Integrated Multi-omics Analysis Using Variational Autoencoders: Application to Pan-cancer Classification, 2019.

---

> ### Author Response · Authors · 2024-12-02
> **Author Follow-up Response to Reviewer u6uR - Part 5/5**
>
> > "Computational cost - I did not "quote the paper out of context", rather pick out the words to clarify my point. The paper still says "The mutual information and the clustering results are precomputed based on the dataset and thus do not introduce more cost for the model training". This is a meaningless statement. The authors propose an algorithm to solve the task which can be viewed in 2 stages, branding one stage "the model" does not make the other less relevant or mean its cost should not be quantified. I acknowledge the authors now state this in limitations, but this raises a question over the value of the method as scalability is a fundamental aspect of machine learning up front, not an afterthought."
>
> It is an inflated criticism and a misinterpretation to say that we think the precomputation is irrelevant and should not be quantized.
>
> Firstly, in the design of our method from the outset, the computation cost of the first stage of the model (computed once on the dataset) is simply outweighed by the second stage, which involves optimizing a deep neural network over many iterations on the dataset.
>
> Secondly, to justify our design, we provide quantification to the computation cost in both stages. For the first stage, we benchmark the runtime by increasing the dimensions of the input signal to over 10,000 dimensions in Figure 9. We also provide discussions in the appendix and in the limitation and conclusion section, where we further suggest a GPU implementation to resolve the computation concern when it comes to extremely high-dimensional data.  For the second stage (the model training), as a direct comparison to training the original MAE, we provide the ratio of added parameters to the MAE, and the runtime for each of the modules, respectively in Table 6 and 7. These are clear demonstrations of our efforts in quantifying the computational cost and addressing the reviewers’ comments on this matter. We don’t understand why the reviewer u6uR frames it like we deliberately hide a computational issue. In reality, we actively try to make our work more complete by thinking about the extreme case. “Do not introduce more cost for the model training” is not a meaningless statement but a factual one, given the context that the comparison is made with training the original MAE, which is stated in the paragraph. The runtime of forward operation is also presented in Table 7 to quantize the computation cost.
>
>
> >**wording** - at this point reviewers should see a (practically) final version of the paper (subject to cosmetic points). Strangely, the paper contains wording such as "we perform a scalability benchmark as suggested by the reviewer.", which is not camera-ready wording and the authors appear to misunderstand what should be written in the rebuttal and what should appear in the paper. This does not affect my current score (since I don't believe the paper is otherwise ready for publication) so I point this out for information as the authors should be more cognisant.
>
> This statement is a bit farfetched as we believe these are academic courtesy during the rebuttal period and are a miner and easily addressable issue. We reassure it is not intended for the final camera-ready version.  Yet it might be unconstructive and unnecessary to claim this as a significant point.

---

> ### Author Response · Authors · 2024-12-04
> **Final Remark on the Technical Debate with Reviewer u6uR**
>
> It is unfortunate that we got into this extended technical debate. All of the debate history will be open and we welcome the community for further evaluation. Technically, we maintain our stand on several disagreements with Reviewer u6uR.
>
> Reviewer u6uR responded right after the revision deadline. During the extended debate, Reviewer u6uR started using ‘**of course**’, ‘meaningless’, ‘which is a shame’, and insists on ignoring the solid literature support we provided, and we felt this debate had started to be irritating. And at this point, both parties lost respect to each other.
>
> Upon reflection, we suspect that the tension might be due to the fact that we were too eager to start the discussion with Reviewer u6uR and sent two reminders before the PDF revision deadline. We sincerely apologize that these reminders might give Reviewer u6uR unnecessary pressure during the rebuttal and after all, we really appreciate your voluntary effort to evaluate our work.

---

### Official Review · Reviewer_o7jg · 2024-11-01

**Soundness:** 3
**Presentation:** 3
**Contribution:** 4
**Rating:** 8
**Confidence:** 4

**Summary:**

The submission proposes a route to implementing masked autoencoders (MAE) when the data does not have a known structure and yet is too granular for each component of the input to be its own “patch”.  In other words, there’s prior knowledge about the dataset that its components could probably be meaningfully grouped, but no grouping is available.
The method clusters the components by mutual information and spectral clustering, with an additional density correction applied.  After components have been grouped, all groups must be mapped to a shared representation space so that the transformer can compare them; for this the authors use a learned projection for each group that they call a self-organizing layer.
There are a fairly comprehensive suite of comparisons: to MAE without grouping the components and to SimCLR with a CNN and ViG architecture, on a couple CIFAR10 variants, gene expression data, and on V1 neural recording data.

**Strengths:**

The premise to handle more diverse datasets than images with a ViT setup (and MAE pretraining, though it seems more general than a specific pretraining method) by clustering input components is strong.  The experiments cover a good range of complexity, starting with standard CIFAR10, permuting the pixels, and then remapping the visual field entirely.  Then the V1 and gene expression experiments represent actual use cases.  The comparisons are sufficient to demonstrate the utility of the method.

**Weaknesses:**

The clustering is the heart of the proposed method and comes with multiple design choices or parameters whose selection is not obvious.  While the experimental results demonstrate the method works, results supporting these design decisions are a bit weak.  As far as I can tell, the only results regarding the effect of the density adjusted spectral clustering are with respect to the foveated CIFAR10 data (Table 3b, Table 4)?  Why should we assume gene expression is best processed with the same parameters, considering how structured the synthesized foveated CIFAR10 is?  By contrast, the self-organizing layer is pretty intuitive, yet a large chunk of the manuscript and appendices are dedicated to it.

Related work could be strengthened.  Consider discussing the relation to FlexiViT (Beyer et al., CVPR 2023), where different patch sizes are mapped to the same representation space by learnable transformations, and PatchGT (Gao et al., LoG 2022), where nodes of a graph are clustered with spectral clustering, and the representation of each “patch” is a learned function of the subgraph.

**Questions:**

- Why is SimCLR with the proposed clustering pipeline (and ViT architecture) not included as a baseline for the CIFAR10-based experiments?  Why not ViG with patches found by URLOST?  Unless I have misunderstood, the proposed method is ultimately about “patchifying” data for use with a downstream transformer, which means there’s nothing tying the paper to MAE.  Such a shift away from exclusively linking the method with MAE would broaden its potential impact.
- There is no need for explicit positional encoding information per cluster, correct?  Presumably the self-organizing layers can encode identifying information per cluster.
- Line 265: “Moreover, since CNN can only process signal with stationarity and topology” is an incomplete sentence and also incorrect, as CNNs can absolutely process signals without stationarity.  Overall, I felt the stationarity part of the motivation to be much less important than the topology.
- Why is MAE bolded in Table 1 when it was outperformed by SimCLR?

---

> ### Author Response · Authors · 2024-11-23
> **Author Response to Reviewer o7jg (1/6)**
>
> Dear reviewer o7jg,
>
> We sincerely thank you for your detailed and thoughtful review, and your constructive questions and comments. Your recognition, encouragement, and clear understanding of our motivation and work are truly appreciated.
>
> We are particularly grateful for the time and effort you dedicated to providing such an in-depth review. The level of detail and insight in your comments clearly indicates that you took significant time to carefully analyze and understand our work. This level of dedication is rare nowadays and is something we deeply value.
>
> Based on your constructive suggestions, we have incorporated several updates, such as integrating your observation about how the self-organizing map could learn positional encoding, conducting additional experiments, including more related works, and discussing future directions you suggested.
>
> Here’s a list of change we made to the revised manuscript according to your suggestion:
>
> Added the intuition on how to select clustering hyperparameters in experiments (line 208) and Appendix A.6
> Included the finding about positional encoding and self-organizing layer.
> Include discussion of FlexiViT, Patch VIT in related work.
> Include potential to extend URLOST to contrastive learning methods like SimCLR in future direction.
> Fixed typo at Line 265.
> Fixed bolding typo in Table 1.
> Including additional experiments for grid-searching the hyperparameter for clustering number and density factors in Appendix A.7
>
> Following are answers to each of your questions:

---

> ### Author Response · Authors · 2024-11-23
> **Author Response to Reviewer o7jg (2/6)**
>
> __Q1__: “The clustering is the heart of the proposed method and comes with multiple design choices or parameters whose selection is not obvious.”
>
> __A__: Indeed spectral clustering or clustering algorithms in general have a number of hyperparameters. Parameters like “density factors” and “number of clusters” affect the shape and the size of the clusters, thus, the performance of the model to some degree. In general, the model is not very sensitive to these hyperparameters. To apply URLOST, one only needs to make sure these hyperparameters are not too extreme. We added the intuition on how to select them in the manuscript. We also add additional experiments to show the effect of these parameters to the model's performance.
>
> **Table 1: \(n\) vs Accuracy**
>
>
> | hyperparameter = \(n\) | 10    | 20    | 30    | 40    | 50    | 60    | 70    | 80    | 90    | 100   |
> |-------------------------|-------|-------|-------|-------|-------|-------|-------|-------|-------|-------|
> | Accuracy (\%)          | 90.5  | 94.5  | 95.2  | 94.6  | 95.2  | 94.5  | 93.6  | 94.7  | 93.9  | 92.9  |
>
>
> ---
>
>
> **Table 2: \(K\) vs Accuracy**
>
>
> | hyperparameter = \(K\) | 4     | 8     | 12    | 16    | 20    | 24    | 28    |
> |-------------------------|-------|-------|-------|-------|-------|-------|-------|
> | Accuracy (\%)          | 95.2  | 95.4  | 95.2  | 95.4  | 95.4  | 95.1  | 95.3  |
>
>
> ---
>
>
> **Table 3: \($\beta\$) vs Accuracy**
>
>
> | hyperparameter = \($\beta$\) | 0.5   | 0.8   | 0.9   | 0.95  | 1     | 1.05  | 1.1   | 1.2   | 1.5   | 2     |
> |----------------------------|-------|-------|-------|-------|-------|-------|-------|-------|-------|-------|
> | Accuracy (\%)             | 85.3  | 93.7  | 94.8  | 95.3  | 94.7  | 94.9  | 95.1  | 95.3  | 94.3  | 93.5  |
>
>
> Intuition for the parameter selection: Both these hyperparameters are related to the size of resulting clusters, which relates to how difficult the task is. Taking the image as an example, if the cluster is too big, then the model needs to learn to predict a big missing area in the image, making the self-supervised learning task too difficult. The model will struggle to learn any useful representation. On the other hand, if the cluster Is too small, for example, if the cluster shrinks down to one pixel, then the prediction tasks become too easy. The model can learn to fill in the missing pixel by averaging the value of nearby pixels. The model essentially learns to become a low-pass filtering for images. In both cases, the model will not learn high quality representations. Same intuition applies to other high dimensional signals. We recovered the topology by clustering based on the pairwise mutual information between dimensions of the signals.  Both the number of clusters and density factors are related to the cluster size. Therefore, to make the self-supervised learning task effective, we set the hyperparameters in order to make sure the size of the cluster is not too big or too small.
>
> Increasing the number of clusters could make each cluster smaller. The density factor controls the variance of the cluster sizes by applying proper normalization to the graph during spectral clustering to make the cluster size more evenly distributed. For an image dataset, clusters should contain semantic sub-structures of the image, like body parts or parts of objects. We pick 64 clusters, which results in clusters of size roughly 4x4. In other words, the compression ratio between the clusters (patches) and the raw signals (image) is roughly 1:64. For other dataset (gene and V1), we perform a grid search around this ratio. Similarly, for the density factors, we also perform a grid search centered at the density factor equal to 1, which is the default normalization used in spectral clustering.

---

> ### Author Response · Authors · 2024-11-23
> **Author Response to Reviewer o7jg (3/6)**
>
> __Q2__: “There is no need for explicit positional encoding information per cluster, correct? Presumably the self-organizing layers can encode identifying information per cluster.”
>
> __A2__: This is an incredibly interesting observation. In fact, the self-organizing layers could be simultaneously learning a form of positional encoding. Standard positional encoding is added onto each token, whereas self-organizing layers transform each token. Let $g^{(t)}$ denote the self-organizing layer at position $t$, which projects token $x^{(t)}$ to the embedding space. If $g(x^{(t)}) = x^{(t)} + b^{(t)}$, then $g$ is exactly positional encoding. In our implementation, we use a linear layer to parametrize the self-organizing layer, i.e. $g(x^{(t)}) = W^{(t)} x^{(t)} + b^{(t)}$. Using this parametrization, we expect $W^{(t)}$ to learn to align all the tokens, while $b^{(t)}$ will serve as positional encoding to learn the positional information for each token.
>
> In figure 4 and line 391 in text, we visualize what self-organizing layers learnt in a designed experiment. We create the data by permuting each patch individually. We would expect the self-organizing layer to learn to “undo” the permutation, in order to align these patches. However, what we actually observe is that the self-organizing layer learns to “undo” the permutation for the most part, but also does some extra transformation to the patches. As shown in the Figure. 4, these transformations are very regular and look similar to geometric transformations like rotation and flip. These transformations likely encode positional information as transformation (group actions) of some simple group structures. Although we hope all the positional information to be encoded in the additive part $b^{(t)}$, some positional encoding is also encoded by $W^{(t)}$.
>
> Why the self-organizing layer learns to add these extra transformation puzzles us for a long time. The reviewer’s comments provide valuable inspiration. We want to acknowledge the reviewer and update the manuscript and include this interesting finding, we will also explore in this direction in our future work.
>
> __Q3__: “As far as I can tell, the only results regarding the effect of the density adjusted spectral clustering are with respect to the foveated CIFAR10 data (Table 3b, Table 4)?”
>
> __A__: For the cifar10 experiments, we only apply density adjustment for foveated cifar10 because only the foveated version is non-stationary. Other than the CIFAR10 experiment, density adjusted spectral clustering is also effective for gene expression. The parameters are listed in the appendix A.6.
>
> __Q4__: “Why should we assume gene expression is best processed with the same parameters, considering how structured the synthesized foveated CIFAR10 is?”
>
> __A__: Your observation is correct. Although we show that the same method is effective across the three diversed data modalities, the hyperparameters of the clustering are not kept the same given how different the datasets are.  The parameters are listed in the appendix A.6.. Since the hyperparameter choices are mentioned by more than one reviewer, we have added additional experiments to show that our method is not constraint to specifically selected hyperparameters. Please refers to the additional experiment in the general reply, and Appendix A.7.

---

> ### Author Response · Authors · 2024-11-23
> **Author Response to Reviewer o7jg (4/6)**
>
> __Q5__: “The self-organizing layer is pretty intuitive, yet a large chunk of the manuscript and appendices are dedicated to it.”
>
> __A__: Thank you for the question and thank you for thinking the design intuitive. In fact, we find it is an advantageof our method that it is conceptually intuitive and empirically effective. Furthermore, in contrast to the deep neural networks, this part of the model enjoys more interpretability and can be visualized to validate our hypothesis, which we find very cool and worthy of showing in the paper.  Since it plays an important role in out method, we also find it necessary to add an ablation study for it. Although it extends the length dedicated to it, we believe the visualization, explanation and the ablation studies will make a clear and rigorous presentation of our work to a wider audience.
>
> __Q6__: “Related work could be strengthened. Consider discussing the relation to FlexiViT (Beyer et al., CVPR 2023), where different patch sizes are mapped to the same representation space by learnable transformations, and PatchGT (Gao et al., LoG 2022)”
>
> __A__: Thank you for pointing this out! After taking a look, we think these two works are very relevant. Following your suggestion, we include these two works and incorporate the following comments in the related work section:
> In particular, the self-organizing layer used in this work is a generalization of the resized operator used in FlexiViT. We also notice that the parameters of the “patch resizer” in FlexiViT are solved via a local objective. In contrast, the parameters of the self-organizing layer are optimized with all other parameters of the network for an end-to-end objective. The method used in FlexiViT imposed more constraints, which has its advantages and could be very effective. Patch VIT is also very related. Unlike in URLOST, where we use a single mutual information graph to define the topology of the high dimensional signal, PatchGT is designed to handle graph data, where each data point is a different graph. It segments a graph into “subgraph” and feeds the subgraph into a GNN, then a transformer. This design choice is definitely more flexible. However, since GNN must align all subgraphs globally, it might impede scalability. It would be an interesting idea to combine Patch VIT and URLOST by replacing the self-organizing layer with GNN.
>
> __Q7__: “Why is SimCLR with the proposed clustering pipeline (and ViT architecture) not included as a baseline for the CIFAR10-based experiments? Why not ViG with patches found by URLOST?”
>
> __A__: These are two great ideas and we thank the reviewer for their active thinking and construction comments. We also thought about using SimCLR for the self-supervised  learning task instead of mask prediction. In short, we choose to use reconstruction-based self-supervised learning methods because they are the most general and flexible for generic data modalities. Among all reconstruction based self-supervised learning tasks, the masked-prediction task has shown to be the most effective one [1,2,3,4,5]. Data augmentation, especially the random-resized-crop augmentation is the essence that makes SimCLR performant.. Therefore, in order to generalize SimCLR to the URLOST setting, we need to define a way to generate “two crops” of the signals. Like you mentioned, we could potentially modify the clustering pipeline to do this. Instead of clusters, we could use spectral embedding to do 2-cut on the graph. This could be an interesting direction to explore and potentially go beyond.  One could use it as a generic graph data augmentation technique. Given the academic merit and the non-trival effort this adoption possibly involves, we think this would count more than just as a baseline with in this work, but rather another version of URLOST or a novel data-augmentation technique for graphs.
> Using patches found by URLOST for VIG is another great idea to try. The masked prediction objective used in URLOST doesn’t technically depend on the neural architecture, but  Transformer is the reasonable choice of architecture for this task given the fact that Transformer, rather than a graph neural network, has shown to be effective in a lot of research in representation learning as well as downstream tasks. This gives us a solid reason for choosing to experiment URLOST with Transformer as our backbone. One could indeed adapt ViG as the backbone for masked-prediction tasks. We use transformer instead of ViG because transformers are more popular and shown scalable, whereas ViG is more niche.

---

> ### Author Response · Authors · 2024-11-23
> **Author Response to Reviewer o7jg (5/6)**
>
> __Q8__: “Unless I have misunderstood, the proposed method is ultimately about “patchifying” data for use with a downstream transformer, which means there’s nothing tying the paper to MAE.”
>
> __A__: That’s correct. This framework is not tied to MAE. The purpose of this work is to propose a method for representation learning when prior knowledge of the data structure is extremely limited. And a learned representation is meaningful if it benefits downstream tasks when labels/supervisions are available.  So one could replace the training objective directly with classification or other downstream tasks when supervision is available. We want to use MAE because we want to build an unsupervised representation for general high-dimension data, such as chemistry, biology, and neuroscience. In many of these domains, labels or downstream tasks are not available for pre-training the model. In our method, once “patchifying” data, mask prediction becomes a natural choice of self-supervised learning objective and that is why we adopted MAE. Like what the reviewer said, the framework consists of “patchify,” neural architecture (transformer) and task (mask-prediction). Although both task and neural architecture could both be replaced, this work shows that the combo of the three elements provide a great way for unsupervised representation learning.
>
> __Q9__: “Line 265 [...] is an incomplete sentence and also incorrect, as CNNs can absolutely process signals without stationarity.”
>
> __A__: Thanks for pointing out the typo. We revise this sentence as the following and will update it in the manuscript:
>
> “Moreover, since CNN works better with stationary signals, we further compared our methods to SimCLR with a graph neural network (GNN) backbone, which doesn’t rely on the stationarity assumption of the data.”
>
> In the original sentence, we missed a comma, making the sentence incomplete. CNNs could operate on signals with topology but without stationarity. However, its performance will significantly drop if the signal is non-stationary. This is due to the nature of the convolutional operations. An example will be a foveated image via a multi-resolution pyramid. In this case, the 2d topology of the image is preserved, but the center has higher resolution than the surroundings. Each kernel in the convolutional neural network has fixed resolution but is applied to all regions. Meanwhile, the information granularity is different from the center to the surroundings because of the foveation, therefore, the same kernel filtering the same piece of information at the center and at the surroundings will produce different outcomes, causing the performance to degrade.
>
> __Q10__: “Why is MAE bolded in Table 1 when it was outperformed by SimCLR?”
>
> __A__: Thanks for pointing this out. The original intention of bolding in Table 1 is to highlight the patch based MAE methods across different dataset settings. And show their performances persist as we remove topology and stationarity but apply URLOST. However, we realized this bolding style is confused and also contradicts with the bolding used in table 2. Thus, we modify it according to the reviewer.

---

> ### Author Response · Authors · 2024-11-23
> **Author Response to Reviewer o7jg (6/6)**
>
> [1] Burgess, Christopher P., et al. 2018. "Understanding disentangling in $\beta $-VAE."
>
> [2] Balestriero, LeCun. 2024. Learning by Reconstruction Produces Uninformative Features For Perception
>
> [3] Bao, Hangbo, at el. 2022. BEiT: BERT Pre-Training of Image Transformers
>
> [4] He, Kaiming, et al. 2021. Masked Autoencoders Are Scalable Vision Learners
>
> [5] Oord, Aaron van den, 2018. Neural Discrete Representation Learning

---

> > ### Comment · Reviewer_o7jg · 2024-11-25
> > **Thorough response**
> >
> > I have read the authors' thorough responses to my review as well as the others.  I think I am on the same page, for the most part, as reviewer sHq9, though I view the contribution of the submission more positively.  Reading over the more negative reviews, I disagree with much of Reviewer 3Rzb's assessment, and think the perceived weaknesses are either off-base or inflated.  t33q brings up some good points, though not enough to change my mind at present.
> >
> > I stand by my original rating of 8, and have increased my confidence to 4.

---

> > > ### Author Response · Authors · 2024-11-27
> > >
> > > Thank you for your positive feedback and increasing confidence score! We greatly appreciate the time and effort you spent for reviewing our work. We also appreciate that you even spent additional time to read other reviewer's comments. Your comments are very valuable and it truly improved the quality of our manuscript!

---

### Official Review · Reviewer_3Rzb · 2024-11-03

**Soundness:** 2
**Presentation:** 2
**Contribution:** 2
**Rating:** 5
**Confidence:** 3

**Summary:**

This work introduces URLOST, a new framework for unsupervised representation learning aiming to overcomes limitations in handling high-dimensional data with unknown stationarity and topology, towards going beyond traditional methods that rely on structured data assumptions, such as grid-like images or time sequences. The proposed method combines a learnable self-organizing layer, density-adjusted spectral clustering, and a masked autoencoder (MAE) to capture learning representations from various data modalities.

Demonstrated on synthetic biological vision data, neural recordings, and gene expressions, URLOST shows a good performance, outperforming existing baselines in capturing complex, irregular structures without prior domain knowledge. The paper highlights areas for future work, such as integrating clustering into the model's end-to-end learning process and enhancing the self-organizing layer.

**Strengths:**

The strengths of the paper lie in its introduction of URLOST, a new combined framework for unsupervised representation learning towadrs addressing the challenges of high-dimensional data with unknown stationarity and topology. By combining a learnable self-organizing layer, density-adjusted spectral clustering, and a masked autoencoder, URLOST can captures learning representations from various data modalities. Its demonstrated performance on synthetic biological vision data, neural recordings, and gene expressions shows that it outperforms baselines, enlightening its capability to handle complex and irregular structures in an unsupervised mode.

The work is nicely conceived and the presentation is tidy. Its bio-inspired sense makes the work basically intriguing.

**Weaknesses:**

Several aspects of the work can be improved:
1) While the paper primarily motivates unsupervised learning, the experiments predominantly focus on supervised tasks (classification). This disconnect diminishes the relevance of the initial claims made in the first half of the paper.
2) The use of foveated preprocessing is an intriguing aspect; however, there is a noticeable drop in performance (in Table 1), likely attributed to this process. This raises the question of whether foveated preprocessing detracts from overall performance.
3)  Although the MAE appears to perform worse on the foveated dataset, this could simply be due to its training on the original dataset. Thus, the comparison does not effectively support the goal of emulating the biological process of foveation for benefits.
4) While the Vision Transformer (ViT) serves as the backbone, the paper lacks clarity on how attention is utilized in the experiments. It would be beneficial for the community to understand the potential relationship between the foveated process and visual attention.
5) The comparison against the MAE seems somewhat rudimentary. For example, the reported performance on CIFAR-10 indicates that several top models significantly outperform the authors' results (Table 1):
	Rank	Model		Percentage correct
	1		ViT-H/14		99.5
	2		DINOv2 		99.5
	3		µ2Net		99.49
	4		ViT-L/16		99.42
It seems the claimed better performance over the baseline was quite embarrassed by the top runners above.
6) The meaning and algorithm behind the permutation of CIFAR-10 images are unclear. Providing examples of permuted images would clarify this aspect. The rationale for permuting pixels should also be addressed, as such transformations may hinder human recognition and contradict the stated bio-inspired motivation.
7) In Section 2, the proposed method appears to aggregate several state-of-the-art techniques with limited novel mathematical contributions. Incorporating theoretical exploration of its bio-inspired aspects would significantly enhance the paper's quality.

**Questions:**

No extra questions. See the above comments.

---

> ### Author Response · Authors · 2024-11-14
> **Author Response to Reviewer 3Rzb (1/2)**
>
> Dear reviewer 3Rzb, thank you for putting effort into reviewing our paper. However, we believe the weakness you pointed out is trivial. Some of these weaknesses are even collectively recognized as strengths by other reviewers. We will address the misunderstanding around these weaknesses. We kindly ask the reviewer to take a second look at their comments. We would also love to invite the reviewer to take a second look at the paper and provide us some constructive feedback when the misunderstanding is cleared.
>
> Addressing weaknesses:
>
> **Q1**: “While the paper primarily motivates unsupervised learning, the experiments predominantly focus on supervised tasks (classification). This disconnect diminishes the relevance of the initial claims made in the first half of the paper.”
>
> **A1**: URLOST is completely unsupervised. After we trained the model in an unsupervised setting, we tested if the representation learned meaningful representation by linear probing. Although this evaluation process is a supervised task, the unsupervised pre-trained URLOST model is frozen. This is a widely adopted and well-known evaluation technique in unsupervised learning and self-supervised learning. We explained this evaluation process in line 268. “After the model is trained without supervision, we use linear probing to evaluate it. This is achieved by training a linear classifier on top of the pre-trained model with the given labels.” Both Reviewer sHq9 and Reviewer o7jg think our experimental evidence is strong.
>
> **Q2**: “The use of foveated preprocessing is an intriguing aspect; however, there is a noticeable drop in performance (in Table 1), likely attributed to this process. This raises the question of whether foveated preprocessing detracts from overall performance.”
>
> **A2**: There seems to be a misunderstanding here. The foveation process will surely lower the performance due to the loss of information in the surrounding part of the images. However, the point of introducing foveation is not to increase the overall performance, but to synthesize a plausible difficult data modality without topology or stationarity. This is outlined in lines 102-103 in the manuscript. We show that URLOST can learn meaningful unsupervised representation from this data modality, but standard SSL methods cannot.
>
> **Q3**: “Although the MAE appears to perform worse on the foveated dataset, this could simply be due to its training on the original dataset. Thus, the comparison does not effectively support the goal of emulating the biological process of foveation for benefits.”
>
> **A3**: The first part of this statement is incorrect. We did not train the model on the original CIFAR10 then tested it on the foveated CIFAR10. We pre-trained and tested the model on the same dataset, in this case, foveated CIFAR10. For the second part, this is a misinterpretation. The goal of simulating foveation is not to show foveation is beneficial for image classification. In the visual system, foveation is only useful when we aggregate many samples, by combining with fast fixational eye movements (constantly looking around unconsciously). The foveated visual signal is a difficult signal to process. We simulate this signal to prove it can be handled by our model but not standard computer vision models.
>
> **Q4**: “While the Vision Transformer (ViT) serves as the backbone, the paper lacks clarity on how attention is utilized in the experiments. It would be beneficial for the community to understand the potential relationship between the foveated process and visual attention.”
>
> **A4**: This is a misunderstanding. How attention is used in transformers is explained in the original transformer paper, which we cited. Our paper doesn’t focus on analyzing the attention mechanism in Transformers and it is irrelevant to the foveation. The purpose of the foveation is restated in response to your question Q3 above. The attention in Transformer is also different from “visual attention”. Visual attention is also not what this work studies.
>
> **Q5**: “The comparison against the MAE seems somewhat rudimentary. For example, the reported performance on CIFAR-10 indicates that several top models significantly outperform the authors' results (Table 1): Rank Model Percentage correct 1 ViT-H/14 99.5 2 DINOv2 99.5 3 µ2Net 99.49 4 ViT-L/16 99.42 It seems the claimed better performance over the baseline was quite embarrassed by the top runners above.”
>
> **A5**: This is a misinterpretation of our work. These results are not relevant because our motivation is not to compete SOTA on CIFAR but to propose a new SSL method that can work under less priors. Additionally, their performances are likely the result of pre-training on larger image datasets such as Imagenet and fine-tuned on CIFAR10. The comparison is unfair. Our setting here is pretraining (SSL) the model on CIFAR10 and testing it on CIFAR10 with linear probing. Both Reviewer sHq9 and Reviewer o7jg think our experimental evidence is strong.

---

> ### Author Response · Authors · 2024-11-14
> **Author Response to Reviewer 3Rzb (2/2)**
>
> **Q6**: “The meaning and algorithm behind the permutation of CIFAR-10 images are unclear. Providing examples of permuted images would clarify this aspect. The rationale for permuting pixels should also be addressed, as such transformations may hinder human recognition and contradict the stated bio-inspired motivation.”
>
> **A6**: The example of a permuted image and how it's processed by URLOST is explained in line 391, figure 3, and figure 4. The meaning behind the permutation of CIFAR-10 images is explained in line 222. In short, we claim our algorithm works on data without topology or stationarity. Permuting image synthesizes dataset without topology.
>
> **Q7**: “In Section 2, the proposed method appears to aggregate several state-of-the-art techniques with limited novel mathematical contributions. Incorporating theoretical exploration of its bio-inspired aspects would significantly enhance the paper's quality.”
>
> **A7**: We want to emphasize combining spectral clustering, self-organizing layers, and masked autoencoders is a novel component of our work. This combination is simple, natural yet effective. There are other works on recovering the topology of the signal, like  [1] or [2]. However, these works cannot be effectively integrated with state-of-the-art self-supervised learning algorithms. Reviewer o7jg, Reviewer u6uR, and Reviewer t33q all think the method is novel, well-motivated and promising.
>
> [1] Roux, N, et al. 2007. Learning the 2-d topology of images.
>
> [2] Kohonen, T. 1990. The self-organizing map

---

> ### Comment · Reviewer_3Rzb · 2024-11-26
>
> Thanks for the authors' responses, which have clarified many details of the work. However, I am still struggling to fully grasp the exciting contributions or unique value this work presents.
>
> 1) Figure 3 illustrates the foveation of images but suggests that foveation worsens performance—a finding that seems contradictory to biological vision, where foveation is typically considered beneficial.
> 2) Figure 4 indicates that the permutation was applied by rotating patches; however, the pixels between patches and within patches remain unaltered. Since vision transformers take these patches as input sequences, it appears that no true permutation occurs beyond the patch rotations.
> 3) While it is evident that masked autoencoders can leverage additional data from patch rotations, the proposed approach seems to be a tweaked extension of masked autoencoders, with limited novelty demonstrated.
>
> I hope these points can help guide further clarification or refinement of the manuscript.

---

> ### Author Response · Authors · 2024-11-27
> **Author Followup Response to Reviewer 3Rzb (1/2)**
>
> Thanks for providing the feedback and for your positive comments! It is relatively straightforward to address these additional questions:
>
> __Q1__. "foveation of images but suggests that foveation worsens performance—a finding that seems contradictory to biological vision"
>
> __A1__: This is a misunderstanding of the foveation experiment. Foveation in this work is introduced to simulate the biological vision rather than to improve the performance. Foveation on existing CIFAR images is going to reduce the resolution and lose information since the center sampling is much denser than the peripheral. When resolution is reduced, it is natural to lose some accuracy.
>
> The major goal of this paper is to address the challenge that biological vision has to face rather than proposing that foveation is a superior strategy than regular pixel grid. In fact, biological vision is full of random evolution paths, resulting in differences in optics and sampling strategies across species [1][2]. Despite this variability, it is insightful to focus on two fundamental challenges that biological systems must overcome: the lack of topology and the absence of stationarity in visual inputs. This work shows it is actually possible to learn visual representation without leveraging the topological grid information and stationary statistics across the visual domain. Further, this method generalizes the unsupervised representation learning to high-dimensional signals without prior topology and stationarity.
>
> __Q2__. “Figure 4 indicates that the permutation was applied by rotating patches; however, the pixels between patches and within patches remain unaltered. Since vision transformers take these patches as input sequences, it appears that no true permutation occurs beyond the patch rotations.”
>
> __A2__: This is a misunderstanding of Figure 4. Figure 4 was provided to support the Ablation Study $-$ Visual Evidence in Section 4. In this ablation, we use the Locally Permuted CIFAR10 experiment to isolate the problem, where we patchify each image into small image patches, as shown in Figure 4A (Left). For image patches at different locations, we use different permutations to shuffle the pixels, as shown in Figure 4A (Middle). After the permutation, the pixel ordering in each patch looks random, as shown in Figure 4A (Middle).  Since the permutations at different locations are saved for visualization purposes, we can reverse the permutation in each patch and recover the original patch, as shown in Figure 4(A) (Right). This unpermute operation is highly useful to visualize the learned features for different patches. Without the unpermute operation, the learned features for different patches would also look random, as shown in Figure 4B (Left). In Figure 4B (Left), we visualize the 12th feature learned at each of the patch locations. These features shall convey similar meanings as their output are the corresponding values in the attention mechanism. Figure 4B (Right) shows this is indeed the case. After we use the unpermute operation to visualize these features, they appear to correspond to a similar structure. However, one interesting phenomenon is that they are not exactly the same. Reviewer o7jg made a very insightful comment that this might be due to the self-organizing layer having learned a different strategy for location encoding! This learned location encoding might be the reason that the same feature at different patches appears to be a rotation of each other. In Figure 4C, we show another feature and its unpermuted visualization. These features seem to have the same rotational relationship. We have added additional discussion to discuss this in further detail in Section 4.2.
>
>
>
> [1] Michael F. Land, Dan-Eric Nilsson, Animal Eyes, Oxford Animal Biology Series, 2012
>
> [2] Michale F. Land, The optical structures of animal eyes, Current Biology, 2005

---

> ### Author Response · Authors · 2024-11-27
> **Author Followup Response to Reviewer 3Rzb (2/2)**
>
> __Q3__. "While it is evident that masked autoencoders can leverage additional data from patch rotations, the proposed approach seems to be a tweaked extension of masked autoencoders, with limited novelty demonstrated."
>
> __A3__: It looks like this question is due to the same misunderstanding from Q2. We didn't introduce any such rotation as an augmentation or claim any superiority of the introduced permutation. We hope A2 has clarified sufficiently. We will revise accordingly to make these points clearer. Regarding novelty, the success of most current state-of-the-art self-supervised representation learning methods relies on the assumption that the data has known stationarity and domain topology. Unsupervised learning on data without stationarity or topology is a largely challenging and unsolved problem. Our method combines aggregating similar dimensions, self-organizing maps, and masked autoencoders. It is not just a “tweaked extension of masked autoencoder” but a simple yet effective solution to the challenging problem. We also want to note that URLOST is not specifically limited to masked autoencoders. As mentioned in the future directions and suggested by Reviewer o7jg, adapting URLOST to support contrastive learning objectives, such as SimCLR, is an interesting direction.

---

> ### Comment · Reviewer_3Rzb · 2024-11-27
>
> I’m not fully convinced about the novelty of the approach. As agreed by the authors, it is essentially a cascaded combination of two existing methods. The value of the permutation is unclear, and it needs to be known for all test datasets.
>
> The updated version is well-presented and tidy. The experimental results are encouraging, showing that the combined approach performs better. I appreciate the authors' responsiveness to questions. Therefore I would raise my score to 6.

---

> > ### Author Response · Authors · 2024-11-27
> > **Author Followup Response to Reviewer 3Rzb (1/2)**
> >
> > We thank the reviewer for the encouraging comments! We respect your judgement, but we think there are still some misunderstandings regarding the contribution of our work.
> >
> > We agree with the reviewer that URLOST combines existing methods, but both us and all other reviewers think the contribution of the submission is beyond. Granted, even highly cited paper like vision transformers simply combine existing methods like transformers and patchification. Similarly, MAE combines VIT and the masked prediction task. Nevertheless, these ideas are simple yet fundamental. They are so effective that they enable computation on a modality that is previously thought impossible. We think the idea of aggregating similar dimensions together, and combining it with MAE is also fundamental. We also believe the other three reviewers (t33q, o7jg, and sHq9) gave us a score of 8 because they see the great potential of this method. For example, reviewer t33q says this work provides “a novel way unsupervised representation learning pipeline.” Reviewer sHq9 says that “the motivation is sound and the experimental results are quite convincing. ”  Finally o7jg says “the premise to handle more diverse datasets than images with a ViT setup by clustering input components is strong.”
> >
> > Still, we respect the reviewer’s judgement, but just want to clear all the misunderstanding. Moreover, we invite the reviewer to take a look at our rebuttal conversation with other reviewers. They provide us with many good experiments to show the effectiveness of each module of URLOST. We included these experiments in the updated manuscript. For example, the model doesn’t work if you simply cascade patchification and masked-autoencoder. We also provide clarification for purpose of the permutation experiment in the following comments:

---

> > ### Author Response · Authors · 2024-11-27
> > **Author Followup Response to Reviewer 3Rzb (2/2)**
> >
> > “The value of the permutation is unclear, and it needs to be known for all test datasets.”
> >
> > The value of the permutation experiment is that it simply serves a sanity check. It shows our model works as expected. For example, it should reach expected accuracy, learn how to cluster pixels and align clusters with each other. The truth value of the method is shown in the following three experiments on: Foveated CIFAR10, gene and V1 data. Moreover, permutation does not need to be known and is actually not known for test datasets. The model is not designed to learn the permutation explicitly. Moreover, in most cases, there is no “correct” permutation. Instead, the model is learnt to aggregate similar dimensions together and learnt to align clusters with each other implicitly. Since the learnt cluster and self-organizing map is part of the model, we assume it has some train-test generalization ability. This is a standard assumption and verified in our experiment.

---

> ### Comment · Reviewer_3Rzb · 2024-12-01
>
> I feel that the experiments on foveated images, permuted images, and V1 & gene data are quite distinct in their focus. The last two effectively highlight the need for non-topological patterns and are strong examples in this regard.
>
> However, the permutation experiment seems to require further justification—it’s not entirely clear or convincing to me at this stage. Why is the permutation process defined in a specific way? Would other permutation algorithms yield similar results?
>
> As for the foveated images, the results seem somewhat underwhelming. Providing a strong rationale for how the foveation process can be understood as a product of biological evolution would strengthen the argument.

---

> ### Author Response · Authors · 2024-12-02
> **Author Followup Response to Reviewer 3Rzb**
>
> "However, the permutation experiment seems to require further justification—it’s not entirely clear or convincing to me at this stage. Why is the permutation process defined in a specific way? Would other permutation algorithms yield similar results?"
>
> We assume the reviewer is asking about Permuted CIFAR-10 experiment. In this experiment, the permutation is fully random and it is not in a specific way. So there are no other permutation algorithms.
>
> "Providing a strong rationale for how the foveation process can be understood as a product of biological evolution would strengthen the argument."
>
> As we mentioned earlier, the major goal of this paper is to address the challenge that biological vision has to face rather than proposing that foveation is a superior strategy than regular pixel grid. In fact, biological vision is full of random evolution paths, resulting in differences in optics and sampling strategies across species [1][2]. Despite this variability, it is insightful to focus on two fundamental challenges that biological systems must overcome: the lack of topology and the absence of stationarity in visual inputs. This work shows it is actually possible to learn visual representation without leveraging the topological grid information and stationary statistics across the visual domain. Further, this method generalizes the unsupervised representation learning to high-dimensional signals without prior topology and stationarity.
>
> [1] Michael F. Land, Dan-Eric Nilsson, Animal Eyes, Oxford Animal Biology Series, 2012
>
> [2] Michale F. Land, The optical structures of animal eyes, Current Biology, 2005

---

### Official Review · Reviewer_sHq9 · 2024-11-06

**Soundness:** 3
**Presentation:** 3
**Contribution:** 3
**Rating:** 8
**Confidence:** 4

**Summary:**

The paper investigates the topic of self-supervised representation learning and how to extend current popular frameworks to data modalities beyond natural images or text for which these methods have been initially designed. This is particularly relevant for the processing of scientific data which at this point would be hard to integrate into frameworks like MAEs. The authors propose to add a preprocessing block before the classic MAE which allows to aggregate data into the correct format even in the absence of prior information about the data's structure. This preprocessing blocks involves spectral clustering with density adjustment followed by a self-organizing map embedding layer. The authors show that the approach is more effective when process high-dimensional unstructured real-world data (2 natural science datasets + a modified version of CIFAR10) than applying the standard MAE framework to this data.

**Strengths:**

- relevance: this paper aims at extending existing SSL methods to novel modalities (notably modalities found in various scientific fields) for which current methods and their associated architecture of choice (e.g., MAE with ViT backbones) are not adapted because designed to suit standard data modalities (e.g., images) with known structure. This topic is relevant as SSL methods are often shown to be powerful but designed (both joint embedding methods and masked image modeling) to suit specific modalities which limit their applicability to a broader range of data. Even for standard modalities like natural images, approaches such as MAE rely on assumptions that might not generalize across datasets.
- presentation: paper is well written and easy to follow, the overall state of the paper is good.
- experimental evidence: the experimental evidence to support the proposed method is quite convincing (see weaknesses below for information and experiments missing) as the authors conduct experiments on 3 real-world datasets on which results show superiority to the standard MAE baseline which is the most relevant related work with confidence intervals.

**Weaknesses:**

- sensitivity of the method to choice of hyperparameters: the proposed method relies on spectral clustering for which a number of hyper parameters should be defined (like the number of clusters and hyperparameters linked to cluster density). These hyperparameters seem crucial in order to achieve high performance and are data-driven.
- motivation: while it is clear that an aggregation/clustering of the input dimensions is necessary, the intuition behind aggregating dimensions that cluster together is missing; While the proposed method seems to work well, the paper would benefit from some additional intuition as to why one might want to combine elements that are similar in the same patch and compare this approach to the information found in standard image patches. There is little understanding of how MAE work, and most of their design choices are empirically driven, therefore it remains unclear whether MAE work better when pixels within a patch are identical.
- missing experimental baseline numbers and details: multiple information regarding the experimental setup is missing thereby reducing the ability to judge the soundness of the experimental setup (see question below) and a couple of experiments seem to be missing in order to confidently conclude that the proposed method is an effective alternative to the standard MAE baseline (see questions below)
- missing information about the work's limitations which might include the computational cost and scalability of the proposed method.

**Questions:**

- can authors elaborate on why a set of linear projectors (non-shared in this case, a different layer for each cluster) cannot replace the use of self-organising maps? seems like the point here is that operation should not be shared between clusters rather than the type of operation used.
- how were representations trained in table 3? standard MAE?
- can authors provide numbers for 1) URLOST with CIFAR10 2) MAE (Patch) with Permuted CIFAR and Foveat CIFAR;
- can authors explain by such a high number of epochs is needed (10,000), is this only necessary for URLOST, prior work recommend 800-1600 epochs for standard MAEs.
- what is the dimensionality of the representation in the beta-VAE vs the MAE; how do the size of models compare? the VAE used seems very shallow.
- what is the range of beta parameter that was considered?
- is the MAE in table 2 also with patch size 4?
- what is the masking ratio in the MAE and URLOST (75%), can authors provide a comparison with 0% masking to show a non-masking scenario with equivalent architecture ?
- how is k selected ? how variable are results _across datasets_ for varying k, alpha, and beta parameters ?

Minor:
- typo in line 55
- bold number in table 1 for CIFAR10 is wrong, should be SimCLR

---

> ### Author Response · Authors · 2024-11-23
> **Author Response to Reviewer sHq9 (1/6)**
>
> Dear reviewer sHq9, we want to thank you for putting lots of effort in reviewing our paper and helping us shape our work better. You made many valuable suggestions. We modified our manuscript and ran several experiments as you suggest. We also provide additional details on implementation and intuitive for selecting hyper-parameters.
>
> __Additional experiment__: \
> Adding additional ablation study “MAE (Patch)” for permuted CIFAR and foveated CIFAR.\
> Adding grid search results for hyperparameters of spectral clustering in Appendix A.5. \
> Adding experiments for 0% masking.
>
> __Change we made in manuscript__:\
> Added the intuition on how to select clustering hyperparameters in experiments (line 208) and Appendix A.6\
> Added explanation on why aggregating dimensions to make clusters in the ablation study section.\
> Explained MAE baseline for v1 and gene data is essentially forming patches randomly as input in section 3.2.\
> Added a section on computational cost, scalability and limitations. \
> Added how self-organizing layers are parametrized for all experiments (line 208) .\
> Added additional details for the beta-vae baseline in Appendix A.5.

---

> ### Author Response · Authors · 2024-11-23
> **Author Response to Reviewer sHq9 (2/6)**
>
> __Q1__: “The proposed method relies on spectral clustering for which a number of hyperparameters should be defined. These hyperparameters seem crucial …”
>
> __A__: Indeed spectral clustering or clustering algorithms in general have a number of hyperparameters. In our case, parameters like “density factors (how nodes are normalized)” and “number of clusters” affect the number and the size of the clusters, thus, the performance of the model to some degree. In general, the model is not very sensitive to these hyperparameters. To apply URLOST, one only needs to make sure these hyperparameters are not too extreme. We added the intuition on how to select them in the manuscript. We also add additional experiments to show the effect of these parameters to the model's performance:
>
> **Table 1: \(n\) vs Accuracy**
>
>
> | hyperparameter = \(n\) | 10    | 20    | 30    | 40    | 50    | 60    | 70    | 80    | 90    | 100   |
> |-------------------------|-------|-------|-------|-------|-------|-------|-------|-------|-------|-------|
> | Accuracy (\%)          | 90.5  | 94.5  | 95.2  | 94.6  | 95.2  | 94.5  | 93.6  | 94.7  | 93.9  | 92.9  |
>
>
> ---
>
>
> **Table 2: \(K\) vs Accuracy**
>
>
> | hyperparameter = \(K\) | 4     | 8     | 12    | 16    | 20    | 24    | 28    |
> |-------------------------|-------|-------|-------|-------|-------|-------|-------|
> | Accuracy (\%)          | 95.2  | 95.4  | 95.2  | 95.4  | 95.4  | 95.1  | 95.3  |
>
>
> ---
>
>
> **Table 3: \(\beta\) vs Accuracy**
>
>
> | hyperparameter = \(\beta\) | 0.5   | 0.8   | 0.9   | 0.95  | 1     | 1.05  | 1.1   | 1.2   | 1.5   | 2     |
> |----------------------------|-------|-------|-------|-------|-------|-------|-------|-------|-------|-------|
> | Accuracy (\%)             | 85.3  | 93.7  | 94.8  | 95.3  | 94.7  | 94.9  | 95.1  | 95.3  | 94.3  | 93.5  |
>
> Intuition: Both these hyperparameters are related to the size of resulting clusters, which relates to how difficult the task is. Taking the image as an example, if the cluster is too big, then the model needs to learn to predict a big missing area in the image, making the self-supervised learning task too difficult. The model will end up not learning any useful representation. On the other hand, if the cluster is too small, for example, if the cluster shrinks down to a few pixels, then the task becomes too easy. The model can learn to fill in the missing pixels by averaging the value of nearby pixels. The model essentially learn to become a low-pass filtering for images. In both cases, the model will not learn high quality representations. Same intuition applies to other high dimensional signals. We recovered the topology by clustering based on the pairwise mutual information between dimensions of the signals.  Both the number of clusters and density factors are related to the cluster size. Therefore, to make the self-supervised learning task effective, we set the hyperparameters in order to make sure the size of the cluster is not too big or too small.
>
> Increasing the number of clusters could make each cluster smaller. The density factor controls the variance of the cluster sizes by applying proper normalization to the graph during spectral clustering to make the cluster size more evenly distributed. For an image dataset, clusters should contain semantic sub-structures of the image, like body parts or parts of objects. We pick 64 clusters, which results in clusters of size roughly 4x4. In other words, the compression ratio between the clusters (patches) and the raw signals (image) is roughly 1:64. For other dataset (gene and V1), we perform a grid search around this ratio. Similarly, for the density factors, we also perform a grid search centered at the density factor equal to 1, which is the default normalization used in spectral clustering.

---

> ### Author Response · Authors · 2024-11-23
> **Author Response to Reviewer sHq9 (3/6)**
>
> __Q2__: “The intuition behind aggregating dimensions that cluster together is missing. it remains unclear whether MAE works better when pixels within a patch are identical.”
>
> __A__: Thank you for the valuable suggestion. As suggested by the reviewer, We provide intuition on why similar dimensions should be aggregated in the manuscript. We also provide additional experiments to show the performance difference between aggregating random dimensions vs aggregating similar dimensions.
>
> One reason for aggregating dimensions at all is to reduce the computational complexity. With a transformer, naive application of self-attention to general high-dimensional signals would require each dimension to attend to every other dimension [1]. This can be computationally expensive. Another reason, more importantly, is to create a nontrivial unsupervised learning task for the model. If we don’t aggregate dimensions at all, the task becomes predicting random missing dimensions. This will lead to trivial solutions. To fill in the missing value of a dimension, the model tends to learn to use the averaged value of similar dimensions. As explained in the previous questions, this is similar to perform a low-pass filtering on the high dimensional signal. In other words, the model learns to solve the task using only low-level information.
>
> Why aggregating similar dimensions instead of aggregating random dimensions? We assume similar dimensions are used to sampling similar region of the underlying signals. Clustering of similar dimensions contains underlying patterns. In order to predict the masked clusters from unmasked clusters, the model needs to learn the underlying patterns from each cluster. For example, for the CIFAR10 example, each dimension of the signal represents a pixel. Similar pixels tend to sample semantically similar regions in the image due to its locality, which together form image patches. For V1 data, similar neurons likely share a similar receptive field. Each v1 neuron codes a simple pattern like oriented edge in the receptive field. Similar neurons together could code patterns like shape and contours, which is more complicated than edge. Masking and predicting at the shape and contours level is a non-trival SSLtask. We also provide experiments on what if we aggregate random pixels into patches. The performance significantly degraded. This served as a baseline in table 2. Here’s experiments we ran for comparing aggregating random dimensions into patches with URLOST. We aggregate all the experiment we ran to support this argument:
>
>
>
> | Dataset             | Patchification                            | Performance |
> |---------------------|-------------------------------------------|-------------|
> | Permuted Cifar10    | URLOST MAE          | 86.4%       |
> | Permuted Cifar10    | MAE (patch)            | 55.7%       |
> | Foveated Cifar10    | URLOST MAE          | 85.4%       |
> | Foveated Cifar10    | MAE (patch)             | 51.1%       |
> | TCGA Gene           | URLOST MAE          | 94.9%       |
> | TCGA Gene           | MAE (patch)            | 91.7%       |
> | V1 Response         | URLOST MAE          | 78.8%       |
> | V1 Response         | MAE (patch)             | 73.9%       |
>
>
>
>
>
> __Q3__: “missing information about the work's limitations which might include the computational cost and scalability of the proposed method”
>
> __A__：Thank you for the suggestion. We provide an additional section in the appendix A.10 to discuss the computational cost and scalability. In short, the computational cost and scalability of URLOST is nearly the same as MAE.
>
>
> __Q4__: “can authors elaborate on why a set of linear projectors (non-shared in this case, a different layer for each cluster) cannot replace the use of self-organizing maps? seems like the point here is that operation should not be shared between clusters rather than the type of operation used.”
>
> __A__: Your intuition is correct. We use a set of non-shared linear projectors for the self-organizing layers. We name them as self-organizing layers (equation 2, line 185) This can be done with linear projectors but not limited to linear projectors, and we want to refer to it for the functionality rather than the specific architecture choice.. Like you said, the point here is to have a set of parametrized transforms with non-shared parameters. We call this set of non-shared projection layers “self-organizing layers” because they learn to organize each patch to align them in an unsupervised fashion. And this idea is similar to the original problem set up in the self-organizing map [2] paper.
>
> The parametrized transform needs to have enough capacity to align the set of clusters. Empirically, we find that the linear layer gives enough modeling capacity. This is briefly mentioned in 4.1 visual evidence. As suggested by the reviewer, we revise the manuscript to make this design choice explicit in the experiment sections.

---

> ### Author Response · Authors · 2024-11-23
> **Author Response to Reviewer sHq9 (4/6)**
>
> __Q5__: can authors explain by such a high number of epochs is needed (10,000), is this only necessary for URLOST, prior work recommend 800-1600 epochs for standard MAEs.
>
> __A__: We also recommend users to train around 3000 epochs. The performance plateaus after 3000 to 5000 training epochs (e.g. on permuted CIFAR: 3000-> 80.8, 5000->84.1, 10000->86.4). We only ran it longer because there’s still some very small performance gain. Based on the observation across all three datasets, we recommend 3000 epochs as a common choice for URLOST.
> We do want to emphasize that the recommendation of 800-1600 epochs in the original MAE paper is for ImageNet. The ImageNet has roughly 20 times more images than the CIFAR10 dataset. In general, for unsupervised learning tasks, training on CIFAR10 takes much more epochs than training on imagenet. We ran our baseline such as MAE with the same number epochs as URLOST.
>
>
> __Q6__: “how were representations trained in table 3? standard MAE? Can authors provide numbers for 1) URLOST with CIFAR10 2) MAE (Patch) with Permuted CIFAR and Foveat CIFAR?”
>
> __A__: The representations are different for each row in table 3 for ablation study. We break down the pipeline of URLOST as the following: patchification, self-organizing layer and mask prediction task. We ablate the patchification method and self-organizing layer to show the effectiveness of these two designs. We re-organized the ablation experiment as the following, would the reviewer think the following table would be a better presentation?
>
> | Exp index | Dataset                   | Patchification                                | Self-organizing layer | Task             | Probing Accuracy |
> |-----------|---------------------------|-----------------------------------------------|-----------------------|------------------|------------------|
> | 1         | Locally-permuted Cifar10  | 4x4 image patch                               | Yes                   | Mask prediction  | 87.6%           |
> | 2         | Locally-permuted Cifar10  | 4x4 image patch                               | No                    | Mask prediction  | 81.4%           |
> | 3         | Permuted Cifar10          | Spectral clustering (no normalization)        | Yes                   | Mask prediction  | 86.4%           |
> | 4         | Permuted Cifar10          | Spectral clustering (no normalization)        | No                    | Mask prediction  | 80.7%           |
> | 5         | Permuted Cifar10          | Randomly aggregated pixels (cluster size: 16) | No                    | Mask prediction  | 55.7%           |
> | 6         | Foveated Cifar10          | Spectral clustering (density adjusted)        | Yes                   | Mask prediction  | 85.4%           |
> | 7         | Foveated Cifar10          | Spectral clustering (no normalization)        | Yes                   | Mask prediction  | 82.7%           |
> | 8         | Foveated Cifar10          | Randomly aggregated pixels (cluster size: 16) | No                    | Mask prediction  | 51.1%           |
>
> Additionally, we agree with the reviewer that MAE (Patch) is a good ablation study to run. We want to remind the reviewer that we did use Patch MAE (aggregating random nodes together) as a baseline for gene and V1 dataset. This is the “MAE” entry in table 2. However, we did not run this experiment on CIFAR10. With this additional experiment suggested by the reviewer, we can show aggregating similar nodes together >  aggregating random nodes together >> do not aggregate nodes together. We want to acknowledge the reviewer for this wonderful feedback. This helps us to deliver our message better. The experimental result also answers the reviewer's previous question on why aggregating similar nodes. In the following table, exp 7 and exp 8 are the new experiment, which corresponds to MAE (Patch) with self-organizing layer on permuted and foveated CIFAR10.

---

> ### Author Response · Authors · 2024-11-23
> **Author Response to Reviewer sHq9 (5/6)**
>
> __Q7__: “what is the dimensionality of the representation in the beta-VAE vs the MAE; how do the size of models compare? the VAE used seems very shallow.”
>
> __A__: The neural network we used for VAE and MAE has similar size, as described in Appendix A.5. For VAE, we use a two-layer MLP encoder and two-layer MLP decoder. Additionally, we have two additional linear layers to map latent embedding to latent to posterior mean and std). For MAE, we use four layers encoder and two layers decoder. The hidden dimension we used for MAE is 1380 and 384 for V1 and Gene dataset correspondingly. The hidden dimension we used for VAE is 2048 and 1024 for V1 and Gene dataset. As suggested by the reviewer, we added this model details in the appendix.
>
>
> __Q8__: “what is the range of beta parameters that was considered?"
>
> __A__: The range is [0,10]. we find the best performing beta = 0.1. We want to emphasize we did extensively hyperparameter tuning for our baseline model. For beta-vae, the performance of our model is on par even with the performance of the fine-tuned beta-vae model (supervised training) [3].
>
> __Q9__: “is the MAE in table 2 also with patch size 4?”
>
> __A__: No, for V1 and gene dataset, the size of each cluster (patch size) is decided automatically by the spectral clustering algorithm. This is the same for permuted and foveated CIFAR10 as well. For all these dataset with no topology, we use “number of clusters” instead of “size of clusters” as the hyperparameters. The size of clusters are reported in table 2.

---

> ### Author Response · Authors · 2024-11-23
> **Author Response to Reviewer sHq9 (6/6)**
>
> __Q10__: “what is the masking ratio in the MAE and URLOST (75%), can authors provide a comparison with 0% masking to show a non-masking scenario with equivalent architecture ?”
>
> __A__: We use 75% for both MAE and URLOST, as suggested in the original MAE paper [4]. 0% masking makes the training objective equivalent to an autoencoder. We did an additional experiment for 0% masking on permuted cifar10. The linear probing accuracy is 32.6%. This is expected because autoencoder is generally not a competitive unsupervised learning algorithm.
>
> __Q11__: “how is k selected ? how variable are results across datasets for varying k, alpha, and beta parameters ?”
>
> __A__: We perform grid search to select k. In fact, the clustering or the performance of the model is not sensitive to the section of k. The performance of the model when varying alpha and beta for foveated CIFAR10 dataset is shown in table 4 in the appendix. We perform additional experiments to show the effectiveness of varying these hyperparameters. Please see the table at the first part of the response (response 1/6).
>
>
> [1] Dosovitskiy, Alexey, et al. 2020. An Image is Worth 16x16 Words: Transformers for Image Recognition at Scale
> [2] Kohonen, T. 1990. The self-organizing map.
> [3] X, Zhang, et al. 2021. OmiEmbed: A Unified Multi-Task Deep Learning Framework for Multi-Omics Data
> [4] He, Kaiming, et al. 2021. Masked Autoencoders Are Scalable Vision Learners

---

> > ### Comment · Reviewer_sHq9 · 2024-11-25
> > **Answer to rebuttal**
> >
> > Thank you to the authors for the in-depth answer.
> >
> > Has more intuition about the proposed method been added to the updated manuscript? I cannot seem to find it.

---

> ### Author Response · Authors · 2024-11-25
> **Author Follow-up Response to Reviewer sHq9**
>
> Dear reviewer,
>
> Thanks for the reminder. We forgot to upload the updated ablation study section. We now uploaded the new section in the revised manuscript. Would you please take a look and provide us some feedback? We made an entirely new ablation study section to talk about the intuition for aggregating similar dimensions. As the reviewer suggested, since aggregating similar dimensions and processing them together is the heart of our method, we believe it should be the first thing in the ablation study. We also include additional experiments for masking individual dimensions instead of patches for V1 and Gene dataset.
>
> We want to acknowledge Reviewer sHq9 for helping us to come up with this additional section in ablation study. It really helps us to shape our work better.
>
> Other additional intuition we included during the rebuttal phase is scattered in the manuscript. For example:
>
> In line 96, we provide intuition for why permuting images destroy their topology.
>
> In Appendix. A9, we provide intuition on the clustering of hyperparameters that could potentially affect the performance of the model.
>
> In Appendix. A8, we provide intuition on data with topology but no stationarity, and data with stationarity but no topology.

---

> > ### Comment · Reviewer_sHq9 · 2024-11-26
> > **Follow-up response**
> >
> > Thank you. I think adding this ablation study helps with conveying the intuition behind the proposed approach.
> > The current manuscript is way above the page limit at this stage so I believe authors should readjust the content of the manuscript to fit the 10 pages limit. The text for the ablation study could also be improved.
> >
> > All in all, I think the authors have addressed my concern regarding the sensitivity over clustering parameters and I am happy with the ablation study proposed by the authors.
> >
> > I believe the novelty of this contribution is relatively limited but the motivation is sound and the experimental results are quite convincing. I think a score of 7 would have accurately described my opinion of this paper. Since there is no 7, I am happy to increase to 8 for an updated version of the manuscript that fits nicely in the page limit.

---

> ### Author Response · Authors · 2024-11-26
>
> We want to thank you so much for your encouragement! For the rebuttal revision, we simply want to include all helpful suggestion from reviewers first, and then adjusting the format. We now trims the manuscript to fit the 10 pages limit. Please check the updated manuscript. We would love to now if you have any additional suggestions. Here's specific list of revision we made to fit in 10 pages limit:
>
> 1. Transposed the table 3 in additional ablation study to make it use the space more efficiently. This saves the most space.
>
> 2. Make the intuition for why aggregating similar dimension more concise. Remove the argument on how making patch alleviate the quadratic cost of transformer. This part seems relatively obvious. Focus more on how aggregating similar dimension creates a better SSL tasks.
>
> 3. Make the additional discussion on flexiVIT and patchGT more concise in related work section.
>
> 4. Move the formulation of SOM from related work to Appendix A.12. This seems less relevant to our work and takes lots of space. But we still keep a small section of SOM in related work.
>
> 5. Trim the much of data synthesize process of foveated CIFAR10. We realized the data synthesize process in method section largely overlap with the data synthesize section in the original Appendix.
>
> 6. Adjusting spacing around tables.
>
> So far I think we address all the reviewer's suggestion and the manuscript fit in the 10 page limits. We still think there's room to make the manuscript even more concise by adjusting the wording. We will polish the wording for the camera-ready version of the paper.

---

> > ### Comment · Reviewer_sHq9 · 2024-11-26
> >
> > Thank you, I have updated my score accordingly.

---

> > > ### Author Response · Authors · 2024-11-27
> > >
> > > Thank you for increasing the score. We greatly appreciate your suggestion on how to revise our manuscript. The additional ablation study section will definitely improve the quality of our manuscript!

---

### Author Response · Authors · 2024-11-23
**General reply (1/3)**

__Acknowledgements__

We thank all our reviewers for their encouraging comments, helpful suggestions, and insightful questions. All reviewers agree that our work is well-motivated and can benefit the community by tackling an important problem. For example,  reviewer sHq9 says our method extending “SSL methods to novel modalities” is very relevant to “various scientific fields.” reviewer o7jg says “The premise to handle more diverse datasets than images is strong” and our experiments “cover a good range of complexity” and are “sufficient to demonstrate the utility of the method.” Additionally, reviewer u6uR and reviewer t33q say our idea is “... novel and well-motivated. ” They think the task we tackled a problem that “is interesting, important and relevant to the community.”

__Questions and suggestions__

Reviewers’ questions and suggestions are very constructive. They help us revise the paper to deliver key messages better. Most of the weaknesses listed by the reviewer can be resolved with further clarification and additional experimental results. In fact, most of the weaknesses listed by the reviewer can be resolved with further clarification. Reviewer sHq9 points out additional ablation experiments we should run to make our work more complete, including the effect of clustering hyperparameters and MAE (patches) on permuted and foveated CIFAR. As suggested by reviewer u6uR, we modified the draft with more elaboration to make the message more accessible to reach a broad audience from the general ML community. We would like to especially thank the dedication and amazing review from reviewer o7jg, not only for their in-depth review indicating they read our manuscript very carefully, but also for pointing out intriguing observations about the features learned in our model which is a great inspiration to us.

We provide a comprehensive response for each reviewer individually. We believe we addressed all the weaknesses. In the following general reply, we list some added experiments of common interest, and a list of revisions to the manuscript. We uploaded the revised manuscript. If reviewers have the time, we would be grateful for your further comments and/or questions.

---

> ### Author Response · Authors · 2024-11-23
> **General reply (2/3)**
>
> __Change in the manuscript:__
>
> Added the intuition on how to select clustering hyperparameters in experiments (line 208) and Appendix A.6\
> Added additional experiments and the results for a range of hyperparameters in Appendix A.7.\
> Added  more explanation on the motivation behind aggregating dimensions to make clusters in the ablation study section (section 4).\
> Added explanation to the MAE baselines for v1 and gene data in section 3.2. [line 337] The MAE baseline takes patch from randomly aggregated dimensions\
> Added a section on computational cost, scalability and limitations in Appendix A.9.\
> Added additional details on how self-organizing layers are parametrized for all experiments (line 208) .\
> Added additional details for the beta-vae baseline in Appendix A.5. \
> Included the finding about positional encoding and self-organizing layer in ablation study section (line 458)\
> Include discussion of FlexiViT, Patch VIT in related work. (line 564) \
> Include potential to extend URLOST to contrastive learning methods like SimCLR in future direction. (line 603) \
> Fixed typo at Line 265.\
> Fixed bolding typo in Table 1.\
> Including additional experiments for grid-searching the hyperparameter for clustering number and density factors in Appendix A.7\
> Added formal definition of stationarity and topology in the context of our work in Appendix A.9.\
> Added an example of data with “topology but no stationarity in Appendix A.9.\
> Added explanation for why permuted cifar10 has stationarity but no topology in Appendix A.9.\
> We modified self-supervised learning baseline to unsupervised representation learning baseline in the experiment section.\
> Added a section of computational cost in Appendix A.10.
> Added benchmark for each computational module of URLOST (Mutual information, clustering, encoder, decoder) in Appendix A.10.
> Added noise robustness experiments.

---

> ### Author Response · Authors · 2024-11-23
> **General reply (3/3)**
>
> __Additional experiments of common interest__:
>
> Additional experiments “MAE (Patch)” for permuted CIFAR and foveated CIFAR.\
> Added results for sweeping hyperparameter choices of spectral clustering in Appendix A.5. \
> Adding experiments for 0% masking ratio.\
> Adding noise robustness to mutual information experiments in Appendix A.11. \
>
>
>
> __Additional Exp 1: Hyperparameters of spectral clustering__
>
>
> We perform additional experiments doing grid search over each hyperparameter in spectral clustering that affects the performance of URLOST. The grid search experiment is performed over the TCGA gene classification dataset. As shown in the table below, the model is not sensitive for most of the hyperparameters. For a large range of hyperparameters, the model performance stays the same. Due to the time constraints, we are still working on the same experiments on V1 and CIFAR and we plan to add them later or in camera ready.
>
>
> **Table 1: \(n\) vs Accuracy**
>
>
> | hyperparameter = \(n\) | 10    | 20    | 30    | 40    | 50    | 60    | 70    | 80    | 90    | 100   |
> |-------------------------|-------|-------|-------|-------|-------|-------|-------|-------|-------|-------|
> | Accuracy (\%)          | 90.5  | 94.5  | 95.2  | 94.6  | 95.2  | 94.5  | 93.6  | 94.7  | 93.9  | 92.9  |
>
>
> ---
>
>
> **Table 2: \(K\) vs Accuracy**
>
>
> | hyperparameter = \(K\) | 4     | 8     | 12    | 16    | 20    | 24    | 28    |
> |-------------------------|-------|-------|-------|-------|-------|-------|-------|
> | Accuracy (\%)          | 95.2  | 95.4  | 95.2  | 95.4  | 95.4  | 95.1  | 95.3  |
>
>
> ---
>
>
> **Table 3: \(\beta\) vs Accuracy**
>
>
> | hyperparameter = \(\beta\) | 0.5   | 0.8   | 0.9   | 0.95  | 1     | 1.05  | 1.1   | 1.2   | 1.5   | 2     |
> |----------------------------|-------|-------|-------|-------|-------|-------|-------|-------|-------|-------|
> | Accuracy (\%)             | 85.3  | 93.7  | 94.8  | 95.3  | 94.7  | 94.9  | 95.1  | 95.3  | 94.3  | 93.5  |
>
> __Additional Exp 2__:
>
> We also provide experiments on aggregating random pixels into patches vs aggregating pixels using URLOST. The performance significantly degraded. This served as an additional baseline in table 2:
>
> | Dataset             | Patchification                            | Performance |
> |---------------------|-------------------------------------------|-------------|
> | Permuted Cifar10    | URLOST MAE          | 86.4%       |
> | Permuted Cifar10    | MAE (patch)             | 55.7%       |
> | Foveated Cifar10    | URLOST MAE          | 85.4%       |
> | Foveated Cifar10    | MAE (patch)            | 51.1%       |

---

### Meta-Review · Area_Chair_JREL · 2024-12-23

**Metareview:**

This paper introduces a novel framework for learning from high-dimensional data without prior knowledge of stationarity or topology. The model integrates a learnable self-organizing layer, spectral clustering, and a masked autoencoder (MAE). Its effectiveness is demonstrated across three diverse data modalities: simulated biological vision data, neural recordings from the primary visual cortex, and gene expressions. This work presents an innovative unsupervised representation learning pipeline with sound motivation and convincing experimental results.

The rebuttal effectively addressed the major concerns raised by the reviewers. The paper received three acceptances, one borderline decision, and one rejection. Despite some disagreements during the discussion, the AC believes the authors adequately explained and justified their claims. The AC recommends accepting the paper and urges the authors to revise it to incorporate key points discussed during the rebuttal phase.

**Additional Comments On Reviewer Discussion:**

Reviewer sHq9 found the paper to be well-written and supported by convincing experimental evidence. Initial concerns included the lack of hyperparameter analysis, unclear motivation, missing baseline numbers and details, and insufficient discussion of limitations. However, the authors provided additional experiments and revisions that addressed these concerns. Following the rebuttal, Reviewer sHq9's issues were resolved, and recommended accepting the paper.

Reviewer 3Rzb acknowledged that the authors addressed all technical concerns raised during the review. However, Reviewer 3Rzb remained unconvinced about the novelty of the approach, viewing it as a cascaded combination of two existing methods. As a result, Reviewer 3Rzb marginally rejected the paper.

Reviewer o7jg, after the rebuttal, was fully satisfied with the authors' responses and voted for acceptance. Reviewer o7jg also disagreed with much of Reviewer 3Rzb's assessment, arguing that the perceived weaknesses were either off-base or inflated.

Reviewer u6uR expressed concerns about the computational complexity of the proposed method and sought clarification on the terminology "topology/stationarity." Despite an intense debate, no agreement was reached, and Reviewer u6uR ultimately voted for rejection.

Reviewer t33q considered the motivation and contributions to be significant. With all concerns addressed by the authors, Reviewer t33q voted for acceptance.

Overall, the paper received one borderline score, one rejection, and three acceptances. The AC believes Reviewer 3Rzb's concern about the contribution is weak. While the method combines existing techniques, it does so in a simple and effective way, with sound motivation and convincing experimental results. Reviewers t33q, o7jg, and sHq9 championed the paper.

Regarding Reviewer u6uR's concerns, the AC feels that the authors provided adequate explanations. Since the discussion extended past the revision deadline, the AC is confident the authors can address these points in a future revision. Therefore, the AC recommends accepting the paper.

---

### Decision · Program_Chairs · 2025-01-22

Accept (Poster)